# Updated European hydraulic pedotransfer functions with communicated uncertainties in the predicted variables (euptfv2)

Brigitta Szabó[1], Melanie Weynants[2], Tobias KD Weber[3]

[1]Institute for Soil Sciences and Agricultural Chemistry, Centre for Agricultural Research, Herman Ottó út 15, 1022 Budapest, Hungary. ORCID: 0000-0003-1485-8908
[2]European Commission Joint Research Centre, via Enrico Fermi 2749, 21027 Ispra, Italy. ORCID: 0000-0002-1447-0105
[3]Institute of Soil Science and Land Evaluation, University of Hohenheim, Emil-Wolff-Straße 27, 70593 Stuttgart, Germany. ORCID: 0000-0002-3448-5208

*Correspondence to*: Brigitta Szabó (toth.brigitta@atk.hu)

**Abstract.** Soil hydraulic properties are often derived indirectly, i.e. computed from easily available soil properties with pedotransfer functions (PTFs), when those are needed for catchment, regional or continental scale applications. When predicted soil hydraulic parameters are used for the modelling of the state and flux of water in soils, uncertainty of the computed values can provide more detailed information when drawing conclusions. The aim of this study was to update the previously published European PTFs (Tóth et al., 2015, euptf v1.4.0) by providing prediction uncertainty calculation built into the transfer functions. The new set of algorithms was derived for point predictions of soil water content at saturation (0 cm matric potential head), field capacity (both -100 and -330 cm matric potential head), wilting point (-15,000 cm matric potential head), plant available water, and saturated hydraulic conductivity, as well as the Mualem-van Genuchten model parameters of the moisture retention and hydraulic conductivity curve. The minimum set of input properties for the prediction is soil depth and sand, silt and clay content. The effect of including additional information like soil organic carbon content, bulk density, calcium carbonate content, pH and cation exchange capacity were extensively analysed. The PTFs were derived adopting the random forest method. The advantage of the new PTFs is that they i) provide information about prediction uncertainty, ii) are significantly more accurate than the euptfv1, iii) can be applied for more predictor variable combinations than the euptfv1, 32 instead of 5, and iv) are now also derived for the prediction of water content at -100 cm matric potential head and plant available water content. A practical guidance on how to use the derived PTFs is provided.

## 1 Introduction

Quantitative information on state and flux of water in the critical zone is important for a wide range of environmental process models and decision support systems related to land surface processes (Lin, 2010; Zhao et al., 2018). Performance of hydrologic, climate, crop and other models related to soil hydrological processes depends on the quality and resolution of soil hydraulic input parameters (Vereecken et al., 2015). Simulations of flow through variably saturated soil media either rely on

simple modelling approaches which only require few directly measureable input variables such as porosity, field capacity, and wilting point, or on the Richards equation. While the former are simple and straightforward to obtain, the Richards equation requires knowledge about the soil hydraulic properties over the full moisture range. In practice, one of the most common approaches to describe the water retention and hydraulic conductivity curves required to solve the Richards equation is

arguably (Weber et al., 2019) the Mualem-van Genuchten model (MVG) (van Genuchten, 1980; Mualem, 1976). Since soil hydraulic measurements in the laboratory or in the field are often time consuming, expensive and difficult, indirect methods for estimating soil hydraulic properties using widely available surrogate data have been developed (Schaap, 2006). To date, a large number of pedotransfer functions have become popular to predict soil hydraulic properties and MVG model parameters (Van Looy et al., 2017).

Information on the uncertainty of the predicted soil hydraulic properties is important for modelling the state and flux of water in soil. The source of prediction uncertainty can be threefold: it can stem from the i) predictor (e.g. measurement uncertainty, non-representativeness of a sample), ii) predicted variables (e.g. uncertainty in the estimated soil hydraulic model parameters), and the iii) algorithm which describes the relation between the two. Information on the uncertainty of the predictor variables is commonly not available in PTFs derived before the 2000s, but has become a more intensively studied topic in the last decade.

For example, Weynants *et al.* (2009) quantified uncertainty of derived PTFs related to experimental, model and fitting errors with the one-step inversion method. Deng et al. (2009) differentiated and quantified intrinsic and input uncertainty of PTFs. Tranter et al. (2010) developed an uncertainty estimation method using fuzzy k-means with extragrades classification that can be applied in any PTF prediction. Kotlar et al. (2019) presented uncertainty assessment of PTFs through deriving PTFs on tens of resamples for train and test sets. Román Dobarco *et al.* (2019) introduced prediction interval coverage probability to assess

prediction uncertainty in PTFs derived on French soils. McNeill et al. (2018) provided estimation of the distribution and confidence intervals of the predicted soil hydraulic property (i.e. water content at 100 cm and 15,000 cm matric potential head and total available water). In the field of soil mapping it is an even more extensively studied topic where different computational methods have been proposed to assess uncertainty of the mapped properties. Examples are estimation of the 90% prediction intervals based on a triangular distribution (Odgers *et al.* 2014), quantification of mapped soil properties

uncertainties by quantile regression forest (Vaysse and Lagacherie, 2017), and a detailed comparison of uncertainties in mapped soil organic carbon content by different geostatistical and machine learning methods (Szatmári and Pásztor, 2019). Machine learning methods can be more robust to construct PTFs in comparison to previous approaches such as linear regression or simple decision trees if relationship between the predictors and response is highly non-linear (Araya and Ghezzehei, 2019). The random forest algorithm (Breiman, 2001) is able to outperform other machine learning methods (Olson

et al., 2018), which was also shown for predicting soil properties (Hengl et al., 2018; Nussbaum et al., 2018). Improvements in computing power, statistical methods and statistical software provide the possibility to apply more easily even complex models on large datasets. Therefore, complexity of a prediction algorithm is no longer a barrier in selecting a suitable algorithm to develop and apply PTFs. Most of the recent machine learning algorithms have the built in possibility to compute the uncertainty in the predicted variable, e.g. by quantile regression forest (Meinshausen, 2006) or generalized boosted regression

(Ridgeway, 2017). If PTFs are derived with these algorithms, the uncertainty of the predicted soil property can be directly estimated when applying the PTF (Szabó et al., 2019a), although this could also be achieved by applying the above mentioned uncertainty assessment methods without using machine learning methods (e.g. Kotlar et al., 2019; Tranter et al., 2010).

Despite the above mentioned developments, the euptfv1 (Tóth et al., 2015) and derived soil hydraulic property maps for Europe on a 1km and 250m grid (Tóth et al., 2017) do not include uncertainties in the prediction. Hence, the aim of our study was to update the euptfv1 by deriving a new set of soil hydraulic PTFs (euptfv2) providing uncertainty calculation built into the PTF model. For this, we rely heavily on the datasets used in the construction of the euptfv1. Methodologically, we constructed new soil hydraulic PTFs on the basis of the random forest method which facilitates quantification of prediction-uncertainties. The predicted variables of interest included soil water content at saturation, field capacity and wilting point, plant available water content, saturated hydraulic conductivity, MVG parameters of the moisture retention and hydraulic conductivity curves. The predictions are based on easily available soil properties. The predictor variables were similar to those of euptfv1, except the topsoil and subsoil distinction, which was replaced by mean soil depth of the sample, since it is typically known, anyway. Additionally, the improved performance of the euptfv2 was assessed against predictions using the earlier version. Moreover, we determined the minimum sufficient predictor variables for 32 input variables combinations.

## 2 Materials and Methods

The construction of a pedotransfer function requires three elements: predictor variables, predicted variables as the property of interest, and a transfer method between the former two. The predicted variables are in this case directly measured soil hydraulic properties on samples contained in a large pan-European dataset, ensuring a representativeness of the PTF for Europe. Additionally, Tóth et al. (2015) had fitted MVG model parameters for each sample dataset individually by inverse modelling, which we reused in this study.

### 2.1 Dataset

The European Hydropedological Data Inventory (EU-HYDI) (Weynants et al., 2013) provided the basis for the preparation of the prediction algorithms. The dataset partitions for training and testing the prediction algorithms were almost identical to the ones used in Tóth et al. (2015), except that the samples had to have information on soil depth as well. Depending on the soil hydraulic property of interest, 76-99% of the originally selected samples were used to derive the new PTFs. It enabled comparison of the performance between the EU-PTFs (Tóth et al., 2015) – built in the euptfv1 (Weynants and Tóth, 2014) – and their improved version (euptfv2). Table 1 shows the number of samples in the training and test sets.

### 2.2 Predicted soil hydraulic properties

Prediction algorithms were derived for each of the following soil hydraulic properties:

–   water content at saturation (THS): water content at 0 cm matric potential head;

- water content at field capacity at
    - -100 cm matric potential head (FC_2), and
    - -330 cm matric potential head and (FC);
- water content at wilting point (WP): water content at -15,000 cm matric potential head;
- plant available water content (AWC) based on the following equations:
    - $AWC = FC - WP$      (1)
    - $AWC\_2 = FC\_2 - WP$      (2)
- saturated hydraulic conductivity (KS): hydraulic conductivity at 0 cm matric potential head;
- Mualem-van Genuchten model parameters (VG; for the water retention model only, MVG; for the water retention and hydraulic conductivity model).

Transformation of predicted variables, and explanation on how (i) the water content at a certain matric potential head values were harmonized and (ii) the Mualem-van Genuchten model parameters were fitted is provided in great detail in (Tóth et al., 2015). Similarly to euptfv1, for the description of the moisture retention curve (MRC), we predicted the VG model parameters: the residual water content ($\theta_r$), the saturated water content ($\theta_s$), and shape parameters $\alpha$ and n. For the hydraulic conductivity curve, two additional parameters: the hydraulic conductivity acting as a matching point at saturation $K_0$ and a shape parameter related to pore tortuosity (L) are estimated too. FC_2 was not predicted in euptfv1 and was determined in this study as follows. In the EU-HYDI, 8231 samples have at least one water content observation in the matric potential head range -110 to -95 cm. 86% of those have a measured water retention value exactly at -100 cm matric potential head. In 10% of the cases, FC_2 was set to the water content measured at the closest matric potential head in the range [-110, -95]. In the absence of a measured value at -100 cm, in 4 % of the cases, FC_2 was computed by linear interpolation between the two closest matric potential heads smaller and greater than -100 cm. In the case of AWC and AWC_2 direct and indirect predictions were analysed, i.e. AWC was once predicted directly from the predictor variables and once computed from the PTF predicted variables WP, and FC and FC_2, respectively.

### 2.3 Predictor variables

As predictors we used the following easily available soil properties: the particle size densities (PSD) characterised by the mass-percentages of clay ($<2\ \mu m$), silt ($2$–$50\ \mu m$) and sand ($50$–$2000\ \mu m$), organic carbon content (OC; mass-%) , bulk density (DB; $g\ cm^{-3}$), calcium carbonate content (CACO3; mass-%), pH in water (PH_H2O; -), cation exchange capacity (CEC; cmol (+) $kg^{-1}$), and replaced the former topsoil and subsoil distinction in euptfv1 with mean soil depth (cm) (DEPTH). At minimum, the predictor variables, clay, silt and sand content, as well as mean soil depth were used regardless of predicted variable. In addition to that, we tested every possible combination of the other above mentioned soil properties (predictor variables) to determine which combination significantly improves the performance of the predictions. A total of 32 different combinations of predictor variables were studied in their respective ability to predict the nine different properties of interest; i.e. the set of soil hydraulic properties and model parameters.

Replacing the topsoil/subsoil distinction with depth for the new PTFs was supported by the fact that this information is commonly available, too, or can be based on expert knowledge. Introducing more accurate information on depth might improve the performance without using machine learning algorithms for the prediction. However, we did not test this hypothesis, because our aim was to provide uncertainty of the predictions related to predictor variables of the PTFs. Tested predictor variables are shown in Table 1 with number of samples used to derive the PTFs and compute their performance.

## 2.4 The Random Forest algorithm to derive PTFs

We derived the PTFs adopting the random forest method (Breiman, 2001), implemented in the 'ranger' R package (Wright and Ziegler, 2017). We selected this method, because (i) it is among the best performing prediction algorithms if there is a complex interaction structure in the dataset (Boulesteix et al., 2012), (ii) it computes quantiles of the predicted values, (iii) parallel processing is supported which saves significant computation time, and (iv) the initially black-box type algorithm can be interpreted based on computing variable importance and analysing partial dependence plots implemented in the 'pdp' R package (Greenwell, 2017b).

In the case of a continuous response variable, a random forest is an ensemble of de-correlated regression trees (Breiman, 2001). The regression tree approach divides the predictor space into non-overlapping regions through minimizing the residual sum of squares. The aim of the method is to subset the data as homogeneously as possible at each split. The observations can be assigned to the defined regions in which the mean of the response variable is the predicted value. Single trees of the forest are noisy and limited in performance, but if many unbiased trees are derived and averaged with bagging, the variance is reduced and performance of the prediction improves (Hastie et al., 2009). Building of de-correlated trees is achieved by randomization at two levels. Firstly, each tree of the forest is grown on a randomly selected two thirds of the data with replacement, which is called bootstrap sample or in-bag fraction. Secondly, at each node of a single tree, randomly selected sets of predictors are analysed to split the data. This feature of randomization allows correlation between the response variables (Ziegler and König, 2014), which is an important advantage in the case of pedotransfer functions where predictors are often highly correlated.

Parameter tuning of the ranger was performed with the 'caret' R package (Kuhn et al., 2017, 2018). With the implemented train function, a fivefold cross-validation was repeated ten times to tune the number of randomly selected predictor variables at each split ($mtry$) and find the best performing splitting rule ($splitrule$) during training. We started the tuning by setting the number of randomly selected predictor variables to two, then added one by one until the number of all available predictors for each input variable combination was reached. All three built-in splitting rules in the ranger function were tuned, namely $variance$, $extratrees$ and $maxstat$. The minimum node size was kept to 10. In addition to the tuning options included in the train function of the caret package, we optimized the number of trees in the forest. The above described tuning was performed by discretely altering the number of trees in the forest in separate tuning steps to 50, 100, 200, 500 and 1000, analysing the results and choosing the best number of trees for the random forest.

We analysed the relevance of predictors and their influence on the response variable. The relevance of predictors was determined by computing the variable importance based on the mean decrease in impurity (Hastie et al., 2009) in the ranger

function. The relative importance was assessed by dividing the variable importance of each predictor by the sum of the importance of all the predictors after Kotlar et al. (2019). The marginal effect of some selected predictors on the response – soil hydraulic parameters – was analysed with partial dependence plots (Greenwell, 2017a, 2017b).

The final prediction algorithm was built on the whole training set based on the result of the tuning. To quantify the prediction uncertainties, quantile regression was used (Meinshausen, 2006). In random forest, as implemented in ranger, it is called quantile regression forest. For each node in each tree, the quantile regression forest not only keeps the mean of the predicted target variable, but all observations that belong to that node from which the full conditional distribution of the predicted variable is estimated. The width of the prediction interval varies with the predictor variables. The smaller the range of the prediction interval, the more accurate the prediction is. We analysed the 90% prediction interval for all predictions, but the derived algorithms (PTFs) provide the possibility to compute the individual predictions of each tree.

## 2.5 Evaluation of derived PTFs

The performance of the PTFs was calculated using the median values predicted by the random forests. It was described with the root mean square error (RMSE) (Eq. 3.), and the coefficient of determination ($R^2$) (Eq. 4.) computed for the training and test sets.

$$RMSE = \sqrt{\frac{1}{N}\sum_{i=1}^{N}(y_i - \hat{y}_i)^2} = \sqrt{MSE} \tag{3}$$

$$R^2 = 1 - \frac{\sum_{i=1}^{N}(y_i-\hat{y}_i)^2}{\sum_{i=1}^{N}(y_i-\bar{y})^2} \tag{4}$$

where $y_i$ is the measured and $\hat{y}_i$ the predicted soil water content or log-transformed saturated or unsaturated hydraulic conductivity, $\bar{y}_i$ is the average of $y_i$, N is the number of $y_i$ and $\hat{y}_i$ data pairs, and MSE is the mean square error. The different data range of the dataset influences the performance of the PTFs when that is compared to the studies in the literature. Therefore, normalized RMSE (NRMSE) was computed (Eq. 5.), where $y_{max}$ and $y_{min}$ are the maximum and minimum value of variable.

$$NRMSE = \frac{RMSE}{y_{max}-y_{min}} \tag{5}$$

For each predicted variable, there was an initial set of 32 predictor combinations (Table 1), whose individual performance for each of the predicted variables was assessed. Based on the test results, we derived recommendations which PTF should be used when certain sets of predictor variables are available. We compared the performance of PTFs to quantify if there are significant differences between the predictions as a consequence of adding certain soil properties to the predictor variables. We also compared the performance of point and parameter estimations for those input combinations, which reflect the most frequently available soil property combination from a practical point of view. The aim of this comparison was to analyse whether point or parametric prediction performs better when only THS and/or FC/FC_2 and/or WP are needed.

Additionally, the performance of the presented random forest based PTFs was compared to that of the euptfv1 (Tóth et al., 2015). For comparison, those PTFs from euptfv2 were selected which corresponded to the analysed input variable combination of the euptfv1.

The comparison of PTFs was done using a non-parametric Kruskal-Wallis test at the 5% significance level applied on the MSE values – computed on TEST_BASIC and/or TEST_CHEM+ sets (Table 1) – using the R package agricolae (De Mendiburu, 2017). Recommendation of PTFs for a given set of predictor variables was based on the performance of euptfv2 on the test sets. If there was no significant difference in performance, the PTF derived from the largest population was selected.

All statistical analysis was performed in R [version 3.6.0] (R Core Team, 2019).

## 3 Results and discussion

### 3.1 General performance

In the process of tuning the random forest parameters, the number of trees was found to be sufficient when set to 200 in all cases. The number of candidate predictors was found to be higher than the recommended square root of the number of available predictor variables (p) in most of the cases, especially when p was greater than 5 (Fig.1). When optimizing the splitting rules to build the trees in the forest, overall, the best performance was achieved by the *extratrees* rule in 54 %, by the *variance* rule in 28%, and by the *maxstat* rule in 18% of the cases (Fig. 1).

The RMSE values were between 0.020 and 0.068 $cm^3$ $cm^{-3}$ for THS (Table 2), 0.046 and 0.055 $cm^3$ $cm^{-3}$ for FC (Table 3), 0.040 and 0.060 $cm^3$ $cm^{-3}$ for FC_2 (Table 4), 0.037 and 0.048 $cm^3$ $cm^{-3}$ for WP (Table 5), 0.043 and 0.053 $cm^3$ $cm^{-3}$ for AWC (Table S1), 0.045 and 0.060 $cm^3$ $cm^{-3}$ for AWC_2 (Table S2), and 0.09 and 1.18 $\log_{10}$ (cm day$^{-1}$) for KS (Table 6) in the case of including different predictor variables computed on the test sets. Table S3 shows the NRMSE for the point predictions computed for the TEST_BASIC and TEST_CHEM+ sets to provide possibility for comparison with other PTFs available from the literature. In the case of VG and MVG, RMSE for the entire matric potential head range was between 0.041 and 0.068 $cm^3$ $cm^{-3}$ for the moisture retention (Table 7) and 0.61 and 0.71 $\log_{10}$ (cm day$^{-1}$) for the hydraulic conductivity (Table 8). These RMSE values are within the range of recently published PTFs (McNeill et al., 2018; Nguyen et al., 2017; Román Dobarco et al., 2019; Zhang and Schaap, 2017).

In the case of the point estimations, Figures 2, S1 depict the scatterplots of measured and predicted soil hydraulic parameters with 90% prediction interval computed on the test sets. Performance of the worst to best PTFs are shown. The addition of predictors that significantly improve the predictions also decreases the uncertainty. The largest reduction in the width of the inner 90% of the prediction interval is visible for THS. Specifically this value decreased from 0.21 to 0.10 $cm^3$ $cm^{-3}$ for THS, from 0.19 to 0.14 $cm^3$ $cm^{-3}$ for FC_2, from 0.17 to 0.14 $cm^3$ $cm^{-3}$ for FC, from 0.15 to 0.14 $cm^3$ $cm^{-3}$ for WP, from 0.19 to 0.17 $cm^3$ $cm^{-3}$ for AWC_2, from 4.1 to 3.2 $\log_{10}$ (cm day$^{-1}$) for KS. In the case of AWC the mean 90 % mean interval did not change (0.15 $cm^3$ $cm^{-3}$).

Figures S2, S4, S6, S8, S10, S12, S14, S16, S19 show the squared error of the derived PTFs computed on the TEST_BASIC and TEST_CHEM+ sets. The PTFs are ordered based on their performance. Density plots of measured and predicted soil hydraulic values are included in Figures S3, S5, S7, S9, S11, S13, S15, S17, S20. Plots show the PTFs that use the most frequently available predictors.

This study strengthens the importance of chemical soil properties in the prediction. CEC was found to be an important predictor by Pachepsky and Rawls (1999) for FC and WP, by Botula et al. (2013) for water retention at several matric potential head values, and by Hodnett and Tomasella (2002) for the VG parameters. Hodnett and Tomasella (2002) showed that pH influenced all four VG parameters. The role of CACO3 was shown to be not significant in the study of Khodaverdiloo et al. (2011). They highlight that a possible influence of CACO3 might already have been indirectly included by bulk density. The role of PSD,

BD and OC has been studied extensively by various authors, e.g. Nemes et al. (2003); Rawls et al. (2003); Vereecken et al. (1989); Weynants et al. (2009); Wösten et al. (1999), which is in line with the general pattern of variable influence we see in this study.

Table S3 summarizes the recommended PTF for each combination of available predictor variables. The importance and influence of soil properties on the performance of hydraulic PTFs and results of partial dependence plots are reported below

by predicted soil hydraulic properties.

### 3.2 Point estimations

The performance of the PTFs was computed for the training and test sets (Tables 2-8 and Tables S1-2) indicating the presence of significant differences. For each predictor variable, the recommended PTF number is indicated and its predictor variables are highlighted in bold font in the respective tables. For easier comparison with euptfv1, the corresponding PTF number used

in Tóth et al. (2015) is additionally provided in each table. In the following, detailed results of the constructed PTFs for the individual predicted variables are presented and discussed.

### Water content at saturation

Table 2, Figures S2 and S3 show the performance of the PTFs predicting THS. The best performing random forest is PTF03. It is also the one trained on the largest population. It uses PSD, DEPTH and BD as predictors. For the prediction of THS, the

most important variable by far is BD (Fig. 3). When BD is not used for the computation of THS, values above 0.60 $cm^3$ $cm^{-3}$ are not well predicted (Fig. S3). The addition of OC or CACO3 or PH_H2O to PSD and DEPTH improves significantly the performance of the PTF. The picture changes if BD is known: if PSD, DEPTH and BD were available, further addition of OC or CACO3 or PH_H2O or CEC does not significantly improve the prediction, neither do their combinations. Figure 4 shows the dependence of THS on OC and BD, considering the average effect of the other predictor variables – i.e. PSD and DEPTH.

When BD is lower than 1.5 g $cm^{-3}$ changes in OC do not influence THS. If BD is larger than 1.5 g $cm^{-3}$, samples with higher OC have higher THS.

**Water content at field capacity**

The performance of the PTFs computed on training and test set are shown in Table 3, Figures S4 and S5 for FC_2 and in Table 4, Figures S6 and S7 for FC. The best performing PTF derived from the largest population is the one using i) PSD, DEPTH, OC, BD and PH_H2O (PTF18) in the case of FC_2, and ii) PSD, DEPTH, OC and BD (PTF07) for FC.

For FC_2, the two most important variables are USSAND and BD (Fig. 3). When BD and USSAND increase, FC_2 decreases (Fig. 4). Adding OC or BD to PSD and DEPTH significantly improves the prediction of FC_2. If either of CACO3, PH_H2O or CEC is added as a further predictor to PSD and DEPTH, the performance of the PTF does not significantly improve. If PSD, DEPTH and BD are available, adding OC or CACO3 or PH_H2O does not significantly improve the prediction. Including CEC as an additional predictor besides PSD, DEPTH and BD, significantly improves the estimation of FC_2.

USSAND and USCLAY are the two most important variables for the prediction of FC (Fig. 3). Instead of analysing these two soil properties, both characterizing the soil texture, we include OC next to USSAND in the partial dependence plot analysis, because the amount of OC can be altered due to change in climate, land use, soil and water management, cropping systems, etc. (Wiesmeier et al., 2019). Within the range of OC in the dataset FC increases with increasing OC regardless of USSAND content by up to 0.08 $cm^3 cm^{-3}$ even when USSAND is greater than 60 % (Fig. 4). Adding OC or CEC to PSD and DEPTH

significantly improves the prediction of FC. The effect of CEC on the prediction of FC was also shown by Pachepsky and Rawls (1999). BD or CaCO3 or PH_H2O do not significantly improve the predictions if PSD, DEPTH, or PSD, DEPTH and OC are available. Predictions significantly improve when both CaCO3 and PH_H2O are added as predictors to PSD, DEPTH and OC.

**Water content at wilting point**

The performance of PTFs derived for WP prediction is shown in Table 5, Figures S8 and S9. Among the best performing PTFs, PTF09 is derived on the largest training set. It uses PSD, DEPTH, OC and PH_H2O as predictors. Even though the most important variables for WP prediction were USCLAY and USSAND (Fig. 3), we included OC on the partial dependence plot (Fig. 4) as in the FC analysis. USCLAY had the strongest influence on WP. The influence of OC on WP can be detected for soils with OC less than 4 % and USCLAY less than 50 %. Below 10 % USCLAY, the WP slightly increases with increasing

OC. When USCLAY is between 10 and 50 % and OC is less than 4%, increasing OC generally decreases WP.

OC significantly improves the prediction of WP if added to PSD and DEPTH. If BD or CACO3 or PH_H2O or CEC are added to PSD and DEPTH, the performance of the prediction does not improve significantly. Adding CACO3 and CEC to PSD, DEPTH and OC significantly improves the prediction.

**Plant available water content**

Table S1, S2 and Figures S1, S10-13 show the performance of AWC and AWC_2 predictions. PTF03 is the best performing algorithm with largest training set for both. It considers PSD, DEPTH and BD for the prediction. For both AWC and AWC_2,

BD is the most important predictor among the analysed variables (Fig. 3). The second most important variable is USCLAY in the case of AWC_2 and USSILT for AWC. Increasing BD and USCLAY decreases AWC_2. In the case of AWC, increasing BD and decreasing USSILT decreases the water content (Fig. 4).

OC and BD significantly improve the prediction of AWC_2 when added as input variables next to PSD and DEPTH. If either BD or OC is already included, adding the respective other, does not significantly improve the prediction. Neither PH_H2O, CACO3 nor CEC improves the prediction.

For the prediction of AWC, further addition of only BD or OC or CACO3 or PH_H2O or CEC to PSD and DEPTH does not significantly improve the prediction. If both OC and BD are included as predictors next to PSD and DEPTH, the prediction significantly improves.

There is no significant difference between direct and indirect predictions, neither for AWC nor for AWC_2. However, the size of the test set used for the statistical analysis is limited. There were only 145 samples in the TEST_BASIC set and 64 samples in TEST_CHEM+ set after merging datasets available for both direct and indirect predictions for analysing AWC, and 70 and 34 samples in the case of AWC_2. Thus, if prediction of FC_2/FC and WP are needed in addition to AWC_2/AWC, we recommend computing AWC from those to save on computing time. Variation in AWC could be explained less efficiently (Table S1, S2) than the other studied water retention values but the performance of the prediction is comparable with that of published in the literature (Li et al., 2016; Malone et al., 2009).

**Saturated hydraulic conductivity**

The performance of KS prediction is shown in Table 6, Figure S14 and S15. The predictors of the best performing PTF derived on the largest training set are PSD, DEPTH and OC (PTF02). The prediction of KS significantly improves if OC is included among the predictor variables next to PSD and DEPTH. No other predictors significantly improve the performance of the PTF. On the training dataset, when OC is greater than 2.5 %, the influence of clay content on KS is more dominant than that of OC (Fig. 4). In the case of KS prediction, the simplest best performing PTF – which was derived on a training dataset with KS ranging between -3.00 and 4.67 $\log_{10}(\text{cm day}^{-1})$ – has an RMSE of 0.94 $\log_{10}(\text{cm day}^{-1})$ and NRMSE 0.14 $\log_{10}(\text{cm day}^{-1})$ (Table S3). PSD and CEC are the most important input variables for the prediction of KS when all nine variables are considered as predictors (Fig. 3). In that case, OC is the fifth and BD is only the eighth most important variable. The prediction performance is influenced by the heterogeneity of measurement methods of KS in the EU-HYDI dataset. When the methods are homogeneous, the RMSE value is usually around 0.6-0.8 $\log_{10}(\text{cm day}^{-1})$ as reviewed by (Zhang and Schaap, 2019). ROSETTA3 PTF with PSD and BD predictors had an RMSE of 0.68 $\log_{10}(\text{cm day}^{-1})$ with an NRMSE of 0.11 $\log_{10}(\text{cm day}^{-1})$ (Zhang and Schaap, 2017). Araya and Ghezzehei (2019) published PTF using PSD, BD and OC predictors with the highest accuracy in the literature with an RMSE of 0.34 $\log_{10}(\text{cm day}^{-1})$ and NRMSE of 0.06 $\log_{10}(\text{cm day}^{-1})$. In Lilly et al. (2008), the performance of the KS predictions and findings were similar to this study. They report an RMSE between 0.95 and 1.08 $\log_{10}(\text{cm day}^{-1})$ – with an NRMSE between 0.17 and 0.20 $\log_{10}(\text{cm day}^{-1})$ – for the KS prediction when analysed with several input combinations. Even when information on soil structure and crack orientation was considered – next to topsoil and subsoil

distinction, PSD, BD and OC – the RMSE was 0.97 $\log_{10}(\text{cm day}^{-1})$. BD would be among the most important variables, but also in their analysis its influence was masked out. They derived the PTFs on the HYPRES dataset (Wösten et al., 1999), which also includes very diverse methods to determine the saturated hydraulic conductivity and part of which is also contained in the EU-HYDI. The uncertainty in the predictions (Fig. 2) could be decreased if the predictions would be differentiated according to the measurement methods, but that might decrease the applicability of the PTFs. On the contrary, this study indicates the necessity to include saturated hydraulic conductivity values determined from many different measurement techniques, otherwise, the PTFs are expected to lose their generality.

### 3.3 Parameter estimations

The performance of parametric PTFs are shown in Tables 7 and 8 and Figures 5, 6, S16-S21. Figure 7 illustrates the importance of variables for the prediction of VG and MVG parameters. The best performing PTF derived on the largest training set is PTF29 – with PSD, DEPTH, OC, BD, PH_H2O and CEC – for MRC and PTF27 – with PSD, DEPTH, OC, BD, CACO3, PH_H2O – for HCC.

For $\theta_r$, overall, BD is the most important predictor while all other predictors show similar variable importance (Fig. 7). Interpretation of this parameter is complex, but it was demonstrated that it is influenced by the soil specific surface area (Assouline and Or, 2013), and the measured data range (Weber et al. 2019). For $\theta_s$, the most important predictor is by far BD, similarly to THS. The importance of CEC has to be noted for the prediction of parameters α, n and L. For the prediction of parameter n – which relates to the pore size distribution – USCLAY and USSAND are the most important variables. $K_0$ is influenced by several soil properties besides those included in the dataset used here, e.g. pore connectivity, tortuosity, primary pore orientation. These properties cannot be directly inferred from other soil properties limiting the explanatory power of the available properties. The prediction of $K_0$ remains complex and challenging. Variable importance of all studied predictors is greater than 70%. Moreover, $K_0$ is influenced by the data quality, and is correlated in parameter space, which is not treated here.

Only a few studies have analysed the importance of CEC for MRC and HCC PTFs (Botula et al., 2013; Hodnett and Tomasella, 2002; Pachepsky and Rawls, 1999) which might be linked to the fact that CEC is rarely available in soil hydraulic datasets. It is noteworthy to highlight that all best performing MRC PTFs (PTF24, PTF28, PTF29, PTF30, PTF31) include CEC among the predictors (Table 7). In addition to that, Hodnett and Tomasella (2002) found that CEC was important for the prediction of $\theta_r$ and α parameters of the van Genuchten model. This is because CEC provides indirect information on soil mineralogy and reflects soil specific surface area, charge density and pore size which influence soil water retention (Lal and Shukla, 2004).

### Moisture retention curve

If BD or OC or CACO3 or CEC or PH_H2O are added as a predictor to information on PSD and DEPTH, the performance of the PTF significantly improves (Table 7., Fig. S16). Adding BD next to PSD and DEPTH improves the predictions more than adding OC (Table 7., Fig. S17). BD and OC together significantly improve the prediction compared to using PSD, DEPTH

together with either BD or OC. Adding OC next to PSD, DEPTH, BD and chemical soil properties (CACO3 and/or CEC and /or PH_H2O) does not significantly improve the prediction. If PSD, DEPTH, CACO3 and CEC are available, further addition of PH_H2O does not improve the prediction. The best performing PTF includes USSAND, USSILT, USCLAY, DEPTH, BD, CACO3, CEC. Figure 5 shows a scatterplot of measured and predicted water content values, including the performance of the worst and the best performing PTF (PTF01 and PTF29). The importance of including chemical properties and most importantly bulk density among the predictors is visible when measured water contents are greater than 0.50 $cm^3$ $cm^{-3}$. Those high water content values are characteristic when the soil is close to saturation, thus indirect information about the structure is needed for more accurate predictions of those water content values. Parametric PTFs underestimate water content near saturation and between -200 and -15,000 cm matric potential head (Fig. S18). Overestimation occurs between -10 and -50 cm matric potential head and above 16000 cm matric potential head. When chemical soil properties are included, the degree of underestimation decreases between -200 and -15,000 cm matric potential head, but overestimation increases between -5 and -10 cm with around 0.02 $cm^3$ $cm^{-3}$.

**Hydraulic conductivity curve**

OC, CACO3, PH_H2O and CEC significantly improves the prediction of HCC when added to PSD and DEPTH. Adding BD next to PSD and DEPTH does not improve the predictions (Table 8, Fig. S19, S20). If PSD, DEPTH and OC are used as predictors, further addition of BD or CACO3 or PH_H2O or CEC does not significantly improve the performance of the PTFs. However, adding CaCO3 and CEC or PH_H2O significantly improve the prediction. The performance of the worst and the best performing PTF is shown on Figure 6. The PTF with only PSD and DEPTH underestimate hydraulic conductivity values smaller than 0.01 cm $day^{-1}$. When OC, BD, PH_H2O and CEC are included, the underestimation decreases. This could be explained by the fact that these predictors contain indirect information on soil particle surface area and surface characteristics, which are some of the governing properties of low hydraulic conductivities.

When soil chemical properties are not used as predictors, hydraulic conductivity is underestimated close to saturation and at matric potential heads smaller than -500 cm; overestimation occurs between -10 and -500 cm matric potential head (Fig. S21). If chemical properties are also considered, hydraulic conductivity is i) underestimated at matric potential head smaller than -5000 cm, and ii) overestimated between -5 and -5000 cm. With added information on chemical properties, the degree of underprediction decreases close to saturation and at the very dry end of the hydraulic conductivity curve. An increase in prediction performance for values lower than 0.1 cm $day^{-1}$ is visible also on Figure 6.

Samples with measurements of the HCC at pressure heads < -1000 cm are less frequent and are not as numerous within a dataset of a single sample, if it was measured. Since the dataset of estimated VG model parameters was identical in this study and in Tóth et al. (2015), differences between the two studies of the unsaturated HCC are related to the PTF methods involved. However, at pressure heads <-1000 cm, the HCC is dominated by non-capillary conductivity (Streck and Weber, 2020; Weber et al., 2019), which is not included in the MVG model. The considerable data mismatch observable for the dry range (Fig. 6) can only be overcome by a different soil hydraulic property model and by a different PTF, because of compensatory effects in

the VG. With this, we mean that better data descriptions in the dry end will lead to a larger mismatch in the wet end, as a consequence of the rigid model structure in the MVG model, which only accounts for capillary storage and conductivity. For better data description at <-1000 cm other more comprehensive models need to be adopted (Weber et al., 2020a).

## 3.4 Comparison of point and parameter predictions

We compared the performance of the best point prediction methods (Table 2-5) with the best parameter estimations (Table 7) on the test sets. In 5 out of 20 cases, point predictions are significantly more accurate and for further 8 cases, RMSE was smaller. In all other cases, we have no significant difference between point and parametric PTFs (Table 9). The reason for higher RMSE in parameter estimation can be that the VG model does not always adequately describe the measured MRC data (Weber et al., 2019). Therefore, when THS, FC, FC_2 and WP are computed with parameter estimation those are not only

affected by the uncertainty of the prediction of VG parameters but by the goodness of VG model fit as well. We found similar results in the case of euptfv1 (Tóth et al., 2015). Tomasella et al. (2003) and Børgesen and Schaap (2005) had comparable findings regarding the performance of point and parametric PTFs. For THS point estimation performed better than parameter estimation. When the moisture retention curve is not needed, but only THS and/or FC/FC_2 and/or WP, we recommend computing those with the point PTFs, more detailed explanation on it is included in Tóth et al. (2015).

## 3.5 Comparison of euptfv1 and v2

In 14 out of 19 cases, the PTFs of euptfv2 perform significantly better predicting the test sets than the PTFs of euptfv1. In the remaining 5 cases, there is no significant difference (Table 10). Predictions of FC and MRC improve in all cases. The most important reason for it can be that the interaction between the target variable and the predictors is more complex for the cases of predicting FC or VG parameters – to describe the MRC –, which can be untangled using random forest. This may provide

a reason the random forest algorithm performed significantly better than the PTFs derived with linear regression or a simple regression tree. For THS, WP, KS, and MVG only those PTFs did not improve significantly, for which comparisons on the TEST_CHEM+ set were possible – which includes a reduced number of samples. The RMSE of THS prediction was somewhat lower for euptfv1 than for euptfv2, but the difference was not significant. It could be due to the close to linear relationship between THS and BD and high relative importance of BD in THS prediction (84 %). This way their interaction can be

efficiently described with the linear regression which is capable to extrapolate as well. Extrapolation with the random forest algorithm is not possible outside the training data, which can limit its performance. The general improvement of the PTFs in euptfv2 is threefold, the better performance  is due to i) using random forest instead of single regression tree or linear regression, ii) including more detailed information on soil sampling depth, not only distinguishing topsoils and subsoils and iii) providing information on prediction uncertainty.

We recommend the use of euptfv2 instead of euptfv1 if continuous soil properties are available. If only texture classes – i.e. no particle size distribution – are available, class PTFs of euptfv1 can be used, that is PTF18 for modified FAO texture classes and PTF19 for USDA texture classes.

**4 Practical guidance on how to use the PTFs**

The minimum input requirements for all PTFs are sand, silt and clay content, and soil depth. Soil depth is defined as the mean sampling depth, if e.g. PSD, BD and OC are provided for a soil sample from a depth of 0-20 cm, then the soil depth input (DEPTH) to the prediction algorithm is set to 10 cm.

If only soil texture information is available for the predictions, the class PTFs from euptfv1 could be applied (Tóth et al., 2015). We emphasise that:

1.  the units of input soil properties (predictors) have to be the same as indicated in the text and that the sand, silt, and clay are defined by the following particle diameters: clay < 2 μm, silt between 2 and 50 μm, and sand between 50 and 2000 μm,

2.  when only specific water content values at saturation, field capacity or wilting point are required (ie. THS, FC_2, FC, WP) it is recommended to use point PTFs. This is also true for the prediction of KS,

3.  for AWC, the most accurate way is by first predicting FC and WP with the point predictions and then compute AWC using Eq. (1), and similarly for AWC_2 using FC_2 and Eq. (2),

4.  it is recommended to do the VG prediction if only moisture retention curve parameters are needed, and

5.  the MVG prediction when both moisture retention and hydraulic conductivity parameters are required.

The VG algorithms predict the following van Genuchten model parameters: the residual water content $\theta_r$ (cm³ cm⁻³), the saturated water content $\theta_s$ (cm³ cm⁻³), and shape parameters $\alpha$ (cm⁻¹) and $n$ (-). Parameter m is provided based on $m=1-1/n$ (van Genuchten, 1980), and for the hydraulic conductivity curve, the two additional parameters: $K_0$ (cm day⁻¹) the hydraulic conductivity acting as a matching point at saturation and L (-), a shape parameter related to pore tortuosity.

Table 11 shows the recommended PTFs for each predicted soil hydraulic property and available predictor variables. The users need to check which basic soil properties are available for the predictions, then look in Table 11 which PTF is recommended to use.

The algorithms have been implemented in a web interface to facilitate the use of the PTFs, where the PTFs' selection is automated based on soil properties available for the predictions and required soil hydraulic property. The Code and data

availability section provides information on how to access this resource.

**5 Conclusions**

The updated EU-PTFs – euptfv2 – perform significantly better than euptfv1 and are applicable for 32 predictor variables combinations. Uncertainties of the predicted soil hydraulic properties and model parameters can be computed. These uncertainties are, without further discrimination, related to the considered input data, predictors and the applied algorithm. The

euptfv2 includes transfer functions to compute soil water content at saturation (0 cm matric potential head), field capacity (both -100 and -330 cm matric potential head) and wilting point (-15,000 cm matric potential head), plant available water content computed with field capacity at -100 and -330 cm matric potential head, saturated hydraulic conductivity, and Mualem-

van Genuchten parameters of the moisture retention and hydraulic conductivity curves. For analyses of the impact as well as the significance of the uncertainties on the predicted soil hydraulic properties and model parameters, further studies are required.

**Code and data availability.** The current version of euptfv2 is available from a user friendly web
interface: https://ptfinterface.rissac.hu (Szabó et al., 2019b) under the Creative Commons Attribution-NonCommercial 3.0 Unported License. The exact version of the model used to produce the results used in this paper is archived on Zenodo (https://doi.org/10.5281/zenodo.3759443, Szabó et al., 2020), as are the R scripts to develop the predictions and the derived pedotransfer functions – in RData format – presented in this paper. The training data set cannot be made publicly available due to legal restrictions of the EU-HYDI dataset, thus only a test sample is provided along with the model code. The R files
are complied into an R package to use the pedotransfer functions, archived on Zenodo (https://doi.org/10.5281/zenodo.4281046, Weber et al., 2020b).

**Supplement.** The Supplement related to this article is available online.

**Author contribution.** BSZ, TKDW and MW conceptualized the study and designed the methodology. BSZ supervised the research. MW cured the EU-HYDI dataset. BSZ, TKDW and MW prepared scripts for the statistical analysis, BSZ carried out
the formal analysis, visualization and coordinated building of the PTF web interface. TKDW and BSZ built the R package with contributions of MW. BSZ and TKDW performed the validation. BSZ and TKDW wrote the paper with considerable input from MW.

**Competing interests.** The authors declare that they have no conflict of interest.

**Acknowledgements.** B. Szabó is supported by the Hungarian National Research, Development and Innovation Office (NRDI)
under grant KH124765 and the János Bolyai Research Scholarship of the Hungarian Academy of Sciences. On behalf of János Bolyai Research Scholarship we thank for the usage of MTA Cloud (https://cloud.mta.hu/) that significantly helped us achieving the results published in this paper. We praise the individual scientists and their institutions in Europe who contributed in the establishment of the EU-HYDI database. We would like to thank Gergely Tóth and Luca Montanarella for their effort in coordinating the formation of the EU-HYDI database. T. K.D Weber's contribution was supported by the Collaborative
Research Center 1253 CAMPOS (Project 7: Stochastic Modelling Framework), funded by the German Research Foundation

(DFG, Grant Agreement SFB 1253/1 2017). We thank to Jeromos Rózsa for defining the computing infrastructure. We also thank to Dávid Gyurkó for building the web interface for the prediction algorithms.

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

**TABLES**

**Table 1.** Number of samples by predictor variable combinations used to derive the new European PTFs (euptfv2). Rows in italic font indicate PTFs with the same predictor variables as were tested in euptfv1 (Tóth et al., 2015).

| Name | Predictor variables[1] | Number of samples in TRAIN set[2] | | | | | | | | |
|------|------------------------|------|------|------|------|------|-------|------|------|------|
| | | THS | FC_2 | FC | WP | KS | AWC_2 | AWC | VG | MVG |
| *PTF01* | *PSD+DEPTH* | *3354* | *5109* | *2196* | *5264* | *3157* | *3528* | *1863* | *4669* | *739* |
| *PTF02* | *PSD+DEPTH+OC* | *2966* | *4131* | *1716* | *4802* | *2620* | *3208* | *1650* | *3708* | *407* |
| PTF03 | PSD+DEPTH+BD | 3305 | 5034 | 2176 | 5197 | 3146 | 3472 | 1849 | 4593 | 726 |
| PTF04 | PSD+DEPTH+CACO3 | 678 | 1670 | 1537 | 1816 | 639 | 1548 | 1531 | 1671 | 273 |
| PTF05 | PSD+DEPTH+PH_H2O | 1203 | 2062 | 1278 | 2039 | 907 | 1849 | 1245 | 1897 | 230 |
| PTF06 | PSD+DEPTH+CEC | 895 | 1649 | 1097 | 1703 | 567 | 1550 | 1092 | 1488 | 141 |
| *PTF07* | *PSD+DEPTH+OC+BD* | *2959* | *4117* | *1711* | *4786* | *2609* | *3197* | *1645* | *3695* | *404* |
| PTF08 | PSD+DEPTH+OC+CACO3 | 673 | 1586 | 1340 | 1599 | 613 | 1464 | 1336 | 1589 | 250 |
| PTF09 | PSD+DEPTH+OC+PH_H2O | 1052 | 1808 | 1100 | 1678 | 862 | 1615 | 1074 | 1663 | 224 |
| PTF10 | PSD+DEPTH+OC+CEC | 744 | 1437 | 1001 | 1459 | 525 | 1358 | 998 | 1293 | 138 |
| PTF11 | PSD+DEPTH+BD+CACO3 | 678 | 1666 | 1526 | 1806 | 639 | 1545 | 1522 | 1670 | 272 |
| PTF12 | PSD+DEPTH+BD+PH_H2O | 1156 | 2008 | 1267 | 1979 | 898 | 1796 | 1236 | 1847 | 229 |
| PTF13 | PSD+DEPTH+BD+CEC | 848 | 1596 | 1093 | 1648 | 558 | 1498 | 1088 | 1437 | 140 |
| PTF14 | PSD+DEPTH+CACO3+PH_H2O | 678 | 1314 | 1235 | 1375 | 620 | 1195 | 1230 | 1264 | 223 |
| PTF15 | PSD+DEPTH+CACO3+CEC | 373 | 770 | 793 | 831 | 405 | 726 | 791 | 758 | 136 |
| PTF16 | PSD+DEPTH+PH_H2O+CEC | 894 | 1350 | 744 | 1349 | 567 | 1255 | 739 | 1188 | 141 |
| PTF17 | PSD+DEPTH+OC+BD+CACO3 | 673 | 1585 | 1338 | 1596 | 613 | 1464 | 1334 | 1588 | 249 |
| *PTF18* | *PSD+DEPTH+OC+BD+PH_H2O* | *1047* | *1799* | *1098* | *1667* | *853* | *1607* | *1072* | *1655* | *223* |
| PTF19 | PSD+DEPTH+OC+BD+CEC | 739 | 1427 | 998 | 1447 | 516 | 1349 | 995 | 1284 | 137 |
| PTF20 | PSD+DEPTH+OC+CACO3+PH_H2O | 673 | 1249 | 1062 | 1183 | 613 | 1130 | 1059 | 1201 | 219 |
| PTF21 | PSD+DEPTH+OC+CACO3+CEC | 369 | 727 | 709 | 743 | 401 | 683 | 707 | 712 | 135 |
| PTF22 | PSD+DEPTH+OC+PH_H2O+CEC | 744 | 1142 | 663 | 1121 | 525 | 1067 | 660 | 996 | 138 |
| PTF23 | PSD+DEPTH+BD+CACO3+PH_H2O | 678 | 1310 | 1224 | 1365 | 620 | 1192 | 1221 | 1263 | 222 |
| PTF24 | PSD+DEPTH+BD+CACO3+CEC | 373 | 768 | 790 | 827 | 405 | 725 | 788 | 757 | 135 |
| PTF25 | PSD+DEPTH+BD+PH_H2O+CEC | 847 | 1298 | 741 | 1295 | 558 | 1204 | 736 | 1138 | 140 |
| PTF26 | PSD+DEPTH+CACO3+PH_H2O+CEC | 373 | 727 | 734 | 772 | 405 | 684 | 732 | 717 | 136 |
| PTF27 | PSD+DEPTH+OC+BD+CACO3+PH_H2O | 673 | 1248 | 1060 | 1180 | 613 | 1130 | 1057 | 1200 | 218 |
| PTF28 | PSD+DEPTH+OC+BD+CACO3+CEC | 369 | 726 | 707 | 740 | 401 | 683 | 705 | 711 | 134 |
| PTF29 | PSD+DEPTH+OC+BD+PH_H2O+CEC | 739 | 1133 | 661 | 1110 | 516 | 1059 | 658 | 988 | 137 |
| *PTF30* | *PSD+DEPTH+OC+CACO3+PH_H2O+CEC* | *369* | *684* | *655* | *689* | *401* | *641* | *653* | *671* | *135* |
| PTF31 | PSD+DEPTH+BD+CACO3+PH_H2O+CEC | 373 | 725 | 731 | 768 | 405 | 683 | 729 | 716 | 135 |
| *PTF32* | *PSD+DEPTH+OC+BD+CACO3+PH_H2O+CEC* | *369* | *683* | *653* | *686* | *401* | *641* | *651* | *670* | *134* |
| **Number of samples in TEST_BASIC set** | | 1247 | 1762 | 801 | 2088 | 1117 | 1372 | 705 | 1591 | 176 |
| **Number of samples in TEST_CHEM+ set** | | 156 | 296 | 280 | 294 | 169 | 274 | 279 | 288 | 57 |

[1]PSD: particle size distribution (sand, 50–2000 μm; silt, 2–50 μm; clay, <2 μm (mass %)); DEPTH: mean soil depth (cm); OC: organic carbon content (mass %); BD: bulk density (g cm$^{-3}$); CACO3: calcium carbonate content (mass %); PH_H2O: pH in water (-); CEC: cation exchange capacity (cmol (+) kg$^{-1}$).
[2]THS: saturated water content (pF 0); FC_2: water content at -100 cm matric potential head (pF 2.0); FC: water content at -330 cm matric potential head (pF 2.5); AWC_2: plant available water content based on FC_2; AWC: plant available water content based on FC; WP: water content at wilting point (pF 4.2); KS: saturated hydraulic conductivity; VG: parameters of the van Genuchten model; MVG: parameters of the Mualem – van Genuchten model; TEST_BASIC: samples with measured PSD, DEPTH, OC and BD; TEST_CHEM+: samples with measured PSD, DEPTH, OC, BD, CACO3, PH_H2O and CEC.

**Table 2.** Performance of pedotransfer functions (PTFs) by input combination on training and test datasets to predict water content at saturation (THS). N: number of samples, RMSE: root mean square error ($cm^3$ $cm^{-3}$), and $R^2$: determination coefficient, TEST_BASIC: samples with measured PSD, DEPTH, OC and BD; TEST_CHEM+: samples with measured PSD, DEPTH, OC, BD, CACO3, PH_H2O and CEC. Recommended PTFs are highlighted in bold.

| Name of PTF in euptfv2 | Predictor variables[1] | Training set N | RMSE | R² | Test set N | RMSE | R² | Sign. difference[2] TEST_BASIC set | TEST_CHEM+ set | Recommended PTF | Pair from euptfv1 |
|---|---|---|---|---|---|---|---|---|---|---|---|
| **PTF01** | **PSD+DEPTH** | 3354 | 0.067 | 0.366 | 1274 | 0.068 | 0.344 | a | a | PTF01 | - |
| **PTF02** | **PSD+DEPTH+OC** | 2966 | 0.053 | 0.577 | 1274 | 0.056 | 0.552 | b | abc | PTF02 | PTF04 |
| **PTF03** | **PSD+DEPTH+BD** | 3305 | 0.029 | 0.880 | 1274 | 0.031 | 0.862 | c | d | PTF03 | - |
| **PTF04** | **PSD+DEPTH+CACO3** | 678 | 0.046 | 0.187 | 156 | 0.057 | 0.053 | - | bc | PTF04 | - |
| **PTF05** | **PSD+DEPTH+PH_H2O** | 1203 | 0.056 | 0.298 | 156 | 0.053 | 0.193 | - | bc | PTF05 | - |
| PTF06 | PSD+DEPTH+CEC | 895 | 0.055 | 0.401 | 156 | 0.057 | 0.048 | - | ab | PTF01 | - |
| PTF07 | PSD+DEPTH+OC+BD | 2959 | 0.027 | 0.888 | 1274 | 0.030 | 0.869 | c | d | PTF03 | PTF05 |
| PTF08 | PSD+DEPTH+OC+CACO3 | 673 | 0.044 | 0.209 | 156 | 0.055 | 0.118 | - | bc | PTF02 | - |
| PTF09 | PSD+DEPTH+OC+PH_H2O | 1052 | 0.046 | 0.457 | 156 | 0.050 | 0.272 | - | c | PTF02 | - |
| PTF10 | PSD+DEPTH+OC+CEC | 744 | 0.046 | 0.519 | 156 | 0.051 | 0.233 | - | abc | PTF02 | - |
| PTF11 | PSD+DEPTH+BD+CACO3 | 678 | 0.023 | 0.791 | 156 | 0.022 | 0.863 | - | d | PTF03 | - |
| PTF12 | PSD+DEPTH+BD+PH_H2O | 1156 | 0.027 | 0.826 | 156 | 0.021 | 0.878 | - | d | PTF03 | - |
| PTF13 | PSD+DEPTH+BD+CEC | 848 | 0.027 | 0.848 | 156 | 0.021 | 0.873 | - | d | PTF03 | - |
| PTF14 | PSD+DEPTH+CACO3+PH_H2O | 678 | 0.045 | 0.231 | 156 | 0.050 | 0.265 | - | bc | PTF05 | - |
| PTF15 | PSD+DEPTH+CACO3+CEC | 373 | 0.045 | 0.257 | 156 | 0.054 | 0.164 | - | abc | PTF04 | - |
| PTF16 | PSD+DEPTH+PH_H2O+CEC | 894 | 0.052 | 0.459 | 156 | 0.055 | 0.132 | - | bc | PTF05 | - |
| PTF17 | PSD+DEPTH+OC+BD+CACO3 | 673 | 0.019 | 0.856 | 156 | 0.021 | 0.872 | - | d | PTF03 | - |
| PTF18 | PSD+DEPTH+OC+BD+PH_H2O | 1047 | 0.024 | 0.848 | 156 | 0.021 | 0.871 | - | d | PTF03 | PTF06 |
| PTF19 | PSD+DEPTH+OC+BD+CEC | 739 | 0.027 | 0.837 | 156 | 0.021 | 0.874 | - | d | PTF03 | - |
| PTF20 | PSD+DEPTH+OC+CACO3+PH_H2O | 673 | 0.043 | 0.251 | 156 | 0.050 | 0.285 | - | c | PTF02 | - |
| PTF21 | PSD+DEPTH+OC+CACO3+CEC | 369 | 0.043 | 0.309 | 156 | 0.051 | 0.242 | - | bc | PTF02 | - |
| PTF22 | PSD+DEPTH+OC+PH_H2O+CEC | 744 | 0.046 | 0.531 | 156 | 0.050 | 0.280 | - | bc | PTF02 | - |
| PTF23 | PSD+DEPTH+BD+CACO3+PH_H2O | 678 | 0.023 | 0.796 | 156 | 0.021 | 0.869 | - | d | PTF03 | - |
| PTF24 | PSD+DEPTH+BD+CACO3+CEC | 373 | 0.021 | 0.841 | 156 | 0.021 | 0.869 | - | d | PTF03 | - |
| PTF25 | PSD+DEPTH+BD+PH_H2O+CEC | 847 | 0.027 | 0.850 | 156 | 0.020 | 0.883 | - | d | PTF03 | - |
| PTF26 | PSD+DEPTH+CACO3+PH_H2O+CEC | 373 | 0.044 | 0.305 | 156 | 0.049 | 0.308 | - | abc | PTF05 | - |
| PTF27 | PSD+DEPTH+OC+BD+CACO3+ PH_H2O | 673 | 0.019 | 0.858 | 156 | 0.022 | 0.865 | - | d | PTF03 | - |
| PTF28 | PSD+DEPTH+OC+BD+CACO3+CEC | 369 | 0.021 | 0.845 | 156 | 0.021 | 0.874 | - | d | PTF03 | - |
| PTF29 | PSD+DEPTH+OC+BD+PH_H2O+CEC | 739 | 0.026 | 0.843 | 156 | 0.020 | 0.880 | - | d | PTF03 | - |
| PTF30 | PSD+DEPTH+OC+CACO3+PH_H2O+ CEC | 369 | 0.042 | 0.356 | 156 | 0.049 | 0.319 | - | bc | PTF02 | PTF04 |
| PTF31 | PSD+DEPTH+BD+CACO3+PH_H2O+ CEC | 373 | 0.021 | 0.843 | 156 | 0.021 | 0.871 | - | d | PTF03 | - |
| PTF32 | PSD+DEPTH+OC+BD+CACO3+ PH_H2O+CEC | 369 | 0.021 | 0.844 | 156 | 0.021 | 0.876 | - | d | PTF03 | PTF06 |

[1]PSD: particle size distribution (sand, 50–2000 μm; silt, 2–50 μm; clay, <2 μm (mass %)); DEPTH: mean soil depth (cm); OC: organic carbon content (mass %); BD: bulk density (g $cm^{-3}$); CACO3: calcium carbonate content (mass %); PH_H2O: pH in water (-); CEC: cation exchange capacity (cmol (+) $kg^{-1}$).
[2]Different letters indicate significant differences at the 0.05 level between the accuracy of the methods based on the squared error; for example performance indicated with the letter c is significantly better than the one noted with letters b and a.

**Table 3.** Performance of pedotransfer functions (PTFs) by input combination on training and test datasets to predict water content at -100 cm matric potential head (FC_2). N: number of samples, RMSE: root mean square error (cm$^3$ cm$^{-3}$), and R$^2$: determination coefficient, TEST_BASIC: samples with measured PSD, DEPTH, OC and BD; TEST_CHEM+: samples with measured PSD, DEPTH, OC, BD, CACO3, PH_H2O and CEC. Recommended PTFs are highlighted in bold. FC_2 was not analysed in euptfv1.

| Name of PTF in euptfv2 | Predictor variables[1] | Training set | | | Test set | | | Sign. difference[2] | | Recom- mended PTF | Pair from euptfv1 |
|---|---|---|---|---|---|---|---|---|---|---|---|
| | | N | RMSE | R$^2$ | N | RMSE | R$^2$ | TEST_BASIC set | TEST_CHEM+ set | | |
| **PTF01** | **PSD+DEPTH** | 5109 | 0.062 | 0.651 | 1762 | 0.060 | 0.669 | a | a | PTF01 | - |
| **PTF02** | **PSD+DEPTH+OC** | 4131 | 0.057 | 0.711 | 1762 | 0.055 | 0.718 | b | ab | PTF02 | - |
| **PTF03** | **PSD+DEPTH+BD** | 5034 | 0.053 | 0.750 | 1762 | 0.052 | 0.745 | bc | bcdef | PTF03 | - |
| **PTF04** | **PSD+DEPTH+CACO3** | 1670 | 0.052 | 0.566 | 296 | 0.054 | 0.467 | - | abcd | PTF01 | - |
| **PTF05** | **PSD+DEPTH+PH_H2O** | 2062 | 0.056 | 0.630 | 296 | 0.056 | 0.419 | - | abc | PTF01 | - |
| **PTF06** | **PSD+DEPTH+CEC** | 1649 | 0.056 | 0.658 | 296 | 0.054 | 0.469 | - | abcde | PTF01 | - |
| **PTF07** | **PSD+DEPTH+OC+BD** | 4117 | 0.051 | 0.769 | 1762 | 0.050 | 0.769 | c | bcdefg | PTF03 | - |
| **PTF08** | **PSD+DEPTH+OC+CACO3** | 1586 | 0.050 | 0.589 | 296 | 0.049 | 0.565 | - | bcdefgh | PTF02 | - |
| **PTF09** | **PSD+DEPTH+OC+PH_H2O** | 1808 | 0.050 | 0.679 | 296 | 0.048 | 0.581 | - | bcdefg | PTF02 | - |
| PTF10 | PSD+DEPTH+OC+CEC | 1437 | 0.051 | 0.688 | 296 | 0.049 | 0.554 | - | cdefghij | PTF06 | - |
| **PTF11** | **PSD+DEPTH+BD+CACO3** | 1666 | 0.044 | 0.701 | 296 | 0.046 | 0.616 | - | fghijklmn | PTF03 | - |
| **PTF12** | **PSD+DEPTH+BD+PH_H2O** | 2008 | 0.046 | 0.746 | 296 | 0.043 | 0.657 | - | efghijkl | PTF03 | - |
| **PTF13** | **PSD+DEPTH+BD+CEC** | 1596 | 0.046 | 0.763 | 296 | 0.046 | 0.614 | - | hijklmn | PTF13 | - |
| PTF14 | PSD+DEPTH+CACO3+PH_H2O | 1314 | 0.051 | 0.600 | 296 | 0.051 | 0.528 | - | bcdef | PTF05 | - |
| PTF15 | PSD+DEPTH+CACO3+CEC | 770 | 0.052 | 0.605 | 296 | 0.051 | 0.520 | - | cdefghij | PTF04 | - |
| PTF16 | PSD+DEPTH+PH_H2O+CEC | 1350 | 0.053 | 0.699 | 296 | 0.049 | 0.556 | - | cdefghi | PTF05 | - |
| PTF17 | PSD+DEPTH+OC+BD+CACO3 | 1585 | 0.043 | 0.689 | 296 | 0.045 | 0.634 | - | ghijklmn | PTF07 | - |
| **PTF18** | **PSD+DEPTH+OC+BD+PH_H2O** | 1799 | 0.044 | 0.749 | 296 | 0.042 | 0.679 | - | ghijklmn | PTF07 | - |
| PTF19 | PSD+DEPTH+OC+BD+CEC | 1427 | 0.045 | 0.753 | 296 | 0.044 | 0.650 | - | jklmn | PTF13 | - |
| PTF20 | PSD+DEPTH+OC+CACO3+PH_H2O | 1249 | 0.049 | 0.613 | 296 | 0.053 | 0.483 | - | bcdefgh | PTF02 | - |
| PTF21 | PSD+DEPTH+OC+CACO3+CEC | 727 | 0.050 | 0.603 | 296 | 0.046 | 0.620 | - | fghijklmn | PTF08 | - |
| PTF22 | PSD+DEPTH+OC+PH_H2O+CEC | 1142 | 0.051 | 0.693 | 296 | 0.045 | 0.630 | - | efghijklm | PTF09 | - |
| PTF23 | PSD+DEPTH+BD+CACO3+PH_H2O | 1310 | 0.044 | 0.701 | 296 | 0.045 | 0.629 | - | defghijkl | PTF03 | - |
| PTF24 | PSD+DEPTH+BD+CACO3+CEC | 768 | 0.043 | 0.722 | 296 | 0.043 | 0.666 | - | lmn | PTF11 | - |
| PTF25 | PSD+DEPTH+BD+PH_H2O+CEC | 1298 | 0.046 | 0.773 | 296 | 0.043 | 0.668 | - | jklmn | PTF12 | - |
| PTF26 | PSD+DEPTH+CACO3+PH_H2O+CEC | 727 | 0.051 | 0.633 | 296 | 0.048 | 0.587 | - | defghijk | PTF05 | - |
| PTF27 | PSD+DEPTH+OC+BD+CACO3+ PH_H2O | 1248 | 0.043 | 0.693 | 296 | 0.044 | 0.653 | - | efghijklm | PTF07 | - |
| PTF28 | PSD+DEPTH+OC+BD+CACO3+CEC | 726 | 0.044 | 0.702 | 296 | 0.041 | 0.687 | - | klmn | PTF11 | - |
| PTF29 | PSD+DEPTH+OC+BD+PH_H2O+CEC | 1133 | 0.046 | 0.757 | 296 | 0.042 | 0.681 | - | ijklmn | PTF12 | - |
| PTF30 | PSD+DEPTH+OC+CACO3+PH_H2O+ CEC | 684 | 0.050 | 0.617 | 296 | 0.051 | 0.533 | - | efghijklm | PTF09 | - |
| PTF31 | PSD+DEPTH+BD+CACO3+PH_H2O+ CEC | 725 | 0.043 | 0.731 | 296 | 0.041 | 0.698 | - | mn | PTF11 | - |
| PTF32 | PSD+DEPTH+OC+BD+CACO3+ PH_H2O+CEC | 683 | 0.044 | 0.712 | 296 | 0.040 | 0.709 | - | n | PTF18 | - |

[1]PSD: particle size distribution (sand, 50–2000 μm; silt, 2–50 μm; clay, <2 μm (mass %)); DEPTH: mean soil depth (cm); OC: organic carbon content (mass %); BD: bulk density (g cm$^{-3}$); CACO3: calcium carbonate content (mass %); PH_H2O: pH in water (-); CEC: cation exchange capacity (cmol (+) kg$^{-1}$).
[2]Different letters indicate significant differences at the 0.05 level between the accuracy of the methods based on the squared error; for example performance indicated with the letter c is significantly better than the one noted with letters b and a.

**Table 4.** Performance of pedotransfer functions (PTFs) by input combination on training and test datasets to predict water content at -330 cm matric potential head, field capacity (FC). N: number of samples, RMSE: root mean square error (cm³ cm⁻³), and R²: determination coefficient, TEST_BASIC: samples with measured PSD, DEPTH, OC and BD; TEST_CHEM+: samples with measured PSD, DEPTH, OC, BD, CACO3, PH_H2O and CEC. Recommended PTFs are highlighted in bold.

| Name of PTF in euptfv2 | Predictor variables[1] | Training set N | RMSE | R² | Test set N | RMSE | R² | TEST_BASIC set | TEST_CHEM+ set | Recom-mended PTF | Pair from euptfv1 |
|---|---|---|---|---|---|---|---|---|---|---|---|
| **PTF01** | **PSD+DEPTH** | 2196 | 0.056 | 0.639 | 801 | 0.054 | 0.595 | a | a | PTF01 | - |
| **PTF02** | **PSD+DEPTH+OC** | 1716 | 0.049 | 0.707 | 801 | 0.050 | 0.650 | b | abc | PTF02 | PTF09 |
| **PTF03** | **PSD+DEPTH+BD** | 2176 | 0.048 | 0.727 | 801 | 0.049 | 0.668 | ab | abcd | PTF01 | - |
| **PTF04** | **PSD+DEPTH+CACO3** | 1537 | 0.047 | 0.650 | 280 | 0.055 | 0.591 | - | abcde | PTF01 | - |
| PTF05 | PSD+DEPTH+PH_H2O | 1278 | 0.048 | 0.653 | 280 | 0.055 | 0.586 | - | ab | PTF01 | - |
| **PTF06** | **PSD+DEPTH+CEC** | 1097 | 0.046 | 0.711 | 280 | 0.052 | 0.630 | - | bcdefghi | PTF06 | - |
| **PTF07** | **PSD+DEPTH+OC+BD** | 1711 | 0.046 | 0.736 | 801 | 0.048 | 0.677 | b | bcdefg | PTF02 | PTF09 |
| **PTF08** | **PSD+DEPTH+OC+CACO3** | 1340 | 0.043 | 0.678 | 280 | 0.053 | 0.616 | - | abcdef | PTF02 | - |
| **PTF09** | **PSD+DEPTH+OC+PH_H2O** | 1100 | 0.044 | 0.687 | 280 | 0.052 | 0.631 | - | abcde | PTF02 | - |
| PTF10 | PSD+DEPTH+OC+CEC | 1001 | 0.044 | 0.720 | 280 | 0.052 | 0.628 | - | bcdefghi | PTF02 | - |
| **PTF11** | **PSD+DEPTH+BD+CACO3** | 1526 | 0.044 | 0.696 | 280 | 0.051 | 0.649 | - | bcdefgh | PTF03 | - |
| PTF12 | PSD+DEPTH+BD+PH_H2O | 1267 | 0.045 | 0.698 | 280 | 0.050 | 0.658 | - | bcdefgh | PTF03 | - |
| PTF13 | PSD+DEPTH+BD+CEC | 1093 | 0.044 | 0.741 | 280 | 0.049 | 0.678 | - | fghi | PTF06 | - |
| **PTF14** | **PSD+DEPTH+CACO3+PH_H2O** | 1235 | 0.048 | 0.667 | 280 | 0.053 | 0.623 | - | bcdef | PTF04 | - |
| PTF15 | PSD+DEPTH+CACO3+CEC | 793 | 0.047 | 0.720 | 280 | 0.052 | 0.639 | - | efghi | PTF04 | - |
| PTF16 | PSD+DEPTH+PH_H2O+CEC | 744 | 0.047 | 0.726 | 280 | 0.051 | 0.651 | - | efghi | PTF06 | - |
| PTF17 | PSD+DEPTH+OC+BD+CACO3 | 1338 | 0.042 | 0.699 | 280 | 0.050 | 0.667 | - | cdefghi | PTF02 | - |
| PTF18 | PSD+DEPTH+OC+BD+PH_H2O | 1098 | 0.043 | 0.704 | 280 | 0.050 | 0.660 | - | bcdefgh | PTF02 | PTF09 |
| PTF19 | PSD+DEPTH+OC+BD+CEC | 998 | 0.042 | 0.739 | 280 | 0.048 | 0.684 | - | fghi | PTF07 | - |
| PTF20 | PSD+DEPTH+OC+CACO3+PH_H2O | 1062 | 0.044 | 0.694 | 280 | 0.052 | 0.634 | - | abcde | PTF02 | - |
| PTF21 | PSD+DEPTH+OC+CACO3+CEC | 709 | 0.045 | 0.709 | 280 | 0.051 | 0.652 | - | efghi | PTF04 | - |
| PTF22 | PSD+DEPTH+OC+PH_H2O+CEC | 663 | 0.046 | 0.706 | 280 | 0.050 | 0.664 | - | defghi | PTF09 | - |
| PTF23 | PSD+DEPTH+BD+CACO3+PH_H2O | 1224 | 0.045 | 0.704 | 280 | 0.051 | 0.651 | - | bcdefgh | PTF03 | - |
| PTF24 | PSD+DEPTH+BD+CACO3+CEC | 790 | 0.044 | 0.744 | 280 | 0.048 | 0.688 | - | hi | PTF11 | - |
| PTF25 | PSD+DEPTH+BD+PH_H2O+CEC | 741 | 0.045 | 0.748 | 280 | 0.048 | 0.682 | - | hi | PTF11 | - |
| PTF26 | PSD+DEPTH+CACO3+PH_H2O+CEC | 734 | 0.046 | 0.742 | 280 | 0.050 | 0.658 | - | fghi | PTF14 | - |
| PTF27 | PSD+DEPTH+OC+BD+CACO3+ PH_H2O | 1060 | 0.042 | 0.712 | 280 | 0.049 | 0.676 | - | bcdefghi | PTF02 | - |
| PTF28 | PSD+DEPTH+OC+BD+CACO3+CEC | 707 | 0.043 | 0.731 | 280 | 0.048 | 0.693 | - | ghi | PTF07 | - |
| PTF29 | PSD+DEPTH+OC+BD+PH_H2O+CEC | 661 | 0.044 | 0.725 | 280 | 0.046 | 0.709 | - | fghi | PTF07 | - |
| PTF30 | PSD+DEPTH+OC+CACO3+PH_H2O+ CEC | 655 | 0.044 | 0.731 | 280 | 0.049 | 0.672 | - | fghi | PTF08 | PTF09 |
| PTF31 | PSD+DEPTH+BD+CACO3+PH_H2O+ CEC | 731 | 0.043 | 0.763 | 280 | 0.047 | 0.700 | - | i | PTF06 | - |
| PTF32 | PSD+DEPTH+OC+BD+CACO3+ PH_H2O+CEC | 653 | 0.043 | 0.743 | 280 | 0.047 | 0.696 | - | fghi | PTF07 | PTF09 |

5   [1]PSD: particle size distribution (sand, 50–2000 μm; silt, 2–50 μm; clay, <2 μm (mass %)); DEPTH: mean soil depth (cm); OC: organic carbon content (mass %); BD: bulk density (g cm⁻³); CACO3: calcium carbonate content (mass %); PH_H2O: pH in water (-); CEC: cation exchange capacity (cmol (+) kg⁻¹).
[2]Different letters indicate significant differences at the 0.05 level between the accuracy of the methods based on the squared error; for example performance indicated with the letter c is significantly better than the one noted with letters b and a.

**Table 5.** Performance of pedotransfer functions (PTFs) by input combination on training and test datasets to predict water content at wilting point (WP). N: number of samples, RMSE: root mean square error ($cm^3$ $cm^{-3}$), and $R^2$: determination coefficient, TEST_BASIC: samples with measured PSD, DEPTH, OC and BD; TEST_CHEM+: samples with measured PSD, DEPTH, OC, BD, CACO3, PH_H2O and CEC. Recommended PTFs are highlighted in bold.

| Name of PTF in euptfv2 | Predictor variables[1] | Training set | | | Test set | | | Sign. difference[2] | | Recom-mended PTF | Pair from euptfv1 |
|---|---|---|---|---|---|---|---|---|---|---|---|
| | | N | RMSE | $R^2$ | N | RMSE | $R^2$ | TEST_BASIC set | TEST_CHEM+ set | | |
| **PTF01** | **PSD+DEPTH** | 5264 | 0.048 | 0.736 | 2088 | 0.048 | 0.728 | a | a | PTF01 | - |
| **PTF02** | **PSD+DEPTH+OC** | 4802 | 0.047 | 0.755 | 2088 | 0.046 | 0.745 | bc | abc | PTF02 | PTF12 |
| PTF03 | PSD+DEPTH+BD | 5197 | 0.046 | 0.757 | 2088 | 0.046 | 0.754 | ab | ab | PTF01 | - |
| PTF04 | PSD+DEPTH+CACO3 | 1816 | 0.042 | 0.693 | 294 | 0.042 | 0.643 | - | a | PTF01 | - |
| PTF05 | PSD+DEPTH+PH_H2O | 2039 | 0.046 | 0.673 | 294 | 0.044 | 0.621 | - | abc | PTF01 | - |
| PTF06 | PSD+DEPTH+CEC | 1703 | 0.043 | 0.725 | 294 | 0.041 | 0.662 | - | a | PTF01 | - |
| **PTF07** | **PSD+DEPTH+OC+BD** | 4786 | 0.045 | 0.769 | 2088 | 0.044 | 0.769 | c | abc | PTF02 | PTF12 |
| **PTF08** | **PSD+DEPTH+OC+CACO3** | 1599 | 0.041 | 0.695 | 294 | 0.041 | 0.671 | - | abcd | PTF02 | - |
| **PTF09** | **PSD+DEPTH+OC+PH_H2O** | 1678 | 0.045 | 0.682 | 294 | 0.041 | 0.661 | - | abcd | PTF02 | - |
| PTF10 | PSD+DEPTH+OC+CEC | 1459 | 0.043 | 0.704 | 294 | 0.040 | 0.674 | - | abcd | PTF02 | - |
| PTF11 | PSD+DEPTH+BD+CACO3 | 1806 | 0.041 | 0.706 | 294 | 0.040 | 0.682 | - | abcd | PTF01 | - |
| PTF12 | PSD+DEPTH+BD+PH_H2O | 1979 | 0.045 | 0.691 | 294 | 0.041 | 0.671 | - | abcd | PTF01 | - |
| PTF13 | PSD+DEPTH+BD+CEC | 1648 | 0.042 | 0.729 | 294 | 0.040 | 0.683 | - | abcd | PTF01 | - |
| PTF14 | PSD+DEPTH+CACO3+PH_H2O | 1375 | 0.043 | 0.689 | 294 | 0.042 | 0.649 | - | abcd | PTF01 | - |
| PTF15 | PSD+DEPTH+CACO3+CEC | 831 | 0.044 | 0.657 | 294 | 0.039 | 0.694 | - | abcd | PTF01 | - |
| PTF16 | PSD+DEPTH+PH_H2O+CEC | 1349 | 0.043 | 0.727 | 294 | 0.040 | 0.681 | - | abc | PTF01 | - |
| **PTF17** | **PSD+DEPTH+OC+BD+CACO3** | 1596 | 0.041 | 0.705 | 294 | 0.039 | 0.702 | - | abcd | PTF07 | - |
| PTF18 | PSD+DEPTH+OC+BD+PH_H2O | 1667 | 0.045 | 0.687 | 294 | 0.040 | 0.674 | - | abcd | PTF07 | PTF12 |
| PTF19 | PSD+DEPTH+OC+BD+CEC | 1447 | 0.042 | 0.714 | 294 | 0.039 | 0.691 | - | abcd | PTF07 | - |
| PTF20 | PSD+DEPTH+OC+CACO3+PH_H2O | 1183 | 0.042 | 0.691 | 294 | 0.040 | 0.686 | - | abcd | PTF02 | - |
| PTF21 | PSD+DEPTH+OC+CACO3+CEC | 743 | 0.044 | 0.638 | 294 | 0.037 | 0.722 | - | d | PTF08 | - |
| PTF22 | PSD+DEPTH+OC+PH_H2O+CEC | 1121 | 0.044 | 0.697 | 294 | 0.039 | 0.701 | - | abcd | PTF07 | - |
| PTF23 | PSD+DEPTH+BD+CACO3+PH_H2O | 1365 | 0.042 | 0.701 | 294 | 0.040 | 0.678 | - | abcd | PTF01 | - |
| PTF24 | PSD+DEPTH+BD+CACO3+CEC | 827 | 0.043 | 0.673 | 294 | 0.038 | 0.708 | - | abcd | PTF01 | - |
| PTF25 | PSD+DEPTH+BD+PH_H2O+CEC | 1295 | 0.043 | 0.726 | 294 | 0.039 | 0.698 | - | abcd | PTF01 | - |
| PTF26 | PSD+DEPTH+CACO3+PH_H2O+CEC | 772 | 0.043 | 0.680 | 294 | 0.039 | 0.702 | - | cd | PTF05 | - |
| PTF27 | PSD+DEPTH+OC+BD+CACO3+PH_H2O | 1180 | 0.042 | 0.698 | 294 | 0.039 | 0.703 | - | abcd | PTF07 | - |
| PTF28 | PSD+DEPTH+OC+BD+CACO3+CEC | 740 | 0.043 | 0.648 | 294 | 0.037 | 0.732 | - | bcd | PTF17 | - |
| PTF29 | PSD+DEPTH+OC+BD+PH_H2O+CEC | 1110 | 0.043 | 0.699 | 294 | 0.038 | 0.712 | - | abcd | PTF07 | - |
| PTF30 | PSD+DEPTH+OC+CACO3+PH_H2O+CEC | 689 | 0.044 | 0.645 | 294 | 0.038 | 0.719 | - | abcd | PTF02 | PTF12 |
| PTF31 | PSD+DEPTH+BD+CACO3+PH_H2O+CEC | 768 | 0.043 | 0.678 | 294 | 0.037 | 0.720 | - | cd | PTF05 | - |
| PTF32 | PSD+DEPTH+OC+BD+CACO3+PH_H2O+CEC | 686 | 0.043 | 0.656 | 294 | 0.037 | 0.723 | - | d | PTF09 | PTF12 |

5  [1]PSD: particle size distribution (sand, 50–2000 μm; silt, 2–50 μm; clay, <2 μm (mass %)); DEPTH: mean soil depth (cm); OC: organic carbon content (mass %); BD: bulk density (g $cm^{-3}$); CACO3: calcium carbonate content (mass %); PH_H2O: pH in water (-); CEC: cation exchange capacity (cmol (+) $kg^{-1}$).
[2]Different letters indicate significant differences at the 0.05 level between the accuracy of the methods based on the squared error; for example performance indicated with the letter c is significantly better than the one noted with letters b and a.

**Table 6.** Performance of pedotransfer functions (PTFs) by input combination on training and test datasets to predict saturated hydraulic conductivity (KS). N: number of samples, RMSE: root mean square error ($\log_{10}$ (cm day$^{-1}$)), and $R^2$: determination coefficient, TEST_BASIC: samples with measured PSD, DEPTH, OC and BD; TEST_CHEM+: samples with measured PSD, DEPTH, OC, BD, CACO3, PH_H2O and CEC. Recommended PTFs are highlighted in bold.

| Name of PTF in euptfv2 | Predictor variables[1] | Training set N | RMSE | $R^2$ | Test set N | RMSE | $R^2$ | Sign. difference[2] TEST_BASIC set | TEST_CHEM+ set | Recommended PTF | Pair from euptfv1 |
|---|---|---|---|---|---|---|---|---|---|---|---|
| **PTF01** | **PSD+DEPTH** | 3157 | 1.200 | 0.434 | 1117 | 1.181 | 0.307 | a | ab | PTF01 | - |
| **PTF02** | **PSD+DEPTH+OC** | 2620 | 0.957 | 0.566 | 1117 | 0.953 | 0.548 | b | bc | PTF02 | PTF16 |
| PTF03 | PSD+DEPTH+BD | 3146 | 1.160 | 0.467 | 1117 | 1.170 | 0.320 | a | a | PTF01 | - |
| PTF04 | PSD+DEPTH+CACO3 | 639 | 0.861 | 0.241 | 169 | 0.959 | 0.123 | - | abc | PTF01 | - |
| **PTF05** | **PSD+DEPTH+PH_H2O** | 907 | 0.875 | 0.213 | 169 | 0.944 | 0.151 | - | bc | PTF01 | - |
| PTF06 | PSD+DEPTH+CEC | 567 | 0.984 | 0.215 | 169 | 0.940 | 0.157 | - | bc | PTF01 | - |
| PTF07 | PSD+DEPTH+OC+BD | 2609 | 0.931 | 0.590 | 1117 | 0.939 | 0.562 | b | bc | PTF02 | PTF16 |
| PTF08 | PSD+DEPTH+OC+CACO3 | 613 | 0.872 | 0.244 | 169 | 0.943 | 0.153 | - | bc | PTF02 | - |
| PTF09 | PSD+DEPTH+OC+PH_H2O | 862 | 0.847 | 0.257 | 169 | 0.938 | 0.162 | - | bc | PTF02 | - |
| PTF10 | PSD+DEPTH+OC+CEC | 525 | 0.977 | 0.223 | 169 | 0.938 | 0.162 | - | bc | PTF02 | - |
| PTF11 | PSD+DEPTH+BD+CACO3 | 639 | 0.851 | 0.259 | 169 | 0.952 | 0.136 | - | bc | PTF01 | - |
| PTF12 | PSD+DEPTH+BD+PH_H2O | 898 | 0.853 | 0.256 | 169 | 0.947 | 0.145 | - | bc | PTF05 | - |
| PTF13 | PSD+DEPTH+BD+CEC | 558 | 0.980 | 0.230 | 169 | 0.941 | 0.157 | - | bc | PTF01 | - |
| PTF14 | PSD+DEPTH+CACO3+PH_H2O | 620 | 0.855 | 0.267 | 169 | 0.923 | 0.189 | - | bc | PTF05 | - |
| PTF15 | PSD+DEPTH+CACO3+CEC | 405 | 0.937 | 0.263 | 169 | 0.941 | 0.156 | - | abc | PTF01 | - |
| PTF16 | PSD+DEPTH+PH_H2O+CEC | 567 | 0.942 | 0.282 | 169 | 0.940 | 0.158 | - | bc | PTF01 | - |
| PTF17 | PSD+DEPTH+OC+BD+CACO3 | 613 | 0.856 | 0.272 | 169 | 0.933 | 0.171 | - | bc | PTF02 | - |
| PTF18 | PSD+DEPTH+OC+BD+PH_H2O | 853 | 0.831 | 0.289 | 169 | 0.932 | 0.172 | - | bc | PTF02 | PTF16 |
| PTF19 | PSD+DEPTH+OC+BD+CEC | 516 | 0.979 | 0.228 | 169 | 0.928 | 0.179 | - | c | PTF02 | - |
| PTF20 | PSD+DEPTH+OC+CACO3+PH_H2O | 613 | 0.860 | 0.264 | 169 | 0.929 | 0.177 | - | bc | PTF02 | - |
| PTF21 | PSD+DEPTH+OC+CACO3+CEC | 401 | 0.935 | 0.271 | 169 | 0.925 | 0.184 | - | bc | PTF02 | - |
| PTF22 | PSD+DEPTH+OC+PH_H2O+CEC | 525 | 0.931 | 0.295 | 169 | 0.933 | 0.170 | - | c | PTF02 | - |
| PTF23 | PSD+DEPTH+BD+CACO3+PH_H2O | 620 | 0.844 | 0.286 | 169 | 0.889 | 0.247 | - | c | PTF05 | - |
| PTF24 | PSD+DEPTH+BD+CACO3+CEC | 405 | 0.922 | 0.286 | 169 | 0.958 | 0.125 | - | abc | PTF01 | - |
| PTF25 | PSD+DEPTH+BD+PH_H2O+CEC | 558 | 0.944 | 0.286 | 169 | 0.950 | 0.140 | - | bc | PTF05 | - |
| PTF26 | PSD+DEPTH+CACO3+PH_H2O+CEC | 405 | 0.922 | 0.286 | 169 | 0.922 | 0.190 | - | bc | PTF05 | - |
| PTF27 | PSD+DEPTH+OC+BD+CACO3+PH_H2O | 613 | 0.844 | 0.293 | 169 | 0.893 | 0.241 | - | c | PTF02 | - |
| PTF28 | PSD+DEPTH+OC+BD+CACO3+CEC | 401 | 0.926 | 0.285 | 169 | 0.925 | 0.185 | - | abc | PTF02 | - |
| PTF29 | PSD+DEPTH+OC+BD+PH_H2O+CEC | 516 | 0.932 | 0.301 | 169 | 0.921 | 0.193 | - | bc | PTF02 | - |
| PTF30 | PSD+DEPTH+OC+CACO3+PH_H2O+CEC | 401 | 0.931 | 0.278 | 169 | 0.887 | 0.250 | - | bc | PTF02 | PTF17 |
| PTF31 | PSD+DEPTH+BD+CACO3+PH_H2O+CEC | 405 | 0.914 | 0.298 | 169 | 0.912 | 0.207 | - | bc | PTF05 | - |
| PTF32 | PSD+DEPTH+OC+BD+CACO3+PH_H2O+CEC | 401 | 0.921 | 0.292 | 169 | 0.916 | 0.201 | - | bc | PTF02 | PTF17 |

5  [1]PSD: particle size distribution (sand, 50–2000 μm; silt, 2–50 μm; clay, <2 μm (mass %)); DEPTH: mean soil depth (cm); OC: organic carbon content (mass %); BD: bulk density (g cm$^{-3}$); CACO3: calcium carbonate content (mass %); PH_H2O: pH in water (-); CEC: cation exchange capacity (cmol (+) kg$^{-1}$).
[2]Different letters indicate significant differences at the 0.05 level between the accuracy of the methods based on the squared error; for example performance indicated with the letter c is significantly better than the one noted with letters b and a.

**Table 7.** Performance of pedotransfer functions (PTFs) by input combination on training and test datasets to predict parameters of the van Genuchten model to describe soil moisture retention curve (VG). N: number of samples, RMSE: root mean square error (cm$^3$ cm$^{-3}$), and R$^2$: determination coefficient, TEST_BASIC: samples with measured PSD, DEPTH, OC and BD; TEST_CHEM+: samples with measured PSD, DEPTH, OC, BD, CACO3, PH_H2O and CEC. Recommended PTFs are highlighted in bold.

| Name of PTF in euptfv2 | Predictor variables[1] | Training set N | Training set RMSE | Training set R$^2$ | Test set N | Test set RMSE | Test set R$^2$ | Sign. difference[2] TEST_BASIC set | Sign. difference[2] TEST_CHEM+ set | Recommended PTF | Pair from euptfv1 |
|---|---|---|---|---|---|---|---|---|---|---|---|
| **PTF01** | **PSD+DEPTH** | 4669 | 0.055 | 0.846 | 1591 | 0.068 | 0.776 | a | a | PTF01 | - |
| **PTF02** | **PSD+DEPTH+OC** | 3708 | 0.047 | 0.887 | 1591 | 0.060 | 0.826 | b | c | PTF02 | PTF19 |
| **PTF03** | **PSD+DEPTH+BD** | 4593 | 0.041 | 0.913 | 1591 | 0.056 | 0.846 | c | hi | PTF03 | - |
| **PTF04** | **PSD+DEPTH+CACO3** | 1671 | 0.039 | 0.911 | 288 | 0.052 | 0.852 | - | d | PTF04 | - |
| **PTF05** | **PSD+DEPTH+PH_H2O** | 1897 | 0.045 | 0.894 | 288 | 0.055 | 0.834 | - | b | PTF05 | - |
| **PTF06** | **PSD+DEPTH+CEC** | 1488 | 0.044 | 0.886 | 288 | 0.054 | 0.839 | - | d | PTF06 | - |
| **PTF07** | **PSD+DEPTH+OC+BD** | 3695 | 0.037 | 0.933 | 1591 | 0.054 | 0.859 | d | fg | PTF07 | PTF21 |
| **PTF08** | **PSD+DEPTH+OC+CACO3** | 1589 | 0.036 | 0.924 | 288 | 0.048 | 0.871 | - | f | PTF08 | - |
| **PTF09** | **PSD+DEPTH+OC+PH_H2O** | 1663 | 0.039 | 0.922 | 288 | 0.050 | 0.865 | - | gh | PTF09 | - |
| **PTF10** | **PSD+DEPTH+OC+CEC** | 1293 | 0.036 | 0.920 | 288 | 0.051 | 0.858 | - | fg | PTF10 | - |
| **PTF11** | **PSD+DEPTH+BD+CACO3** | 1670 | 0.034 | 0.934 | 288 | 0.043 | 0.900 | - | mn | PTF11 | - |
| **PTF12** | **PSD+DEPTH+BD+PH_H2O** | 1847 | 0.038 | 0.926 | 288 | 0.044 | 0.892 | - | l | PTF12 | - |
| **PTF13** | **PSD+DEPTH+BD+CEC** | 1437 | 0.039 | 0.908 | 288 | 0.044 | 0.892 | - | lm | PTF13 | - |
| **PTF14** | **PSD+DEPTH+CACO3+PH_H2O** | 1264 | 0.037 | 0.928 | 288 | 0.052 | 0.854 | - | e | PTF14 | - |
| **PTF15** | **PSD+DEPTH+CACO3+CEC** | 758 | 0.040 | 0.907 | 288 | 0.049 | 0.870 | - | ij | PTF15 | - |
| **PTF16** | **PSD+DEPTH+PH_H2O+CEC** | 1188 | 0.042 | 0.905 | 288 | 0.051 | 0.858 | - | f | PTF16 | - |
| PTF17 | PSD+DEPTH+OC+BD+CACO3 | 1588 | 0.031 | 0.944 | 288 | 0.042 | 0.904 | - | n | PTF11 | - |
| PTF18 | PSD+DEPTH+OC+BD+PH_H2O | 1655 | 0.033 | 0.943 | 288 | 0.043 | 0.900 | - | l | PTF12 | PTF22 |
| PTF19 | PSD+DEPTH+OC+BD+CEC | 1284 | 0.033 | 0.934 | 288 | 0.044 | 0.892 | - | lm | PTF13 | - |
| PTF20 | PSD+DEPTH+OC+CACO3+PH_H2O | 1201 | 0.033 | 0.943 | 288 | 0.048 | 0.874 | - | f | PTF09 | - |
| **PTF21** | **PSD+DEPTH+OC+CACO3+CEC** | 712 | 0.035 | 0.932 | 288 | 0.047 | 0.881 | - | l | PTF21 | - |
| **PTF22** | **PSD+DEPTH+OC+PH_H2O+CEC** | 996 | 0.033 | 0.939 | 288 | 0.049 | 0.869 | - | i | PTF22 | - |
| PTF23 | PSD+DEPTH+BD+CACO3+PH_H2O | 1263 | 0.032 | 0.948 | 288 | 0.044 | 0.895 | - | lm | PTF11 | - |
| **PTF24** | **PSD+DEPTH+BD+CACO3+CEC** | 757 | 0.033 | 0.939 | 288 | 0.041 | 0.906 | - | o | PTF24 | - |
| **PTF25** | **PSD+DEPTH+BD+PH_H2O+CEC** | 1138 | 0.038 | 0.922 | 288 | 0.042 | 0.902 | - | n | PTF25 | - |
| PTF26 | PSD+DEPTH+CACO3+PH_H2O+CEC | 717 | 0.037 | 0.924 | 288 | 0.047 | 0.878 | - | jk | PTF15 | - |
| PTF27 | PSD+DEPTH+OC+BD+CACO3+PH_H2O | 1200 | 0.030 | 0.953 | 288 | 0.043 | 0.897 | - | lm | PTF11 | - |
| PTF28 | PSD+DEPTH+OC+BD+CACO3+CEC | 711 | 0.032 | 0.941 | 288 | 0.041 | 0.906 | - | o | PTF24 | - |
| **PTF29** | **PSD+DEPTH+OC+BD+PH_H2O+CEC** | 988 | 0.032 | 0.945 | 288 | 0.041 | 0.906 | - | o | PTF29 | - |
| PTF30 | PSD+DEPTH+OC+CACO3+PH_H2O+CEC | 671 | 0.031 | 0.946 | 288 | 0.047 | 0.880 | - | k | PTF21 | PTF20 |
| PTF31 | PSD+DEPTH+BD+CACO3+PH_H2O+CEC | 716 | 0.031 | 0.948 | 288 | 0.042 | 0.904 | - | o | PTF24 | - |
| PTF32 | PSD+DEPTH+OC+BD+CACO3+PH_H2O+CEC | 670 | 0.031 | 0.948 | 288 | 0.042 | 0.903 | - | o | PTF29 | PTF22 |

[1]PSD: particle size distribution (sand, 50–2000 μm; silt, 2–50 μm; clay, <2 μm (mass %)); DEPTH: mean soil depth (cm); OC: organic carbon content (mass %); BD: bulk density (g cm$^{-3}$); CACO3: calcium carbonate content (mass %); PH_H2O: pH in water (-); CEC: cation exchange capacity (cmol (+) kg$^{-1}$).
[2]Different letters indicate significant differences at the 0.05 level between the accuracy of the methods based on the squared error; for example performance indicated with the letter c is significantly better than the one noted with letters b and a.

**Table 8.** Performance of pedotransfer functions (PTFs) by input combination on training and test datasets to predict parameters of the Mualem-van Genuchten model to describe soil moisture retention and hydraulic conductivity curve (MVG). N: number of samples, RMSE: root mean square error ($\log_{10}$ (cm day$^{-1}$)), and R$^2$: determination coefficient, TEST_BASIC: samples with measured PSD, DEPTH, OC and BD; TEST_CHEM+: samples with measured PSD, DEPTH, OC, BD, CACO3, PH_H2O and CEC. Recommended PTFs are highlighted in bold.

| Name of PTF in euptfv2 | Predictor variables[1] | Training set | | | Test set | | | Sign. difference[2] | | Recommended PTF | Pair from euptfv1 |
|---|---|---|---|---|---|---|---|---|---|---|---|
| | | N | RMSE | R$^2$ | N | RMSE | R$^2$ | TEST_BASIC set | TEST_CHEM+ set | | |
| **PTF01** | **PSD+DEPTH** | 739 | 0.604 | 0.804 | 176 | 0.708 | 0.796 | a | b | PTF01 | - |
| **PTF02** | **PSD+DEPTH+OC** | 407 | 0.619 | 0.829 | 176 | 0.676 | 0.814 | b | jkl | PTF02 | PTF19 |
| PTF03 | PSD+DEPTH+BD | 726 | 0.568 | 0.824 | 176 | 0.688 | 0.808 | a | ab | PTF01 | - |
| **PTF04** | **PSD+DEPTH+CACO3** | 273 | 0.587 | 0.878 | 57 | 0.644 | 0.863 | - | ijk | PTF04 | - |
| **PTF05** | **PSD+DEPTH+PH_H2O** | 230 | 0.578 | 0.889 | 57 | 0.663 | 0.855 | - | def | PTF05 | - |
| **PTF06** | **PSD+DEPTH+CEC** | 141 | 0.672 | 0.858 | 57 | 0.662 | 0.856 | - | fghij | PTF06 | - |
| PTF07 | PSD+DEPTH+OC+BD | 404 | 0.529 | 0.873 | 176 | 0.659 | 0.824 | b | a | PTF02 | PTF19 |
| PTF08 | PSD+DEPTH+OC+CACO3 | 250 | 0.587 | 0.880 | 57 | 0.699 | 0.839 | - | b | PTF02 | - |
| PTF09 | PSD+DEPTH+OC+PH_H2O | 224 | 0.597 | 0.882 | 57 | 0.686 | 0.845 | - | fghi | PTF02 | - |
| PTF10 | PSD+DEPTH+OC+CEC | 138 | 0.699 | 0.846 | 57 | 0.702 | 0.837 | - | cde | PTF02 | - |
| PTF11 | PSD+DEPTH+BD+CACO3 | 272 | 0.542 | 0.895 | 57 | 0.637 | 0.866 | - | defg | PTF04 | - |
| **PTF12** | **PSD+DEPTH+BD+PH_H2O** | 229 | 0.520 | 0.909 | 57 | 0.620 | 0.873 | - | jklm | PTF12 | - |
| **PTF13** | **PSD+DEPTH+BD+CEC** | 140 | 0.644 | 0.866 | 57 | 0.637 | 0.866 | - | lm | PTF13 | - |
| PTF14 | PSD+DEPTH+CACO3+PH_H2O | 223 | 0.539 | 0.904 | 57 | 0.691 | 0.842 | - | c | PTF04 | - |
| PTF15 | PSD+DEPTH+CACO3+CEC | 136 | 0.735 | 0.830 | 57 | 0.684 | 0.846 | - | c | PTF04 | - |
| PTF16 | PSD+DEPTH+PH_H2O+CEC | 141 | 0.666 | 0.860 | 57 | 0.666 | 0.854 | - | hijk | PTF06 | - |
| PTF17 | PSD+DEPTH+OC+BD+CACO3 | 249 | 0.526 | 0.902 | 57 | 0.662 | 0.855 | - | ab | PTF02 | - |
| PTF18 | PSD+DEPTH+OC+BD+PH_H2O | 223 | 0.553 | 0.897 | 57 | 0.642 | 0.864 | - | klm | PTF02 | PTF19 |
| PTF19 | PSD+DEPTH+OC+BD+CEC | 137 | 0.619 | 0.876 | 57 | 0.676 | 0.849 | - | b | PTF02 | - |
| **PTF20** | **PSD+DEPTH+OC+CACO3+PH_H2O** | 219 | 0.573 | 0.891 | 57 | 0.661 | 0.856 | - | n | PTF20 | - |
| **PTF21** | **PSD+DEPTH+OC+CACO3+CEC** | 135 | 0.730 | 0.831 | 57 | 0.653 | 0.860 | - | m | PTF21 | - |
| PTF22 | PSD+DEPTH+OC+PH_H2O+CEC | 138 | 0.699 | 0.846 | 57 | 0.664 | 0.855 | - | lm | PTF02 | - |
| **PTF23** | **PSD+DEPTH+BD+CACO3+PH_H2O** | 222 | 0.515 | 0.911 | 57 | 0.639 | 0.865 | - | lm | PTF23 | - |
| PTF24 | PSD+DEPTH+BD+CACO3+CEC | 135 | 0.678 | 0.852 | 57 | 0.656 | 0.858 | - | c | PTF04 | - |
| PTF25 | PSD+DEPTH+BD+PH_H2O+CEC | 140 | 0.595 | 0.885 | 57 | 0.646 | 0.862 | - | ghijk | PTF12 | - |
| PTF26 | PSD+DEPTH+CACO3+PH_H2O+CEC | 136 | 0.712 | 0.841 | 57 | 0.669 | 0.852 | - | cd | PTF04 | - |
| **PTF27** | **PSD+DEPTH+OC+BD+CACO3+PH_H2O** | 218 | 0.524 | 0.907 | 57 | 0.606 | 0.879 | - | o | PTF27 | - |
| **PTF28** | **PSD+DEPTH+OC+BD+CACO3+CEC** | 134 | 0.656 | 0.860 | 57 | 0.639 | 0.865 | - | n | PTF28 | - |
| **PTF29** | **PSD+DEPTH+OC+BD+PH_H2O+CEC** | 137 | 0.646 | 0.865 | 57 | 0.638 | 0.866 | - | n | PTF29 | - |
| PTF30 | PSD+DEPTH+OC+CACO3+PH_H2O+CEC | 135 | 0.726 | 0.833 | 57 | 0.680 | 0.847 | - | fghi | PTF20 | PTF19 |
| PTF31 | PSD+DEPTH+BD+CACO3+PH_H2O+CEC | 135 | 0.679 | 0.851 | 57 | 0.668 | 0.853 | - | c | PTF12 | - |
| PTF32 | PSD+DEPTH+OC+BD+CACO3+PH_H2O+CEC | 134 | 0.645 | 0.864 | 57 | 0.678 | 0.848 | - | efgh | PTF27 | PTF19 |

[1]PSD: particle size distribution (sand, 50–2000 μm; silt, 2–50 μm; clay, <2 μm (mass %)); DEPTH: mean soil depth (cm); OC: organic carbon content (mass %); BD: bulk density (g cm$^{-3}$); CACO3: calcium carbonate content (mass %); PH_H2O: pH in water (-); CEC: cation exchange capacity (cmol (+) kg$^{-1}$).
[2]Different letters indicate significant differences at the 0.05 level between the accuracy of the methods based on the squared error; for example performance indicated with the letter c is significantly better than the one noted with letters b and a.

**Table 9.** The results of comparing the performance of parametric and point pedotransfer functions (PTFs) on the test sets of EU-HYDI to predict saturated water content (THS), water content at -100 cm matric potential head (FC_2), water content at -330 cm matric potential head (FC), water content at wilting point (WP). Rows in italic indicate cases where there was no significant difference between the two PTFs.

| Predicted soil hydraulic property | Available predictor variables[1] | Performance of parameter estimation (MRC with VG)[2] | | Performance of point estimation | | Number of samples in test dataset |
|---|---|---|---|---|---|---|
| | | Recommended PTF number | RMSE | Recommended PTF number | RMSE | |
| **THS** ($cm^3\ cm^{-3}$) | *PSD+DEPTH_M+OC* | *PTF02[a]* | *0.065* | *PTF02[a]* | *0.061* | *216* |
| | PSD+DEPTH_M+OC+BD | PTF07[a] | 0.041 | PTF03[b] | 0.032 | 216 |
| | PSD+DEPTH_M+OC+BD+PH_H2O | PTF12[a] | 0.028 | PTF03[b] | 0.022 | 63 |
| | *PSD+DEPTH_M+OC+CACO3+PH_H2O+CEC* | *PTF21[a]* | *0.051* | *PTF02[a]* | *0.060* | *63* |
| | *PSD+DEPTH_M+OC+BD+CACO3+PH_H2O+CEC* | *PTF29[a]* | *0.028* | *PTF03[a]* | *0.022* | *63* |
| **FC_2** ($cm^3\ cm^{-3}$) | PSD+DEPTH_M+OC | PTF02[a] | 0.057 | PTF02[b] | 0.054 | 424 |
| | *PSD+DEPTH_M+OC+BD* | *PTF07[a]* | *0.051* | *PTF03[a]* | *0.051* | *424* |
| | *PSD+DEPTH_M+OC+BD+PH_H2O* | *PTF12[a]* | *0.043* | *PTF07[a]* | *0.049* | *68* |
| | *PSD+DEPTH_M+OC+CACO3+PH_H2O+CEC* | *PTF21[a]* | *0.043* | *PTF09[a]* | *0.047* | *68* |
| | *PSD+DEPTH_M+OC+BD+CACO3+PH_H2O+CEC* | *PTF29[a]* | *0.036* | *PTF18[a]* | *0.043* | *68* |
| **FC** ($cm^3\ cm^{-3}$) | *PSD+DEPTH_M+OC* | *PTF02[a]* | *0.057* | *PTF02[a]* | *0.048* | *319* |
| | *PSD+DEPTH_M+OC+BD* | *PTF07[a]* | *0.056* | *PTF02[a]* | *0.048* | *319* |
| | *PSD+DEPTH_M+OC+BD+PH_H2O* | *PTF12[a]* | *0.047* | *PTF02[a]* | *0.047* | *129* |
| | *PSD+DEPTH_M+OC+CACO3+PH_H2O+CEC* | *PTF21[a]* | *0.046* | *PTF08[a]* | *0.045* | *129* |
| | *PSD+DEPTH_M+OC+BD+CACO3+PH_H2O+CEC* | *PTF29[a]* | *0.041* | *PTF07[a]* | *0.046* | *129* |
| **WP** ($cm^3\ cm^{-3}$) | PSD+DEPTH_M+OC | PTF02[a] | 0.064 | PTF02[b] | 0.047 | 429 |
| | PSD+DEPTH_M+OC+BD | PTF07[a] | 0.061 | PTF02[b] | 0.047 | 429 |
| | *PSD+DEPTH_M+OC+BD+PH_H2O* | *PTF12[a]* | *0.053* | *PTF07[a]* | *0.045* | *91* |
| | *PSD+DEPTH_M+OC+CACO3+PH_H2O+CEC* | *PTF21[a]* | *0.051* | *PTF07[a]* | *0.045* | *91* |
| | *PSD+DEPTH_M+OC+BD+CACO3+PH_H2O+CEC* | *PTF29[a]* | *0.054* | *PTF09[a]* | *0.039* | *91* |

[1]PSD: particle size distribution (sand, 50–2000 μm; silt, 2–50 μm; clay, <2 μm (mass %)); DEPTH: mean soil depth (cm); OC: organic carbon content (mass %); BD: bulk density (g cm$^{-3}$); CACO3: calcium carbonate content (mass %); PH_H2O: pH in water (-); CEC: cation exchange capacity (cmol (+) kg$^{-1}$).
[2]MRC: moisture retention curve; VG: parameters of the van Genuchten model. Different letters in a row indicate significant differences at the 0.05 level between the accuracy of the methods based on the squared error; for example performance indicated with the letter b is significantly better than the one noted with letter a. RMSE: root mean squared error.

**Table 10.** The results of comparing the performance of euptfv1 and euptfv2 on the test sets of EU-HYDI to predict soil hydraulic properties. Rows in italic indicate cases where there was no significant difference between the two PTFs.

| Predicted soil hydraulic property[1] | euptfv1 | | euptfv2 | | Name of test set | Number of samples in test datasets |
|---|---|---|---|---|---|---|
| | Name of PTF | RMSE | Name of PTF | RMSE | | |
| **THS** | PTF04[a] | 0.063 | PTF02[b] | 0.056 | TEST_BASIC | 1274 |
| (cm$^3$ cm$^{-3}$) | PTF05[a] | 0.034 | PTF03[b] | 0.031 | TEST_BASIC | 1274 |
| | *PTF06[a]* | *0.020* | *PTF03[a]* | *0.024* | *TEST_CHEM+* | *156* |
| **FC** | PTF09[a] | 0.054 | PTF02[b] | 0.050 | TEST_BASIC | 801 |
| (cm$^3$ cm$^{-3}$) | PTF09[a] | 0.054 | PTF07[b] | 0.048 | TEST_BASIC | 801 |
| | PTF09[a] | 0.058 | PTF08[b] | 0.053 | TEST_CHEM+ | 280 |
| **WP** | PTF12[a] | 0.048 | PTF02[b] | 0.046 | TEST_BASIC | 2088 |
| (cm$^3$ cm$^{-3}$) | PTF12[a] | 0.048 | PTF07[b] | 0.044 | TEST_BASIC | 2088 |
| | *PTF12[a]* | *0.043* | *PTF09[a]* | *0.041* | *TEST_CHEM+* | *294* |
| **KS** | PTF16[a] | 1.06 | PTF02[b] | 0.95 | TEST_BASIC | 1117 |
| (log$_{10}$ cm day$^{-1}$) | *PTF17[a]* | *1.00* | *PTF02[a]* | *0.91* | *TEST_CHEM+* | *169* |
| **VG** | PTF19[a] | 0.068 | PTF02[b] | 0.060 | TEST_BASIC | 1591 |
| (cm$^3$ cm$^{-3}$) | PTF21[a] | 0.064 | PTF07[b] | 0.054 | TEST_BASIC | 1591 |
| | PTF22[a] | 0.046 | PTF12[b] | 0.044 | TEST_CHEM+ | 288 |
| | PTF20[a] | 0.054 | PTF21[b] | 0.047 | TEST_CHEM+ | 288 |
| | PTF22[a] | 0.046 | PTF29[b] | 0.041 | TEST_CHEM+ | 288 |
| **MVG** | PTF19[a] | 0.77 | PTF02[b] | 0.68 | TEST_BASIC | 176 |
| (log$_{10}$ cm day$^{-1}$) | *PTF19[a]* | *0.66* | *PTF20[a]* | *0.66* | *TEST_CHEM+* | *57* |
| | *PTF19[a]* | *0.66* | *PTF27[a]* | *0.61* | *TEST_CHEM+* | *57* |

[1]THS: saturated water content (pF 0); FC_2: water content at -100 cm matric potential head (pF 2.0); FC: water content at -330 cm matric potential head (pF 2.5); WP: water content at wilting point (pF 4.2); KS: saturated hydraulic conductivity; VG: parameters of the van Genuchten model; MVG: parameters of the Mualem – van Genuchten model.

[2]Different letters in a row indicate significant differences at the 0.05 level between the accuracy of the methods based on the squared error;for example performance indicated with the letter b is significantly better than the one noted with letter a. RMSE: root mean squared error; TEST_BASIC: samples with measured PSD, DEPTH, OC and BD; TEST_CHEM+: samples with measured PSD, DEPTH, OC, BD, CACO3, PH_H2O and CEC; N: number of samples.

**Table 11.** List of recommended pedotransfer functions (PTFs) by predicted soil hydraulic property and available predictor variables.

| Predictor variables[1] | Recommended PTFs[2] | | | | | | | | |
|---|---|---|---|---|---|---|---|---|---|
| | THS | FC_2 | FC | WP | AWC_ | AWC | KS | VG | MVG |
| PSD+DEPTH_M | PTF01 | PTF01 | PTF01 | PTF01 | PTF01 | PTF01 | PTF01 | PTF01 | PTF01 |
| PSD+DEPTH_M+OC | PTF02 | PTF02 | PTF02 | PTF02 | PTF02 | PTF01 | PTF02 | PTF02 | PTF02 |
| PSD+DEPTH_M+BD | PTF03 | PTF03 | PTF01 | PTF01 | PTF03 | PTF01 | PTF01 | PTF03 | PTF01 |
| PSD+DEPTH_M+CACO3 | PTF04 | PTF01 | PTF01 | PTF01 | PTF01 | PTF01 | PTF01 | PTF04 | PTF04 |
| PSD+DEPTH_M+PH_H2O | PTF05 | PTF01 | PTF01 | PTF01 | PTF01 | PTF01 | PTF01 | PTF05 | PTF05 |
| PSD+DEPTH_M+CEC | PTF01 | PTF01 | PTF06 | PTF01 | PTF01 | PTF01 | PTF01 | PTF06 | PTF06 |
| PSD+DEPTH_M+OC+BD | PTF03 | PTF03 | PTF02 | PTF02 | PTF03 | PTF03 | PTF02 | PTF07 | PTF02 |
| PSD+DEPTH_M+OC+CACO3 | PTF02 | PTF02 | PTF02 | PTF02 | PTF02 | PTF01 | PTF02 | PTF08 | PTF02 |
| PSD+DEPTH_M+OC+PH_H2O | PTF02 | PTF02 | PTF02 | PTF02 | PTF02 | PTF01 | PTF02 | PTF09 | PTF02 |
| PSD+DEPTH_M+OC+CEC | PTF02 | PTF06 | PTF02 | PTF02 | PTF02 | PTF01 | PTF02 | PTF10 | PTF02 |
| PSD+DEPTH_M+BD+CACO3 | PTF03 | PTF03 | PTF03 | PTF01 | PTF03 | PTF01 | PTF01 | PTF11 | PTF04 |
| PSD+DEPTH_M+BD+PH_H2O | PTF03 | PTF03 | PTF03 | PTF01 | PTF03 | PTF01 | PTF05 | PTF12 | PTF12 |
| PSD+DEPTH_M+BD+CEC | PTF03 | PTF13 | PTF06 | PTF01 | PTF03 | PTF01 | PTF01 | PTF13 | PTF13 |
| PSD+DEPTH_M+CACO3+PH_H2O | PTF05 | PTF05 | PTF04 | PTF01 | PTF01 | PTF01 | PTF05 | PTF14 | PTF04 |
| PSD+DEPTH_M+CACO3+CEC | PTF04 | PTF04 | PTF04 | PTF01 | PTF01 | PTF01 | PTF01 | PTF15 | PTF04 |
| PSD+DEPTH_M+PH_H2O+CEC | PTF05 | PTF05 | PTF06 | PTF01 | PTF01 | PTF01 | PTF01 | PTF16 | PTF06 |
| PSD+DEPTH_M+OC+BD+CACO3 | PTF03 | PTF07 | PTF02 | PTF07 | PTF03 | PTF03 | PTF02 | PTF11 | PTF02 |
| PSD+DEPTH_M+OC+BD+PH_H2O | PTF03 | PTF07 | PTF02 | PTF07 | PTF03 | PTF03 | PTF02 | PTF12 | PTF02 |
| PSD+DEPTH_M+OC+BD+CEC | PTF03 | PTF13 | PTF07 | PTF07 | PTF03 | PTF03 | PTF02 | PTF13 | PTF02 |
| PSD+DEPTH_M+OC+CACO3+PH_H2O | PTF02 | PTF02 | PTF02 | PTF02 | PTF02 | PTF01 | PTF02 | PTF09 | PTF20 |
| PSD+DEPTH_M+OC+CACO3+CEC | PTF02 | PTF08 | PTF04 | PTF08 | PTF02 | PTF01 | PTF02 | PTF21 | PTF21 |
| PSD+DEPTH_M+OC+PH_H2O+CEC | PTF02 | PTF09 | PTF09 | PTF02 | PTF02 | PTF01 | PTF02 | PTF22 | PTF02 |
| PSD+DEPTH_M+BD+CACO3+PH_H2O | PTF03 | PTF03 | PTF03 | PTF01 | PTF03 | PTF01 | PTF05 | PTF11 | PTF23 |
| PSD+DEPTH_M+BD+CACO3+CEC | PTF03 | PTF11 | PTF11 | PTF01 | PTF03 | PTF01 | PTF01 | PTF24 | PTF04 |
| PSD+DEPTH_M+BD+PH_H2O+CEC | PTF03 | PTF12 | PTF11 | PTF01 | PTF03 | PTF01 | PTF05 | PTF25 | PTF12 |
| PSD+DEPTH_M+CACO3+PH_H2O+CEC | PTF05 | PTF05 | PTF14 | PTF05 | PTF01 | PTF01 | PTF05 | PTF15 | PTF04 |
| PSD+DEPTH_M+OC+BD+CACO3+PH_H2O | PTF03 | PTF07 | PTF02 | PTF07 | PTF03 | PTF03 | PTF02 | PTF11 | PTF27 |
| PSD+DEPTH_M+OC+BD+CACO3+CEC | PTF03 | PTF11 | PTF07 | PTF17 | PTF03 | PTF03 | PTF02 | PTF24 | PTF28 |
| PSD+DEPTH_M+OC+BD+PH_H2O+CEC | PTF03 | PTF12 | PTF07 | PTF07 | PTF03 | PTF03 | PTF02 | PTF29 | PTF29 |
| PSD+DEPTH_M+OC+CACO3+PH_H2O+CE | PTF02 | PTF09 | PTF08 | PTF02 | PTF02 | PTF01 | PTF02 | PTF21 | PTF20 |
| PSD+DEPTH_M+BD+CACO3+PH_H2O+CE | PTF03 | PTF11 | PTF06 | PTF05 | PTF03 | PTF01 | PTF05 | PTF24 | PTF12 |
| PSD+DEPTH_M+OC+BD+CACO3+PH_H2O | PTF03 | PTF18 | PTF07 | PTF09 | PTF03 | PTF03 | PTF02 | PTF29 | PTF27 |

[1]PSD: particle size distribution (sand, 50–2000 μm; silt, 2–50 μm; clay, <2 μm (mass %)); DEPTH: mean soil depth (cm); OC: organic carbon content (mass %); BD: bulk density (g cm$^{-3}$); CACO3: calcium carbonate content (mass %); PH_H2O: pH in water (-); CEC: cation exchange capacity (cmol (+) kg$^{-1}$).
[2]THS: saturated water content (pF 0); FC_2: water content at -100 cm matric potential head (pF 2.0); FC: water content at -330 cm matric potential head (pF 2.5); AWC_2: plant available water content based on FC_2; AWC: plant available water content based on FC; WP: water content at wilting point (pF 4.2); KS: saturated hydraulic conductivity; VG: parameters of the van Genuchten model; MVG: parameters of the Mualem – van Genuchten model; TEST_BASIC: samples with measured PSD, DEPTH, OC and BD; TEST_CHEM+: samples with measured PSD, DEPTH, OC, BD, CACO3, PH_H2O and CEC.

## FIGURES

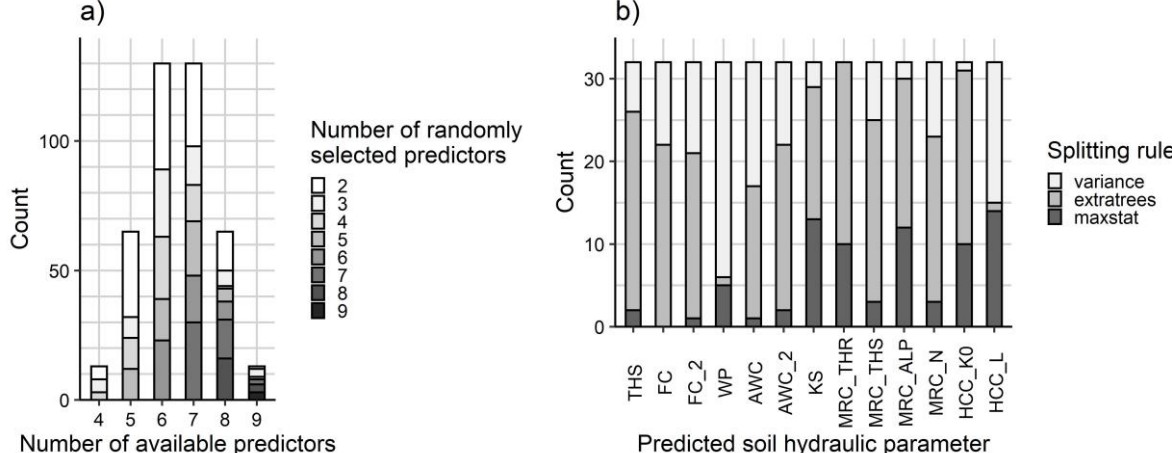

**Figure 1.** Results of parameter tuning of the random forest: optimization of a) the number of randomly selected predictors at each split by number of available predictors and b) splitting rule applied to build the trees in the random forest.

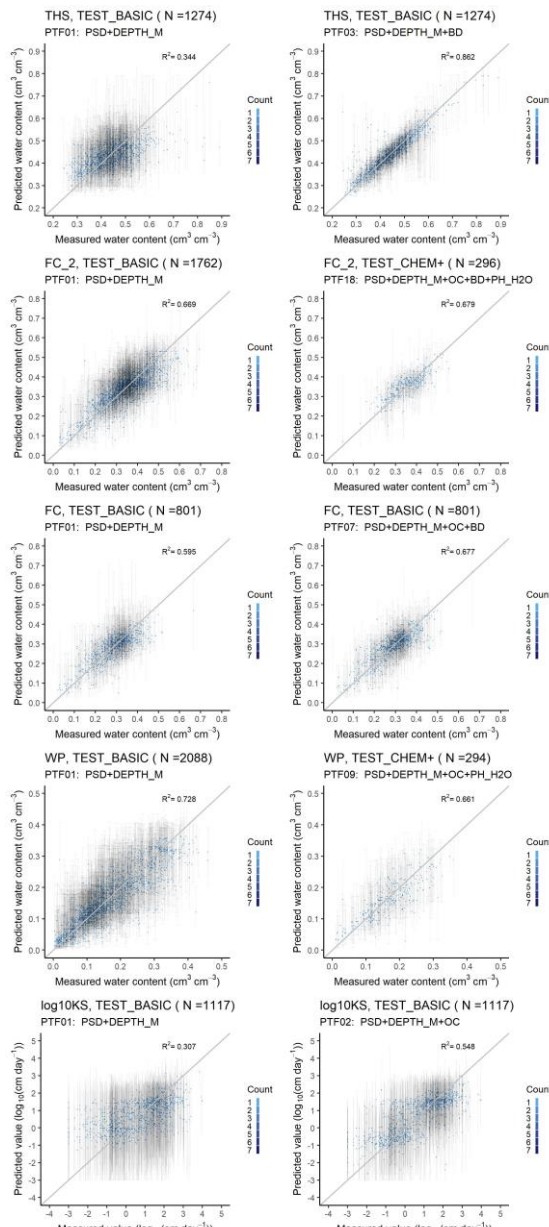

**Figure 2.** Scatter plot of the measured versus median predicted water retention values of the worst and best performing PTF
with 90% prediction interval on test datasets. THS: saturated water content (PTF01 vs. PTF03); FC_2: water content at -100
cm matric potential head (PTF01 vs. PTF18); FC: water content at -330 cm matric potential head (PTF01 vs. PTF07); WP:
water content at wilting point (PTF01 vs. PTF09); log10KS: saturated hydraulic conductivity (PTF01 vs. PTF02); PSD: particle
size distribution (sand, 50–2000 μm; silt, 2–50 μm; clay, <2 μm (mass %)); DEPTH_M: mean soil depth (cm); OC: organic
carbon content (mass %); BD: bulk density (g cm$^{-3}$); PH_H2O: pH in water (-); Count: the number of cases in each rectangle.

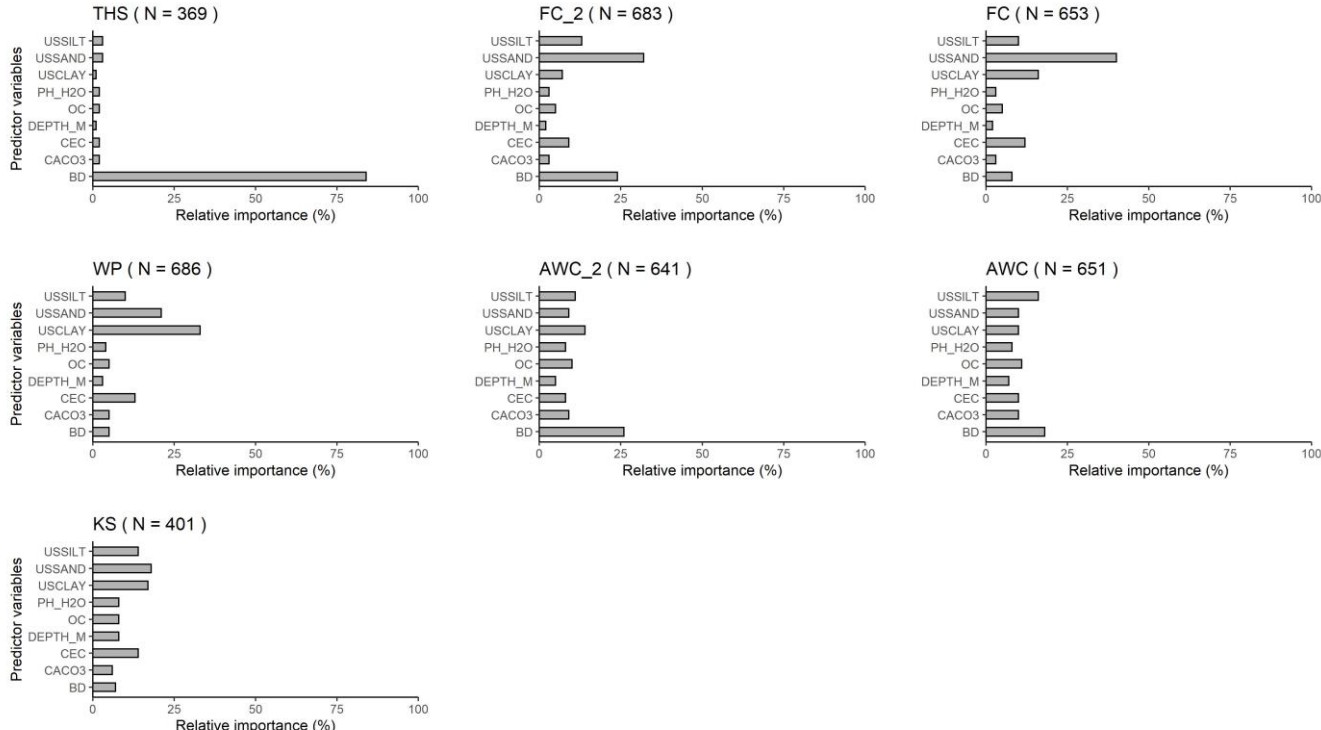

2 **Figure 3.** Relative variable importance computed with the random forest algorithm for the prediction of water content with

3 PTF32 at saturation (THS), at field capacity; -100 (FC_2) and -330 (FC) matric potential head, at wilting point (WP), of the

4 plant available water content based on FC_2 (AWC_2) and FC (AWC), and the saturated hydraulic conductivity (KS).

5 USSILT: silt content (2–50 μm (mass %)); USSAND: sand content (50–2000 μm (mass %)); USCLAY: clay content ( <2 μm

6 (mass %)); PH_H2O: pH in water (-); OC: organic carbon content (mass %); DEPTH_M: mean soil depth (cm); OC: organic

7 carbon content (mass %); CEC: cation exchange capacity (cmol (+) kg$^{-1}$); CACO3: calcium carbonate content (mass %); BD:

8 bulk density (g cm$^{-3}$).

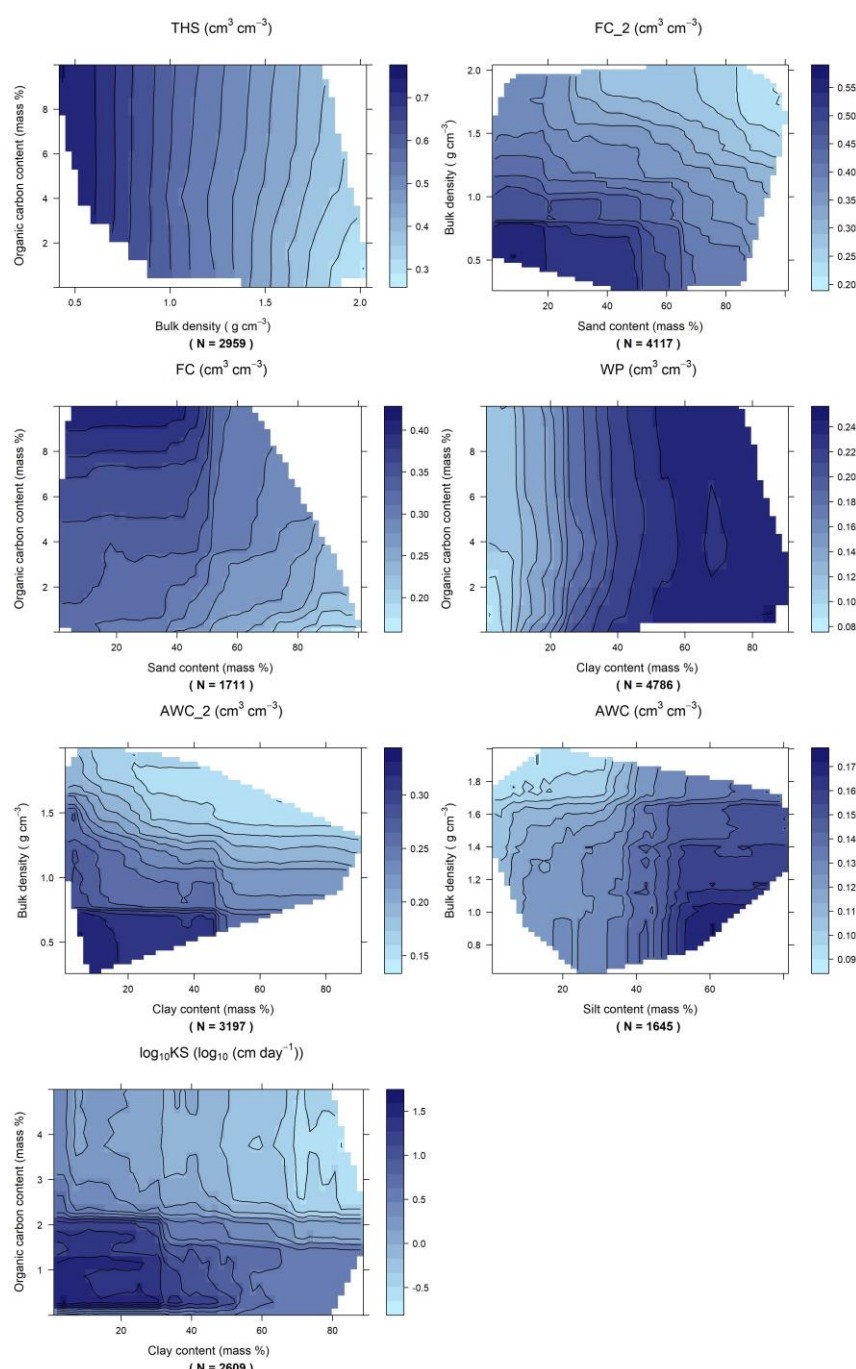

2  **Figure 4.** Partial dependence plot computed based on the random forest algorithm (PTF07) for the prediction of water content

3  at saturation (THS), field capacity at -100 (FC_2) and -330 (FC) matric potential head, wilting point (WP), plant available

4  water content computed with field capacity at -100 and -330 cm matric potential head (AWC_2, AWC) and saturated hydraulic

5  conductivity (KS) for selected predictors.

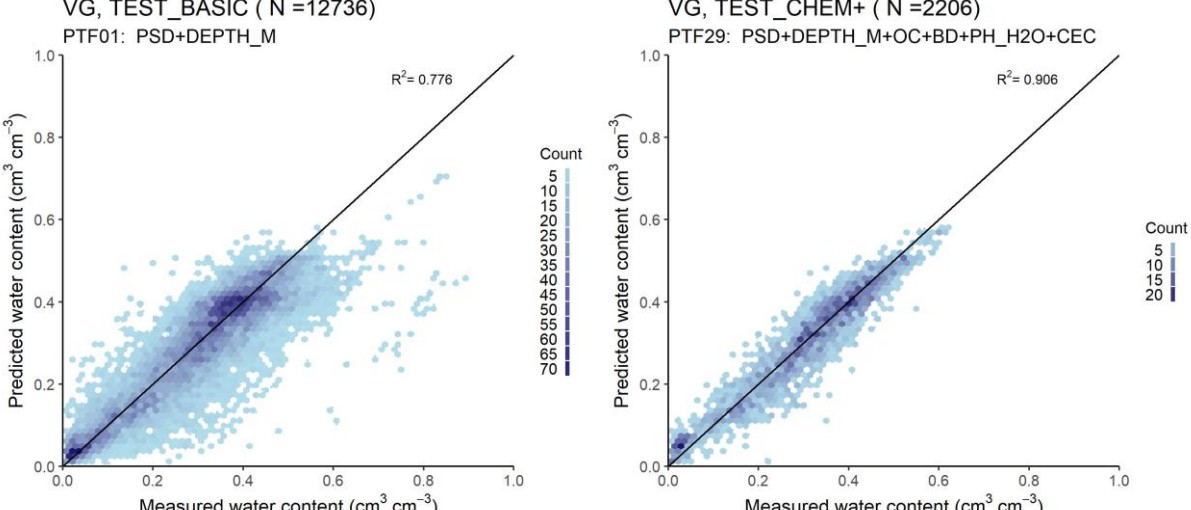

**Figure 5.** Scatter plot of the measured versus median predicted water retention values computed with the van Genuchten (VG) model (PTF01 vs. PTF29, i.e. the worst versus best performing PTF). PSD: particle size distribution (sand, 50–2000 μm; silt, 2–50 μm; clay, <2 μm (mass %)); DEPTH_M: mean soil depth (cm); OC: organic carbon content (mass %); BD: bulk density (g cm$^{-3}$); PH_H2O: pH in water (-); CEC: cation exchange capacity (cmol (+) kg$^{-1}$); Count: the number of cases in each hexagon.

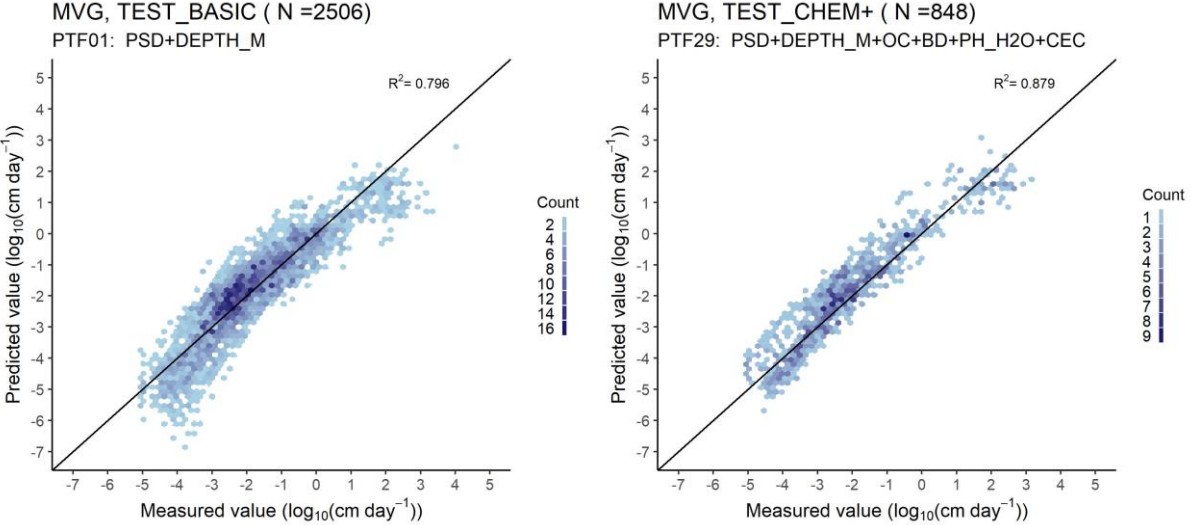

**Figure 6.** Scatter plot of the measured versus median predicted hydraulic conductivity values computed with the Mualem-van Genuchten (MVG) model (PTF01 vs. PTF27, i.e. the worst versus best performing PTF). PSD: particle size distribution (sand, 50–2000 μm; silt, 2–50 μm; clay, <2 μm (mass %)); DEPTH_M: mean soil depth (cm); OC: organic carbon content (mass %); BD: bulk density (g cm$^{-3}$); CACO3: calcium carbonate content (mass %); PH_H2O: pH in water (-); Count: the number of cases in each hexagon.

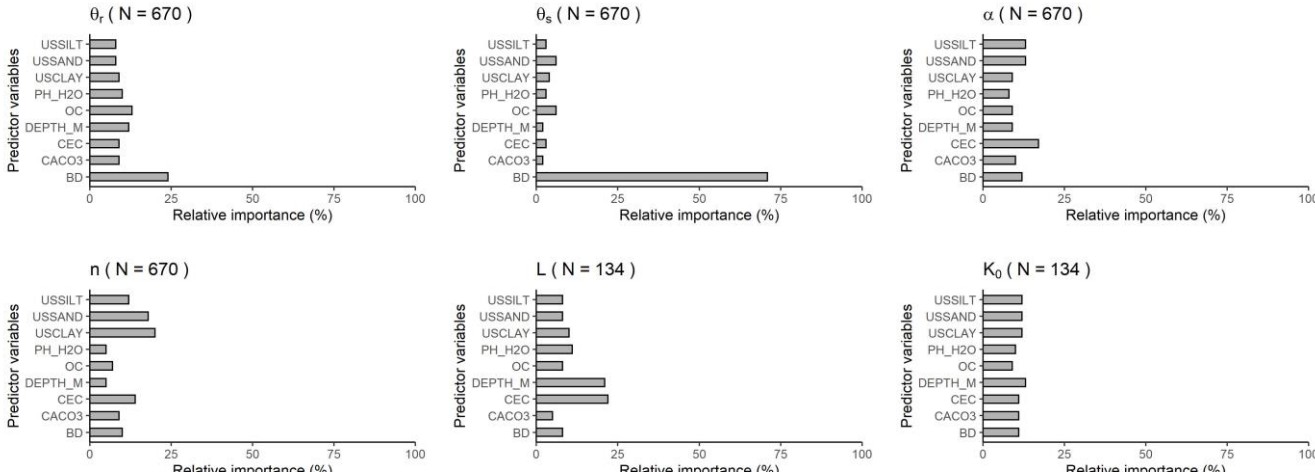

2 **Figure 7.** Relative variable importance computed with the random forest algorithm for the prediction of parameters of the van

3 Genuchten and Mualem-van Genuchten models based on PTF32. $\theta_r$: residual water content (cm³ cm⁻³); $\theta_s$: saturated water

4 content (cm³ cm⁻³); $\alpha$ (cm⁻¹), $n$ (-): shape parameters; $K_0$: the hydraulic conductivity acting as a matching point at saturation

5 (cm day⁻¹); $L$: shape parameter related to pore tortuosity (-); USSILT: silt content (2–50 μm (mass %)); USSAND: sand content

6 (50–2000 μm (mass %)); USCLAY: clay content,( <2 μm (mass %)); PH_H2O: pH in water (-); OC: organic carbon content

7 (mass %); DEPTH_M: mean soil depth (cm); OC: organic carbon content (mass %); CEC: cation exchange capacity (cmol (+)

8 kg⁻¹);> CACO3: calcium carbonate content (mass %); BD: bulk density (g cm⁻³).