# Peer review of "Updated European hydraulic pedotransfer functions with communicated uncertainties in the predicted variables (euptfv2)"

_Geoscientific Model Development, 2020_

## Referee Comment (RC1) · Anonymous Referee #1 · 17 May 2020

interesting paper, well structured and practical impact.

---

## Referee Comment (RC2) · Anonymous Referee #2 · 30 May 2020

Soil pedotransfer functions are important when used for estimation of soil hydraulic parameters in catchment, regional, or continental scale applications. This manuscript improves the estimation of euptfv1 and provides information about prediction uncertainty, and can be applied for more predictor variable combinations than the euptfv1. Overall, the manuscript is interesting, important, well written, and organized in a logical well. Therefore, I recommend accepting this manuscript after minor revisions that are required to address the general and specific comments provided below.

General comments:

1. The authors compared the estimation of water content at saturation, field capacity,

wilting point, plant available water content, saturated hydraulic conductivity, etc., individually. I think these sections are somewhat lengthy. However, the most interesting part of the comparisons between point and parameter predictions and euptfv1 and v2 are very short. Is it possible to extend the comparisons and the discussion?

2. The authors listed so many PTFs. When I was reading the conclusion part, I can not find which PTF I should use. Is it possible to make some concluding remarks regarding which PTFs should be used for corresponding predictors? I think this will be very helpful for future readers.

Specific comments:

Additional minor comments:

1. Figures 2, 5, and 6: Is it possible to include R2 in these figures? This will make the comparison between different figures easier.

2. In the abstract and conclusion sections: -15.000 should be -15,000

3. Page 6, line 4: why did the authors utilize median values instead of mean values?

4. Page 7, line 19: "in the study of (Khodaverdiloo et al., 2011)" should be "in the study of Khodaverdiloo et al. (2011)"

5. Page 10, line 4: "and RMSE" should be "an RMSE"

6. Page 10, Line 27: ";" should be ","

7. Page 12, line 14: add a connection/linking word before "it is due to"
* * *

---

## Referee Comment (RC3) · Anonymous Referee #3 · 12 Jun 2020

This manuscript aims to update the previously developed PTF for European soils called euptfv1. More importantly, euptfv2 contributes to the understudied issue of uncertainty in PTFs for potential users. Despite the existing large amount of results, the paper is easy to follow with some possibilities to improve. The authors also provide a detailed and user-friendly website from euptfv2, however, no library called eutptf exists in R Repository, even the available zip file has problems to be run. Many comparisons among the possibilities of PTFs for different soil hydraulic properties were done. These series of "euptfv(i)" will contribute to the modelling of soil processes. I recommend this paper for publication, however, I outlined some questions and comments as below (L denotes line and P for page)

[Figure]

L30, P1. variably saturated fluxes? do you mean flow through variably saturated soil media?

L31. P.2. Not necessary to machine learning-based methods are able to calculate uncertainty because the sampling effect can propagate parameter uncertainty, which can be implemented even in simple regression-based models. Tens of resamples for training and testing with different distributions can be drawn from the population (Tranter et al., 2010; Kotlar et al., 2019).

Do train and test datasets in bootstraps follow the same distributions?

Tranter, G., Minasny, B. and McBratney, A.B., 2010. Estimating pedotransfer function prediction limits using fuzzy k-means with extragrades. Soil Science Society of America Journal, 74(6), pp.1967-1975.

Kotlar, A.M., de Jong van Lier, Q., Barros, A.H.C., Iversen, B.V. and Vereecken, H., 2019. Development and Uncertainty Assessment of Pedotransfer Functions for Predicting Water Contents at Specific Pressure Heads. Vadose Zone Journal, 18(1).

Table1. P.18. Numbers are not aligned exactly below the names.

Correlation matrix of observations would be useful information (in appendix) at least for the dataset used for the best PTFs.

L6, P5: Please calculate variable importance of parameters in PTFs as relative which makes summation of all 100%. (e.g. Figure 3)

L3, P7: To give a better view of the performance of PTFs, compare the mean values of measured parameters with RMSE of predictions. Compared to Toth et al., (2015), improvement in the prediction of THS is less than FC and WP, why?

Figure S2, please replace SE by RMSE so the reader doesn't lose the track of comparison criteria.

L1, P8. Please mention the correlation between THS and BD, lets arguably consider

[Figure]

THS equal to total porosity, does the 1-BD/PD, assuming PD=2.65 give better RMSE than PTF03 for THS? or you might easily obtain the best PD to predict THS by this formula. In PTF 32, the relative importance of BD is almost 100%.

L31, P10. Elaborate the range of Ks values used in training for PTF02, so reader can judge how low is RMSE of 0.94.

L1-8, P10. You can compare the randomized RMSE by PTF02 (RMSE/(maxKs-minKs)) by some studies in the literature (preferably Europe or at least temperate soils)

L19, P10. I expect to see the high importance of clay in THETAr. It is not clear exactly how to estimate VG and MVG parameters.

L23, P10. K0, matching point should be defined earlier.

L25-30, P11. How many of K data are obtained from evaporation method, this method usually goes up to -1000cm, is it why overestimation occurs in Fig S21 in drier conditions or another reason? Note that in this dry region K data is obviously small and mean error of about 0.8 is significant.

Moreover, comparing Fig s21 with Fig S1b (Toth et al., 2015), there less error in this dry region was observed.

Fig2, 5. Explain the term "count" in legend

Table 7. RMSE is log10(cm/d) but this belongs to retention curve.

Table 8. this RMSE was computed only by K(h) data? Did you consider Lambda=0.5?

L5, P 12. That's interesting to show Comparison of point and parameter predictions, however, you should emphasize that this works only when water retention curve matters. Because one can use the n value of WRC and l=0.5 for K function.

During some trials to run the package, I have faced with various errors such as

Error in source_data

("https://github.com/TothSzaboBrigitta/ euptfv2/blob/master/suggested_PTFs/ FC_EUHYDI/FC_PTF07.rdata?raw=True") : could not find function "source_data"

please check the files again in the attached zip files. I could not also find neither euptf1 nor 2 in CRAN repository.

---

## Author Comment (AC1) · 27 Jul 2020

Response to Anonymous Referee #2

Thank you for the review and constructive comments. We will address the comments in a revised version of the article. Below we give details on exactly how we address the concerns raised by anonymous referee 2. Please note the following during reading the responses: - the supplement file includes the structured and formatted version of the responses, - the responses are in blue regular font and follow the referee's questions (RC2), - new text parts that will be added to the manuscript are in blue italic font, - the reference to the lines and pages relates to the discussion paper available from:

https://gmd.copernicus.org/preprints/gmd-2020-36/gmd-2020-36.pdf .

RC2: Soil pedotransfer functions are important when used for estimation of soil hydraulic parameters in catchment, regional, or continental scale applications. This manuscript improves the estimation of euptfv1 and provides information about prediction uncertainty, and can be applied for more predictor variable combinations than the euptfv1. Overall, the manuscript is interesting, important, well written, and organized in a logical well. Therefore, I recommend accepting this manuscript after minor revisions that are required to address the general and specific comments provided below.

A: Thank you for the positive general comment.

RC2: 1. The authors compared the estimation of water content at saturation, field capacity, wilting point, plant available water content, saturated hydraulic conductivity, etc., individually. I think these sections are somewhat lengthy. However, the most interesting part of the comparisons between point and parameter predictions and euptfv1 and v2 are very short. Is it possible to extend the comparisons and the discussion?

AGREED.

A1: Regarding comparison between point and parameter predictions, we will be more specific by adding the following:

P12 L5: " . . . more accurate and for further 8 cases RMSE were smaller."

P12 L6: "The reason for higher RMSE in parameter estimation can be that the VG model does not always adequately describe the measured MRC data (Weber et al., 2019). Therefore, when THS, FC, FC_2 and WP are computed with parameter estimation those are not only affected by the uncertainty of the prediction of VG parameters but by the goodness of VG model fit as well."

P12 L8: will rephrase the sentence to make it clearer, which now reads: "For THS point estimation performed better than parameter estimation. When the moisture retention curve is not needed, but only THS and/or FC/FC_2 and/or WP, we recommend to

compute those with . . ."

In order to include the suggested comparison, between euptfv1 and v2, we will include the following sentences

P12 L12: "The most important reason for it can be that the interaction between the target variable and the predictors is more complex for the cases of predicting FC or VG parameters – to describe the MRC –, which can be untangled using random forest. This may provide a reasons the random forest algorithm performed significantly better than the PTFs derived with linear regression or a simple regression tree."

P12 L14: "The RMSE of THS prediction was somewhat lower for euptfv1 than for euptfv2, but the difference was not significant. It could be due to the close to linear relationship between THS and BD and high relative importance of BD in THS prediction (84 %). This way their interaction can be efficiently described with the linear regression which is capable to extrapolate as well. Extrapolation with the random forest algorithm is not possible outside the training data, which can limit its performance. The general improvement of the PTFs in euptfv2 is threefold, it is due to i) using random forest instead of single regression tree or linear regression, ii) including more detailed information on soil sampling depth, not only distinguishing topsoils and subsoils and iii) providing information on prediction uncertainty.

Regarding the description of the individual point and parameter estimations we will keep the details because we think it is instructive to provide information about the importance of specific predictors.

RC2: 2. The authors listed so many PTFs. When I was reading the conclusion part, I cannot find which PTF I should use. Is it possible to make some concluding remarks regarding which PTFs should be used for corresponding predictors? I think this will be very helpful for future readers.

AGREED.

A2: Thank you very much for this very helpful comment. Indeed, it is very important that users should easily understand which PTF to select and apply. To achieve this, 1.) we will add a dedicated paragraph on it above the Conclusion section, 2.) will highlight in the abstract and short summary that this section is provided, 3) we will move Table S3 from supplementary material to the manuscript as Table 11.

The new paragraph 4 reads:

"4. Practical guidance on how to use the PTFs The minimum input requirements for all PTFs are sand, silt and clay content, and soil depth. Soil depth needs to be considered in regard to the depth of the other input properties and soil hydraulic data needs, e.g. if the soil hydraulic properties of the top 20 cm (0-20 cm) is needed, then depth needs to be set at 10 cm in the input data of the prediction. If only soil texture information is available for the predictions, the class PTFs from euptfv1 could be applied (Tóth et al., 2015). We emphasise that: 1. the units of input soil properties (predictors) have to be the same as indicated in the text and that the sand, silt, and clay are defined by the following particle diameters: clay < 2 $\mu$m, silt between 2 and 50 $\mu$m, and sand between 50 and 2000 $\mu$m, 2. when only specific water content values at saturation, field capacity or wilting point are required (ie. THS, FC_2, FC, WP) it is recommended to use point PTFs. This is also true for the prediction of KS, 3. for AWC, the most accurate way is by first predicting FC and WP with the point predictions and then compute AWC using Eq. (1), and similarly for AWC_2 using FC_2 and Eq. (2), 4. it is recommended to do the VG prediction if only moisture retention curve parameters are needed, and 5. the MVG prediction when both moisture retention and hydraulic conductivity parameters are required. The VG algorithms predict the following van Genuchten model parameters: the residual water content ÏŚr (cm$^3$ cm-3), the saturated water content ÏŚs (cm$^3$ cm-3), and shape parameters $\alpha$ (cm-1) and n (-). Parameter m is provided based on m=1-1/n (van Genuchten, 1980), and for the hydraulic conductivity curve, the two additional parameters: K0 (cm day-1) the hydraulic conductivity acting as a matching point at saturation and L, the shape parameter related to pore tortuosity (-).

Table 11 shows the recommended PTFs for each predicted soil hydraulic property and available predictor variables. The users need to check which basic soil properties are available for the predictions, then look in Table 11 which PTF is recommended to use. The algorithms have been implemented in a web interface to facilitate the use of the PTFs, where the PTFs' selection is automated based on soil properties available for the predictions and required soil hydraulic property. The Code and data availability section provides information on how to access this resource."

The additional text in the short summary and the abstract is given by:

Short summary: "... The influence of predictor variables on predicted soil hydraulic properties is explored and practical guidance on how to use the derived PTFs is provided. ..."

Abstract: "... for the prediction of water content at -100 cm matric potential head and plant available water content. A practical guidance on how to use the derived PTFs is provided."

SPECIFIC COMMENTS:

RC2: 1. Figures 2, 5, and 6: Is it possible to include R2 in these figures? This will make the comparison between different figures easier.

AGREED

A1: We will add R2 to Figures 2, 5, 6 and S1, e.g.: Fig_1_response.

RC2: 2. In the abstract and conclusion sections: -15.000 should be -15,000

AGREED

A2: Thank you for noting it, we will correct it in the entire text.

RC2: 3. Page 6, line 4: why did the authors utilize median values instead of mean values? Nothing changed.

A3: Our aim was to provide information about the uncertainty of the predictions, therefore we applied quantile regression forests. This way the most probable predicted response value is at the 50th percentile, i.e. the median, which is considered more robust against the outliers than the mean. In this way we decided to use the median as the predicted value (yhat) rather than the mean.

RC2: 4. Page 7, line 19: "in the study of (Khodaverdiloo et al., 2011)" should be "in the study of Khodaverdiloo et al. (2011)"

AGREED

A4: Thank you, we will correct it.

RC2: 5. Page 10, line 4: "and RMSE" should be "an RMSE"

AGREED

A5: Thank you, we will correct it.

RC2: 6. Page 10, Line 27: ";" should be ","

AGREED

A6: Thank you, we will correct it.

RC2: 7. Page 12, line 14: add a connection/linking word before "it is due to"

AGREED

A7: The text will be rephrased to: "The improvement of the PTFs is twofold, the better performance is due to . . .".

Please also note the supplement to this comment:
https://gmd.copernicus.org/preprints/gmd-2020-36/gmd-2020-36-AC1-supplement.pdf

―――――――――――――――――

[Figure]

**Fig. 1.** Fig_1_response

---

## Author Comment (AC2) · 27 Jul 2020

Response to Anonymous Referee #3

Thank you for the detailed review and suggestions for further improvements. In the following, we give a detailed presentation of how we will address all the questions and issues raised. Below we would like to answer and provide possible solutions for the comments and recommendations, following the referee's questions (RC3). Please note the following during reading the responses: - the supplement file includes the structured and formatted version of the responses: https://gmd.copernicus.org/preprints/gmd-2020-36/gmd-2020-36-AC2-supplement.pdf, - the responses are in blue regular font

and inserted under the referee's questions, - new text parts that will be added to the manuscript are in blue italic font, - the reference to the lines and pages relates to the discussion paper available from: https://gmd.copernicus.org/preprints/gmd-2020-36/gmd-2020-36.pdf .

RC3: This manuscript aims to update the previously developed PTF for European soils called euptfv1. More importantly, euptfv2 contributes to the understudied issue of uncertainty in PTFs for potential users. Despite the existing large amount of results, the paper is easy to follow with some possibilities to improve.

A: Thank you for the positive general comment.

RC3: The authors also provide a detailed and user-friendly website from euptfv2, however, no library called eutptf exists in R Repository, even the available zip file has problems to be run.

A: The R package of euptfv2 is under construction. The available zip files include the R scripts used to develop the predictions and the derived pedotransfer functions. The dataset which we used for training and testing the algorithms cannot be shared according to the agreement between the data holders. Regarding the model development the following information is included separately for point and parameter estimations: i) loading data, define path, input variables and function to compute performance of the PTFs (setupRF.R), ii) parameter tuning of the random forest (tuneRF.R), iii) building final random forest (buildfinalRF.R), iv) compute performance of the final random forest on the test set (testRF.R) . In a separate folder (https://github.com/TothSzaboBrigitta/euptfv2/tree/master/help) a sample input dataset (data_sample.csv) and an R script (apply_PTFs_script.R) - which shows some examples on how to apply the PTFs in R – have been added to the repository.

RC3: Many comparisons among the possibilities of PTFs for different soil hydraulic properties were done. These series of "euptfv(i)" will contribute to the modelling of soil processes. I recommend this paper for publication, however, I outlined some questions

and comments as below (L denotes line and P for page)

A: Thank you for considering the usability of the euptfs.

RC3: L30, P1. variably saturated fluxes? do you mean flow through variably saturated soil media?

AGREED

A: Yes, thank you for noting it, we will correct it: "Simulations of flow through variably saturated soil media either rely on . . ."

RC3: L31. P.2. Not necessary to machine learning-based methods are able to calculate uncertainty because the sampling effect can propagate parameter uncertainty, which can be implemented even in simple regression-based models. Tens of resamples for training and testing with different distributions can be drawn from the population (Tranter etal., 2010; Kotlar et al., 2019). Do train and test datasets in bootstraps follow the same distributions? Tranter, G., Minasny, B. and McBratney, A.B., 2010. Estimating pedotransfer function prediction limits using fuzzy k-means with extragrades. Soil Science Society of AmericaJournal, 74(6), pp.1967-1975. Kotlar, A.M., de Jong van Lier, Q., Barros, A.H.C., Iversen, B.V. and Vereecken, H.,2019. Development and Uncertainty Assessment of Pedotransfer Functions for Predicting Water Contents at Specific Pressure Heads. Vadose Zone Journal, 18(1).

AGREED

A: Thank you to highlight it with references, we will add this information to P2 L16:

"Tranter et al. (2010) developed an uncertainty estimation method using fuzzy k-means with extragrades classification that can be applied in any PTF prediction. Kotlar et al. (2019) presented uncertainty assessment of PTFs through deriving PTFs on tens of resamples for train and test sets."

and we will add the following to P2 L33:

"If PTFs are derived with these algorithms, the uncertainty of the predicted soil property can be directly estimated when applying the PTF (Szabó et al., 2019a), although this could also be achieved by applying the above mentioned uncertainty assessment methods without using machine learning methods (e.g. Kotlar et al., 2019; Tranter et al., 2010)."

Using Kolmogorov–Smirnov tests, we tested whether training and test sets have the same frequency distributions, please find the results in Table 1 (please find readable version here: https://gmd.copernicus.org/preprints/gmd-2020-36/gmd-2020-36-AC2-supplement.pdf). For THS, FC and WP the distribution of training and TEST_BASIC set is equal in almost all the cases of the most important basic soil properties. For KS, the distribution of sand and organic carbon content is equal in the training and TEST_BASIC set, in case of FC_2 only the distribution of sand content is equal based on the statistical test. The distributions of training and TEST_CHEM+ sets are equal only in case of FC. For the other sets, at least one soil property has equal distribution in the two sets.

Table 1. Results of the Kolmogorov–Smirnov test (p value of 0.05) computed to compare distribution of the most important basic soil properties of training and test datasets. Soil hydraulic property Input variable p-value of Kolgomorov-Smirnov test Training vs. TEST_BASIC set Training vs. TEST_CHEM+ set THS USSAND 0,137 0,000 USCLAY 0,022 0,000 OC 0,598 0,004 BD 0,483 0,021 FC_2 USSAND 0,616 0,112 USCLAY 0,004 0,000 OC 0,018 0,000 BD 0,023 0,000 FC USSAND 0,019 0,157 USCLAY 0,172 0,078 OC 0,662 0,737 BD 0,313 0,489 WP USSAND 0,730 0,007 USCLAY 0,372 0,003 OC 0,649 0,000 BD 0,047 0,074 KS USSAND 0,396 0,000 USCLAY 0,001 0,008 OC 0,755 0,001 BD 0,000 0,000

Train and test datasets in bootstraps are divided in the following way: in the random forest algorithm for each tree 63% of the data is selected with replacement to build the tree, i.e. number of selected data will be increased to reach the number of samples of the training set with the replacement, this way some samples will be used multiple

times in a single tree. Each tree of the forest is trained on different samples. The forest includes 200 trees and the predicted value is the median of all 200 trees. However, it is difficult to compute the Kolgomorov-Smirnov test for all the 200 in-bag and out-of-bag samples by each predicted soil hydraulic properties, we could confirm based on the literature (Hastie et al., 2009), that the forest will neither be biased nor overfitted to the data because of the two step randomization – bagging process and split-variable randomization – implemented in the algorithm.

RC3: Table1. P.18. Numbers are not aligned exactly below the names. Correlation matrix of observations would be useful information (in appendix) at least for the dataset used for the best PTFs.

AGREED

A: The columns' names will be aligned with the numbers below. The correlation plots of the best PTFs are inserted below the answers (Fig_responses_1 – Fig_responses_7), however descriptive power of them are limited because the relationship between predicted parameters and predictors are not linear. This is the reason why PTFs are derived with a machine learning algorithm and partial dependence plots are shown in the manuscript. We feel that the correlation plots might not provide indispensable information. Please find Fig_responses_1 - Fig_responses_7 below the text.

RC3: L6, P5: Please calculate variable importance of parameters in PTFs as relative which makes summation of all 100%. (e.g. Figure 3)

AGREED

A: Please find here the relative importance plots, with which we will replace Figure 3 and 7 and specify that relative variable importance is shown. Please find Figure 3 and 7 below the text.

Figure 3. Relative variable importance computed with the random forest algorithm for the prediction of water content with PTF32 at saturation (THS), at field capacity;

-100 (FC_2) and -330 (FC) matric potential head, at wilting point (WP), of the plant available water content based on FC_2 (AWC_2) and FC (AWC), and the saturated hydraulic conductivity (KS). USSILT: silt content (2–50 $\mu$m (mass %)); USSAND: sand content (50–2000 $\mu$m (mass %)); USCLAY: clay content ( <2 $\mu$m (mass %)); PH_H2O: pH in water (-); OC: organic carbon content (mass %); DEPTH_M: mean soil depth (cm); OC: organic carbon content (mass %); CEC: cation exchange capacity (cmol (+) kg$-1$); CACO3: calcium carbonate content (mass %); BD: bulk density (g cm$-3$).

Figure 7. Relative variable importance computed with the random forest algorithm for the prediction of parameters of the van Genuchten and Mualem-van Genuchten models based on PTF32. $\theta$r: residual water content (cm$^3$ cm-3); $\theta$s: saturated water content (cm$^3$ cm-3); $\alpha$ (cm-1), n (-): fitting parameters; K0: the hydraulic conductivity acting as a matching point at saturation (cm day-1); L: shape parameter related to pore tortuosity (-); USSILT: silt content (2–50 $\mu$m (mass %)); USSAND: sand content (50–2000 $\mu$m (mass %)); USCLAY: clay content,( <2 $\mu$m (mass %)); PH_H2O: pH in water (-); OC: organic carbon content (mass %); DEPTH_M: mean soil depth (cm); OC: organic carbon content (mass %); CEC: cation exchange capacity (cmol (+) kg$-1$);> CACO3: calcium carbonate content (mass %); BD: bulk density (g cm$-3$).

We will add the following text in P5 L30:

"The relative importance was assessed by dividing the variable importance of each predictor by the sum of the importance of all the predictors after Kotlar et al. (2019)."

RC3: L3, P7: To give a better view of the performance of PTFs, compare the mean values of measured parameters with RMSE of predictions. Compared to Toth et al., (2015), improvement in the prediction of THS is less than FC and WP, why?

AGREED

A: Thank you for the suggestion. We will add the normalized RMSE (RMSE/(ymax-ymin)), which was also suggested by the reviewer under "L1-8, P10".

The following texts will be added: P6 L10: "The different data range of the dataset influences the performance of the PTFs when that is compared to the studies in the literature. Therefore, normalized RMSE (NRMSE) was computed (Eq. 5.), where $y\_max$ and $y\_min$ are the maximum and minimum value of variable . $NRMSE=RMSE/(y\_max-y\_min)$ (5)"

and in P7 L6 we add "Table S3 shows the NRMSE for the point predictions computed for the TEST_BASIC and TEST_CHEM+ sets to provide possibility for comparison with other PTFs available from the literature."

And the following table (please find readable version here: https://gmd.copernicus.org/preprints/gmd-2020-36/gmd-2020-36-AC2-supplement.pdf) will be included in the supplementary material as Table S3 (The original TableS3 of the supplementary material available from https://gmd.copernicus.org/preprints/gmd-2020-36/gmd-2020-36-supplement.pdf will be moved to the manuscript as Table 11).

Table S3. Normalized root mean squared error (NRMSE) of the point predictions by soil hydraulic properties computed on the test datasets in cm3 cm-3 for water retention and log10 (cm day-1) for saturated hydraulic conductivity. In case of PTF01, 02, 03 and 07 TEST_BASIC set was used for the analysis, for the rest of the PTFs TEST_CHEM+ set was considered.

| Name of PTF in euptfv2 | Predictor variables1 | THS | FC_2 | FC | WP | AWC_2 | AWC | KS |
|---|---|---|---|---|---|---|---|---|
| PTF01 | PSD+DEPTH_M | 0.104 | 0.090 | 0.082 | 0.105 | 0.126 | 0.140 | 0.17 |
| PTF02 | PSD+DEPTH_M+OC | 0.086 | 0.083 | 0.076 | 0.102 | 0.112 | 0.132 | 0.14 |
| PTF03 | PSD+DEPTH_M+BD | 0.048 | 0.079 | 0.074 | 0.100 | 0.111 | 0.132 | 0.17 |
| PTF04 | PSD+DEPTH_M+CACO3 | 0.191 | 0.107 | 0.113 | 0.122 | 0.164 | 0.145 | 0.19 |
| PTF05 | PSD+DEPTH_M+PH_H2O | 0.176 | 0.112 | 0.114 | 0.126 | 0.164 | 0.142 | 0.19 |
| PTF06 | PSD+DEPTH_M+CEC | 0.191 | 0.107 | 0.107 | 0.118 | 0.181 | 0.156 | 0.19 |
| PTF07 | PSD+DEPTH_M+OC+BD | 0.047 | 0.075 | 0.073 | 0.097 | 0.107 | 0.127 | 0.14 |
| PTF08 | PSD+DEPTH_M+OC+CACO3 | 0.184 | 0.097 | 0.109 | 0.117 | 0.160 | 0.143 | 0.19 |
| PTF09 | PSD+DEPTH_M+OC+PH_H2O | 0.167 | 0.095 | 0.107 | 0.119 | 0.158 | 0.141 | |

0.18 PTF10 PSD+DEPTH_M+OC+CEC 0.172 0.098 0.108 0.116 0.158 0.150 0.18 PTF11 PSD+DEPTH_M+BD+CACO3 0.072 0.091 0.105 0.115 0.144 0.140 0.19 PTF12 PSD+DEPTH_M+BD+PH_H2O 0.069 0.086 0.103 0.117 0.143 0.137 0.19 PTF13 PSD+DEPTH_M+BD+CEC 0.070 0.091 0.100 0.115 0.144 0.142 0.19 PTF14 PSD+DEPTH_M+CACO3+PH_H2O 0.168 0.101 0.109 0.121 0.157 0.139 0.18 PTF15 PSD+DEPTH_M+CACO3+CEC 0.179 0.102 0.106 0.113 0.155 0.144 0.19 PTF16 PSD+DEPTH_M+PH_H2O+CEC 0.183 0.098 0.104 0.115 0.152 0.142 0.19 PTF17 PSD+DEPTH_M+OC+BD+CACO3 0.070 0.089 0.102 0.111 0.145 0.139 0.18 PTF18 PSD+DEPTH_M+OC+BD+PH_H2O 0.070 0.083 0.103 0.116 0.143 0.136 0.18 PTF19 PSD+DEPTH_M+OC+BD+CEC 0.070 0.087 0.099 0.113 0.139 0.143 0.18 PTF20 PSD+DEPTH_M+OC+CACO3+PH_H2O 0.166 0.105 0.107 0.114 0.154 0.137 0.18 PTF21 PSD+DEPTH_M+OC+CACO3+CEC 0.171 0.090 0.104 0.108 0.149 0.142 0.18 PTF22 PSD+DEPTH_M+OC+PH_H2O+CEC 0.166 0.089 0.102 0.111 0.148 0.140 0.18 PTF23 PSD+DEPTH_M+BD+CACO3+PH_H2O 0.071 0.089 0.104 0.116 0.147 0.139 0.18 PTF24 PSD+DEPTH_M+BD+CACO3+CEC 0.071 0.085 0.099 0.110 0.138 0.139 0.19 PTF25 PSD+DEPTH_M+BD+PH_H2O+CEC 0.067 0.084 0.100 0.112 0.137 0.135 0.19 PTF26 PSD+DEPTH_M+CACO3+PH_H2O+CEC 0.163 0.094 0.103 0.111 0.145 0.140 0.18 PTF27 PSD+DEPTH_M+OC+BD+CACO3+PH_H2O 0.072 0.086 0.101 0.111 0.148 0.135 0.18 PTF28 PSD+DEPTH_M+OC+BD+CACO3+CEC 0.070 0.082 0.098 0.106 0.136 0.138 0.18 PTF29 PSD+DEPTH_M+OC+BD+PH_H2O+CEC 0.068 0.083 0.095 0.109 0.135 0.134 0.18 PTF30 PSD+DEPTH_M+OC+CACO3+PH_H2O+CEC 0.162 0.100 0.101 0.108 0.145 0.138 0.17 PTF31 PSD+DEPTH_M+BD+CACO3+PH_H2O+CEC 0.070 0.081 0.097 0.108 0.134 0.137 0.18 PTF32 PSD+DEPTH_M+OC+BD+CACO3+PH_H2O+CEC 0.069 0.079 0.097 0.107 0.135 0.135 0.18 1PSD: particle size distribution (sand, 50–2000 $\mu$m; silt, 2–50 $\mu$m; clay, <2 $\mu$m (mass %)); DEPTH: mean soil depth (cm); OC: organic carbon content (mass %); BD: bulk density (g cm$-3$); CACO3: calcium carbonate content (mass %); PH_H2O: pH in water (-); CEC: cation exchange capacity (cmol (+) kg$-1$). 2THS: saturated water content (pF 0); FC_2: water content at -100

cm matric potential head (pF 2.0); FC: water content at -330 cm matric potential head (pF 2.5); AWC_2: plant available water content based on FC_2; AWC: plant available water content based on FC; WP: water content at wilting point (pF 4.2); KS: saturated hydraulic conductivity;

Comparison to Toth et al. (2015): thank you for the reviewer's comment on THS and BD, which helps to clarify findings related to comparison of euptfv1 and v2. There was no significant difference between euptfv1 and v2 in case of THS when BD was available for the prediction and euptfv1 was derived with linear regression. The reason for it – which was mentioned by the reviewer as well – that the relative importance of BD is 84% in the prediction of THS and the relationship between THS and BD is close to linear. In this case random forest could not significantly improve the prediction. In case of FC and WP the interaction between the target variable and the predictors is more complex, this way the random forest algorithm performed significantly better than the PTFs derived with linear regression or a simple regression tree. We will add the following information in P12 L12:

"The most important reason for it can be that the interaction between the target variable and the predictors is more complex for the cases of predicting FC or VG parameters – to describe the MRC, which can be untangled using random forest. This may provide a reason the random forest algorithm performed significantly better than the PTFs derived with linear regression or a simple regression tree. For THS, WP, KS, and MVG only those PTFs did not improve significantly, for which comparisons on the TEST_CHEM+ set were possible – which includes reduced number of samples. The RMSE of THS prediction was somewhat lower for euptfv1 than for euptfv2, but the difference was not significant. It could be due to the close to linear relationship between THS and BD and high relative importance of BD in THS prediction (84 %). This way their interaction can be efficiently described with the linear regression which is capable to extrapolate as well. Extrapolation with the random forest algorithm is not possible, which can limit its performance."

RC3: Figure S2, please replace SE by RMSE so the reader doesn't lose the track of comparison criteria.

AGREED

A: We will replace Figure S2, S4, S6, S8, S10, S12, S14, S16, S19 showing SE with the one showing RMSE, e.g.: Figure S2, which is inserted below the text.

Figure S2. Root mean squared error (RMSE) of the pedotransfer functions derived to predict water content at saturation (THS) computed on TEST_BASIC (N=1274) and TEST_CHEM+ set (N=156). USSAND: sand (50–2000 $\mu$m) content (mass %); US-SILT: silt (2–50 $\mu$m) content (mass %), USCLAY: clay (<2 $\mu$m) content (mass %); DEPTH_M: mean soil depth (cm); OC: organic carbon content (mass %); BD: bulk density (g cm$-3$); CACO3: calcium carbonate content (mass %); PH_H2O: pH in water (-); CEC: cation exchange capacity (cmol (+) kg$-1$).

RC3: L1, P8. Please mention the correlation between THS and BD, lets arguably consider THS equal to total porosity, does the 1-BD/PD, assuming PD=2.65 give better RMSE than PTF03 for THS? or you might easily obtain the best PD to predict THS by this formula. In PTF 32, the relative importance of BD is almost 100%.

DISAGREE

A: Thank you for this idea, however, to remain consistent in methodology and make use of the better performing PTF based on the random forest. The reason: the correlation between THS and BD is -0.92. We have computed the porosity on the test dataset of PTF03 (N = 1274) based on BD and PD (=2.65 g/cm3), then the RMSE of it. We found that the RMSE of PTF03 is smaller than that of porosity (POR_calc), please find the performance of POR_calc and PTF03 in the below table (please find its readable version here: https://gmd.copernicus.org/preprints/gmd-2020-36/gmd-2020-36-AC2-supplement.pdf).

Method ME (cm3 cm-3) RMSE (cm3 cm-3) R2 N POR_calc -0.007 0.038 0.789 1274

PTF03 0.000 0.031 0.862 1274

RC3: L31, P10. Elaborate the range of Ks values used in training for PTF02, so reader can judge how low is RMSE of 0.94.

AGREED.

A: Thank you for giving this helpful viewpoint. We will add it in that sentence: "In the case of KS prediction, the simplest best performing PTF – which was derived on a training dataset with KS ranging between -3.00 and 4.67 log10(cm day-1) – has an RMSE of 0.94 log10(cm day-1) . . ."

RC3: L1-8, P10. You can compare the randomized RMSE by PTF02 (RMSE/(maxKs-minKs)) by some studies in the literature (preferably Europe or at least temperate soils)

AGREED

A: Thank you for this suggestion. We computed it for all the derived PTFs and will highlight this error measure in the case of KS and call it normalized RMSE (NRMSE). We also computed the NRMSE for - the literature referred in the manuscript: - Zhang and Schaap (2017) (ROSETTA3): 0.11 (cm/day) (PSD+BD) - Lilly et al. (2018) (HYPRES) 0.18 log10 (cm/day) (topsoil/subsoil distinction+USDA soil texture class+PSD+BD+OC), - Araya and Ghezehei (2019) (USKSAT database) 0.06 log10 (cm/day) (PSD+BD+OC), - Nemes et al. (2005) (HYPRES) 0.15 log10 (cm/day).

We will add the information on computing NRMSE to P6 L10 in the manuscript as mentioned above, and the following:

P9 L31: ". . . has an RMSE of 0.94 log10(cm day-1) and NRMSE 0.14 log10(cm day-1) (Table S3)."

P10 L3-7: "ROSETTA3 PTF with PSD and BD predictors had and RMSE of 0.68 log10 (cm day-1) with an NRMSE of 0.11 log10 (cm day-1) (Zhang and Schaap, 2017). Araya and Ghezzehei (2019) published PTF using PSD, BD and OC predictors with highest
accuracy in the literature with an RMSE of 0.34 log10 (cm day-1) and NRMSE of 0.06 log10 (cm day-1). In Lilly et al. (2008), the performance of the KS predictions and findings were similar to this study. They report an RMSE between 0.95 and 1.08 log10(cm day-1) – with an NRMSE between 0.17 and 0.20 log10(cm day-1) – for the KS prediction when analysed with several input combinations."

RC3: L19, P10. I expect to see the high importance of clay in THETAr. It is not clear exactly how to estimate VG and MVG parameters.

AGREE SOMEWHAT

A: It is right, expectation is not supported by the data, please see our answer above related to correlation plot: scatterplot of THR vs USCLAY. The reason for it can be that THETAr is a fitting parameter and for most of the samples it was close to 0. Please find here the histogram of THETAr and clay content based on all EU-HYDI samples that has measured chemical properties and fitted THETAr values. Please find the histograms on Fig_responses_8 below the text.

We also point out, that during the estimation of THR in the original model fitting of VG or MVG, THR is not only influenced by clay content, but also by pore connectivity, next to other soil structural properties. Importantly, THR is also influenced by the data range available during the fitting of the original data (Weber et al., 2020), which is a viable reason for the correlation between THR and USCLAY not to be as pronounced as one would expect.

Weber, T.K.D., Finkel, M., Conceição Gonçalves, M., Vereecken, H., Diamantopoulos, E., 2020. Pedotransfer function for the Brunswick soil hydraulic property model and comparison to the van Genuchten‐Mualem model. Water Resour. Res. https://doi.org/10.1029/2019WR026820

Each VG and MVG parameters are predicted separately with random forest models.

RC3: L23, P10. K0, matching point should be defined earlier.

AGREED

A: The following text will be added to P4 L10: Similarly to euptfv1, for the description of the moisture retention curve (MRC), we predicted the VG model parameters: the residual water content ($\theta r$), the saturated water content ($\theta s$), and shape parameters $\alpha$ and n. For the hydraulic conductivity curve, two additional parameters: the hydraulic conductivity acting as a matching point at saturation K0 and a shape parameter related to pore tortuosity (L) are estimated too.

RC3: L25-30, P11. How many of K data are obtained from evaporation method, this method usually goes up to -1000 cm, is it why overestimation occurs in Fig S21 in drier conditions or another reason? Note that in this dry region K data is obviously small and mean error of about 0.8 is significant. Moreover, comparing Fig s21 with Fig S1b (Toth et al., 2015), there less error in this dry region was observed.

AGREED

A: We will delete the sentence starting with "In parts, this is . . ." (P11 L32- P12 L2) and add the following text to P12 L2:

"Samples with measurements of the HCC at pressure heads < -1000 cm are less frequent and are not as numerous within a dataset of a single sample, if it was measured. Since the dataset of estimated VG model parameters were identical in this study and in Tóth et al. (2015), differences between the two studies of the unsaturated HCC are related to the PTF methods involved. However, at pressure heads <-1000 cm, the HCC is dominated by non-capillary conductivity (Weber et al., 2019, Streck and Weber 2020), which is not included in the MVG model. The considerable data mismatch observable for the dry range (Fig. 6) can only be overcome by a different soil hydraulic property model and by a different PTF, because of compensatory effects in the VG. With this we mean that better data descriptions in the dry end, will lead to a larger mismatch in the wet end, as a consequence of the rigid model structure in the MVG model, which only accounts for capillary storage and conductivity. For better data description at <-1000

cm other more comprehensive models need to be adopted (Weber et al. 2020)."

Streck, T., Weber, T.K.D., 2020. Analytical expressions for noncapillary soil water retention based on popular capillary retention models. Vadose Zo. J. 19, 1–5. https://doi.org/10.1002/vzj2.20042 Weber, T.K.D., Finkel, M., Conceição Gonçalves, M., Vereecken, H., Diamantopoulos, E., 2020. Pedotransfer function for the Brunswick soil hydraulic property model and comparison to the van Genuchten‐Mualem model. Water Resour. Res. https://doi.org/10.1029/2019WR026820

RC3: Fig2, 5. Explain the term "count" in legend

AGREED

A: The following will be added to Figure 2 and Figure S1: "; Count: the number of cases in each rectangle."

Figures 5 and 6: "; Count: the number of cases in each hexagon."

RC3: Table 7. RMSE is log10(cm/d) but this belongs to retention curve.

AGREED

A: Thank you for noting it, the unit was wrongly written in the title, we will correct it to cm3 cm-3.

RC3: Table 8. this RMSE was computed only by K(h) data? Did you consider Lambda=0.5?

AGREED

A: Yes, the RMSE is based on the predicted and measured K(h) data. We did not set Lambda = 0.5, but fitted it for the dataset based on measured K(h) data. For the description of the hydraulic conductivity curve we predicted all of the following parameters: $\theta$r: residual water content (cm$^3$ cm-3), $\theta$s: saturated water content (cm$^3$ cm-3), $\alpha$ (cm-1) and n (-): fitting parameters, K0: the hydraulic conductivity acting as a matching

point at saturation (cm day-1) and L: shape parameter related to pore tortuosity (-). Parameter m is provided based on m=1-1/n (van Genuchten, 1980). Thank you for highlighting it.

We will add a paragraph entitled "Practical guidance on how to use the PTFs" on P12 L20, in which we shortly summarize what parameters are predicted with euptfv2.

RC3: L5, P 12. That's interesting to show Comparison of point and parameter predictions, however, you should emphasize that this works only when water retention curve matters. Because one can use the n value of WRC and l=0.5 for K function.

AGREED

A: We will strengthen the description on why point and parameter predictions were compared. To overcome this confusion, we will add to P6 L16:

"The aim of this comparison was to analyse whether point or parametric prediction performs better when only THS and/or FC/FC_2 and/or WP are needed."

and add the complementary information on P12 L8:

"When moisture retention curve is not needed, but only THS and/or FC/FC_2 and/or WP, we recommend to compute those with the point PTFs, more detailed explanation on it is included in Tóth et al. (2015)."

RC3: During some trials to run the package, I have faced with various errors such as Error in source_data ("https://github.com/TothSzaboBrigitta/euptfv2/blob/master/suggested_PTFs/FC_EUHYDI/FC_PTF07.rdata?raw=True") : could not find function "source_data" please check the files again in the attached zip files. I could not also find neither euptf1nor 2 in CRAN repository.

AGREED

A: As mentioned above, the github repository includes the R scripts, that were used to develop the predictions and the derived pedotransfer functions.

The dataset which we used for training and testing the algoritms can not be shared according to the agreement between the data holders. euptfv1 is available from: https://esdac.jrc.ec.europa.eu/themes/soil-hydraulic-properties , https://esdac.jrc.ec.europa.eu/public_path/shared_folder/themes/euptf.zip . The PTFs of euptfv2 are available from the web interface which can be used without any coding skills. The R package is under construction. After finalizing the package it will be available from the European Soil Data Center site of the EC JRC (https://esdac.jrc.ec.europa.eu/). It will not be possible to have the package in the CRAN repository because it will have too large size for it – it will include several RF models.

Please also note the supplement to this comment:
https://gmd.copernicus.org/preprints/gmd-2020-36/gmd-2020-36-AC2-supplement.pdf
* * *
[Figure]

**Fig. 1.** Fig_responses_1

[Figure]

**Fig. 2.** Fig_responses_2

[Figure]

**Fig. 3.** Fig_responses_3

[Figure]

[Figure]

**Fig. 4.** Fig_responses_4

[Figure]

**Fig. 5.** Fig_responses_5

[Figure]

**Fig. 6.** Fig_responses_6

**Fig. 7.** Fig_responses_7

[Figure]

**Fig. 8.** Figure 3

[Figure]

**Fig. 9.** Figure 7

[Figure]

**Fig. 10.** Figure S2

[Figure]

[Figure]

**Fig. 11.** Fig_responses_8

---

## Author Comment (AC3) · 27 Jul 2020

Thank you for the positive review.

---

## Author Response (AR1)

**Revision of manuscript on "Updated European hydraulic pedotransfer functions with communicated uncertainties in the predicted variables (euptfv2)"**

**Content of the document**

**I. POINT BY POINT AUTHORS' RESPONSE TO THE REVIEWERS**

**1. RESPONSE TO REFEREE #1**

Thank you for the positive review.

**2. RESPONSE TO REFEREE #2**

Thank you for the review and constructive comments. We addressed the comments in a revised version of the article. Below we give details on exactly how we addressed the concerns raised by anonymous referee 2. Please note the following during reading the responses:

– the responses are in blue regular font and follow the referee's questions (RC2),

– new text parts that were added to the manuscript are in blue italic font,

– the reference to the lines (L) and pages (P) relates to the marked up version of the manuscript available in III. MARKED-UP MANUSCRIPT AND SUPPLEMENTARY MATERIAL VERSION section of this document.

**RC2:**

Soil pedotransfer functions are important when used for estimation of soil hydraulic parameters in catchment, regional, or continental scale applications. This manuscript improves the estimation of euptfv1 and provides information about prediction uncertainty, and can be applied for more predictor variable combinations than the euptfv1. Overall, the manuscript is interesting, important, well written, and organized in a logical well. Therefore, I recommend accepting this manuscript after minor revisions that are required to address the general and specific comments provided below.

    A:    Thank you for the positive general comment.

**RC2:**

1. The authors compared the estimation of water content at saturation, field capacity, wilting point, plant available water content, saturated hydraulic conductivity, etc., individually. I think these sections are somewhat lengthy. However, the most interesting part of the comparisons between point and parameter predictions and euptfv1 and v2 are very short. Is it possible to extend the comparisons and the discussion?

AGREED.

A1: Regarding comparison between point and parameter predictions, we added the following to be more specific:

P12 L30-31:
*„ … more accurate and for further 8 cases RMSE were smaller."*

P12 L31 – P13 L2:
*„The reason for higher RMSE in parameter estimation can be that the VG model does not always adequately describe the measured MRC data (Weber et al., 2019). Therefore, when THS, FC, FC_2 and WP are computed with parameter estimation those are not only affected by the uncertainty of the prediction of VG parameters but by the goodness of VG model fit as well."*

P13 L4-6: we rephrased the sentence to make it clearer, which now reads:
*„For THS point estimation performed better than parameter estimation. When the moisture retention curve is not needed, but only THS and/or FC/FC_2 and/or WP, we recommend to compute those with …"*

In order to include the suggested comparison, between euptfv1 and v2, we included the following sentences

P13 L9-13:
*"The most important reason for it can be that the interaction between the target variable and the predictors is more complex for the cases of predicting FC or VG parameters – to describe the MRC –, which can be untangled using random forest. This may provide a reasons the random forest algorithm performed significantly better than the PTFs derived with linear regression or a simple regression tree."*

P13 L14-L21:
*„The RMSE of THS prediction was somewhat lower for euptfv1 than for euptfv2, but the difference was not significant. It could be due to the close to linear relationship between THS and BD and high relative importance of BD in THS prediction (84 %). This way their interaction can be efficiently described with the linear regression which is capable to extrapolate as well. Extrapolation with the random forest algorithm is not possible outside the training data, which can limit its performance. The general improvement of the PTFs in euptfv2 is threefold, it is due to i) using random forest instead of single regression tree or linear regression, ii) including more detailed information on soil sampling depth, not only distinguishing topsoils and subsoils and iii) providing information on prediction uncertainty.*

Regarding the description of the individual point and parameter estimations we kept the details because we think it is instructive to provide information about the importance of specific predictors.

**RC2:**
2. The authors listed so many PTFs. When I was reading the conclusion part, I cannot find which PTF I should use. Is it possible to make some concluding remarks regarding which PTFs should be used for corresponding predictors? I think this will be very helpful for future readers.
AGREED.

A2:     Thank you very much for this very helpful comment. Indeed, it is very important that users should easily understand which PTF to select and apply. To achieve this, we i) added a dedicated paragraph on it above the Conclusion section, ii) highlighted in the abstract and short summary that this section was provided, iii) moved Table S3 from supplementary material to the manuscript as Table 11.

The new paragraph 4 reads P13 L25 – P14 L19:

*"4. Practical guidance on how to use the PTFs*
*The minimum input requirements for all PTFs are sand, silt and clay content, and soil depth. Soil depth needs to be considered in regard to the depth of the other input properties and soil hydraulic data needs, e.g. if the soil hydraulic properties of the top 20 cm (0-20 cm) is needed, then depth needs to be set at 10 cm in the input data of the prediction.*
*If only soil texture information is available for the predictions, the class PTFs from euptfv1 could be applied (Tóth et al., 2015).*
*We emphasise that:*
1. *the units of input soil properties (predictors) have to be the same as indicated in the text and that the sand, silt, and clay are defined by the following particle diameters: clay < 2 µm, silt between 2 and 50 µm, and sand between 50 and 2000 µm,*
2. *when only specific water content values at saturation, field capacity or wilting point are required (ie. THS, FC_2, FC, WP) it is recommended to use point PTFs. This is also true for the prediction of KS,*
3. *for AWC, the most accurate way is by first predicting FC and WP with the point predictions and then compute AWC using Eq. (1), and similarly for AWC_2 using FC_2 and Eq. (2),*
4. *it is recommended to do the VG prediction if only moisture retention curve parameters are needed, and*
5. *the MVG prediction when both moisture retention and hydraulic conductivity parameters are required.*
*The VG algorithms predict the following van Genuchten model parameters: the residual water content $\theta_r$ (cm³ cm⁻³), the saturated water content $\theta_s$ (cm³ cm⁻³), and shape parameters $\alpha$ (cm⁻¹) and n (-). Parameter m is provided based on m=1-1/n (van Genuchten, 1980), and for the hydraulic conductivity curve, the two additional parameters: $K_0$ (cm day⁻¹) the hydraulic conductivity acting as a matching point at saturation and L, the shape parameter related to pore tortuosity (-).*

*Table 11 shows the recommended PTFs for each predicted soil hydraulic property and available predictor variables. The users need to check which basic soil properties are available for the predictions, then look in Table 11 which PTF is recommended to use. The algorithms have been implemented in a web interface to facilitate the use of the PTFs, where the PTFs' selection is automated based on soil properties available for the predictions and required soil hydraulic property. The Code and data availability section provides information on how to access this resource."*

The additional text in the short summary and the abstract is given by:

Short summary:
*"... The influence of predictor variables on predicted soil hydraulic properties is explored and practical guidance on how to use the derived PTFs is provided. ..."*

Abstract (P1 L25):
*"... for the prediction of water content at -100 cm matric potential head and plant available water content. A practical guidance on how to use the derived PTFs is provided."*

**Specific comments:**

**RC2:**
1. Figures 2, 5, and 6: Is it possible to include R2 in these figures? This will make the comparison between different figures easier.
AGREED
A1:    We added R2 to Figures 2 (P34), 5 (P37), 6 (P38) and S1 (P2 in supplementary material).

**RC2:**
2. In the abstract and conclusion sections: -15.000 should be -15,000
AGREED
A2:    Thank you for noting it, we corrected it in the entire text.

**RC2:**
3. Page 6, line 4: why did the authors utilize median values instead of mean values?
Nothing changed.
A3:    Our aim was to provide information about the uncertainty of the predictions, therefore we applied quantile regression forests. This way the most probable predicted response value is at the 50th percentile, i.e. the median, which is considered more robust against the outliers than the mean. In this way we decided to use the median as the predicted value (yhat) rather than the mean.

**RC2:**
4. Page 7, line 19: "in the study of (Khodaverdiloo et al., 2011)" should be "in the study of Khodaverdiloo et al. (2011)"
AGREED
A4:    Thank you, we corrected it (P7 L30).

**RC2:**
5. Page 10, line 4: "and RMSE" should be "an RMSE"
AGREED
A5:    Thank you, we corrected it (P10 L19).

**RC2:**
6. Page 10, Line 27: ";" should be ","
AGREED

**RC2:**
7. Page 12, line 14: add a connection/linking word before "it is due to"
AGREED
A7:     The text is rephrased to (P13 L18-19): *"The general improvement of the PTFs in euptfv2 is threefold, the better performance is due to …"*.

**3. RESPONSE TO REFEREE #3**

Thank you for the detailed review and suggestions for further improvements. In the following, we give a detailed presentation of how we addressed the questions and issues raised, following the referee's questions (RC3). Please note the following during reading the responses:

– the responses are in blue regular font and inserted under the referee's questions,
– new text parts that were added to the manuscript are in blue italic font,
– the reference to the lines (L) and pages (P) relates to the marked up version of the manuscript available in III. MARKED-UP MANUSCRIPT AND SUPPLEMENTARY MATERIAL VERSION section of this document.

**RC3:**
This manuscript aims to update the previously developed PTF for European soils called euptfv1. More importantly, euptfv2 contributes to the understudied issue of uncertainty in PTFs for potential users. Despite the existing large amount of results, the paper is easy to follow with some possibilities to improve.
A:      Thank you for the positive general comment.

**RC3:**
The authors also provide a detailed and user-friendly website from euptfv2, however, no library called eutptf exists in R Repository, even the available zip file has problems to be run.
A:      The R package of euptfv2 is under construction.
        The available zip files include the R scripts used to develop the predictions and the derived pedotransfer functions. The dataset which we used for training and testing the algorithms cannot be shared according to the agreement between the data holders. Regarding the model development the following information is included separately for point and parameter estimations: i) loading data, define path, input variables and function to compute performance of the PTFs (setupRF.R), ii) parameter tuning of the random forest (tuneRF.R), iii) building final random forest (buildfinalRF.R), iv) compute performance of the final random forest on the test set (testRF.R) .
        In a separate folder (https://github.com/TothSzaboBrigitta/euptfv2/tree/master/help) a sample input dataset (data_sample.csv) and an R script (apply_PTFs_script.R) - which shows some examples on how to apply the PTFs in R – have been added to the repository.

**RC3:**
Many comparisons among the possibilities of PTFs for different soil hydraulic properties were done. These series of "euptfv(i)" will contribute to the modelling of soil processes. I recommend this paper

for publication, however, I outlined some questions and comments as below (L denotes line and P for page)

A:    Thank you for considering the usability of the euptfs.

**RC3:**
L30, P1. variably saturated fluxes? do you mean flow through variably saturated soil media?
AGREED

A:    Yes, thank you for noting it, we corrected it (P1 L30 – P2 L1): "Simulations of flow through variably saturated soil media either rely on …"

**RC3:**
L31. P.2. Not necessary to machine learning-based methods are able to calculate uncertainty because the sampling effect can propagate parameter uncertainty, which can be implemented even in simple regression-based models. Tens of resamples for training and testing with different distributions can be drawn from the population (Tranter etal., 2010; Kotlar et al., 2019).
Do train and test datasets in bootstraps follow the same distributions?
Tranter, G., Minasny, B. and McBratney, A.B., 2010. Estimating pedotransfer function prediction limits using fuzzy k-means with extragrades. Soil Science Society of AmericaJournal, 74(6), pp.1967-1975.
Kotlar, A.M., de Jong van Lier, Q., Barros, A.H.C., Iversen, B.V. and Vereecken, H.,2019. Development and Uncertainty Assessment of Pedotransfer Functions for Predicting Water Contents at Specific Pressure Heads. Vadose Zone Journal, 18(1).
AGREED

A:    Thank you to highlight it with references, we added this information to P2 L17-19:

*"Tranter et al. (2010) developed an uncertainty estimation method using fuzzy k-means with extragrades classification that can be applied in any PTF prediction. Kotlar et al. (2019) presented uncertainty assessment of PTFs through deriving PTFs on tens of resamples for train and test sets."*

and the following to P3 L1-3:

*"If PTFs are derived with these algorithms, the uncertainty of the predicted soil property can be directly estimated when applying the PTF (Szabó et al., 2019a), although this could also be achieved by applying the above mentioned uncertainty assessment methods without using machine learning methods (e.g. Kotlar et al., 2019; Tranter et al., 2010)."*

Using Kolmogorov–Smirnov tests, we tested whether training and test sets have the same frequency distributions, please find the results in Table 1. For THS, FC and WP the distribution of training and TEST_BASIC set is equal in almost all the cases of the most important basic soil properties. For KS, the distribution of sand and organic carbon content is equal in the training and TEST_BASIC set, in case of FC_2 only the distribution of sand content is equal based on the statistical test. The distributions of training and TEST_CHEM+ sets are equal only in case of FC. For the other sets, at least one soil property has equal distribution in the two sets.

Table 1. Results of the Kolmogorov–Smirnov test (p value of 0.05) computed to compare distribution of the most important basic soil properties of training and test datasets.

| Soil hydraulic property | Input variable | p-value of Kolgomorov-Smirnov test | |
|---|---|---|---|
| | | Training vs. TEST_BASIC set | Training vs. TEST_CHEM+ set |
| THS | USSAND | 0,137 | 0,000 |
| | USCLAY | 0,022 | 0,000 |
| | OC | 0,598 | 0,004 |
| | BD | 0,483 | 0,021 |
| FC_2 | USSAND | 0,616 | 0,112 |
| | USCLAY | 0,004 | 0,000 |
| | OC | 0,018 | 0,000 |
| | BD | 0,023 | 0,000 |
| FC | USSAND | 0,019 | 0,157 |
| | USCLAY | 0,172 | 0,078 |
| | OC | 0,662 | 0,737 |
| | BD | 0,313 | 0,489 |
| WP | USSAND | 0,730 | 0,007 |
| | USCLAY | 0,372 | 0,003 |
| | OC | 0,649 | 0,000 |
| | BD | 0,047 | 0,074 |
| KS | USSAND | 0,396 | 0,000 |
| | USCLAY | 0,001 | 0,008 |
| | OC | 0,755 | 0,001 |
| | BD | 0,000 | 0,000 |

Train and test datasets in bootstraps are divided in the following way: in the random forest algorithm for each tree 63% of the data is selected with replacement to build the tree, i.e. number of selected data will be increased to reach the number of samples of the training set with the replacement, this way some samples will be used multiple times in a single tree. Each tree of the forest is trained on different samples. The forest includes 200 trees and the predicted value is the median of all 200 trees. However, it is difficult to compute the Kolgomorov-Smirnov test for all the 200 in-bag and out-of-bag samples by each predicted soil hydraulic properties, we could confirm based on the literature (Hastie et al., 2009), that the forest will neither be biased nor overfitted to the data because of the two step randomization – bagging process and split-variable randomization – implemented in the algorithm.

**RC3:**
Table1. P.18. Numbers are not aligned exactly below the names.
Correlation matrix of observations would be useful information (in appendix) at least for the dataset used for the best PTFs.
AGREED
> A:     The columns' names were aligned with the numbers below (Table 1 P21).
> The correlation plots of the best PTFs are inserted below the answers (Fig_responses_1 – Fig_responses_7), however descriptive power of them are limited because the relationship between predicted parameters and predictors are not linear. This is the reason why PTFs are derived with a machine learning algorithm and partial dependence plots are shown in the manuscript. We feel that the correlation plots might not provide indispensable information.

[Figure]

Fig_responses_1

[Figure]

Fig_responses_2

[Figure]

Fig_responses_3

[Figure]

Fig_responses_4

[Figure]

Fig_responses_5

[Figure]

Fig_responses_6

[Figure]

Fig_responses_7

**RC3:**

L6, P5: Please calculate variable importance of parameters in PTFs as relative which makes summation of all 100%. (e.g. Figure 3)

AGREED

A:    We replaced Figure 3 (P35) and 7 (P39) with relative importance plots and specified in the figures' caption that relative variable importance is shown.

We added the following text in P6 L1-2:

*"The relative importance was assessed by dividing the variable importance of each predictor by the sum of the importance of all the predictors after Kotlar et al. (2019)."*

**RC3:**

L3, P7: To give a better view of the performance of PTFs, compare the mean values of measured parameters with RMSE of predictions. Compared to Toth et al., (2015), improvement in the prediction of THS is less than FC and WP, why?

AGREED

A:    Thank you for the suggestion. We added the normalized RMSE (RMSE/(ymax-ymin)), which was also suggested by the reviewer under "L1-8, P10".

The following texts was added in the revised manuscript:
P6 L15-19:
*"The different data range of the dataset influences the performance of the PTFs when that is compared to the studies in the literature. Therefore, normalized RMSE (NRMSE) was computed (Eq. 5.), where $y_{max}$ and $y_{min}$ are the maximum and minimum value of variable ."*

$$NRMSE = \frac{RMSE}{y_{max} - y_{min}} \tag{5}"$$

and in P7 L16-18 we add

*"Table S3 shows the NRMSE for the point predictions computed for the TEST_BASIC and TEST_CHEM+ sets to provide possibility for comparison with other PTFs available from the literature."*

And a table on "Normalized root mean square error (NRMSE) of the point predictions by soil hydraulic properties computed on the test datasets in $cm^3$ $cm^{-3}$ for water retention and $log_{10}$ (cm $day^{-1}$) for saturated hydraulic conductivity. In case of PTF01, 02, 03 and 07 TEST_BASIC set was used for the analysis, for the rest of the PTFs TEST_CHEM+ set was considered" was added to the supplementary material as Table S3 (P5 of the revised supplementary material). The original TableS3 of the supplementary material available from https://gmd.copernicus.org/preprints/gmd-2020-36/gmd-2020-36-supplement.pdf was moved to the manuscript as Table 11.

Comparison to Toth et al. (2015): thank you for the reviewer's comment on THS and BD, which helps to clarify findings related to comparison of euptfv1 and v2. There was no significant difference between euptfv1 and v2 in case of THS when BD was available for the prediction and euptfv1 was derived with linear regression. The reason for it – which was mentioned by the reviewer as well – that the relative importance of BD is 84% in the prediction of THS and the relationship between THS and BD is close to linear. In this case random forest could not significantly improve the prediction. In case of FC and WP the interaction between the target variable and the predictors is more complex, this way the random forest algorithm performed significantly better than the PTFs derived with linear regression or a simple regression tree. We added the following information in P13 L9-18:

*"The most important reason for it can be that the interaction between the target variable and the predictors is more complex for the cases of predicting FC or VG parameters – to describe the MRC, which can be untangled using random forest. This may provide a reason the random forest algorithm performed significantly better than the PTFs derived with linear regression or a simple regression tree. For THS, WP, KS, and MVG only those PTFs did not improve significantly, for which comparisons on the TEST_CHEM+ set were possible – which includes reduced number of samples. The RMSE of THS prediction was somewhat lower for euptfv1 than for euptfv2, but the difference was not significant. It could be due to the close to linear relationship between THS and BD and high relative importance of BD in THS prediction (84 %). This way their interaction can be efficiently described with the linear regression which is capable to extrapolate as well. Extrapolation with the random forest algorithm is not possible, which can limit its performance."*

**RC3:**
Figure S2, please replace SE by RMSE so the reader doesn't lose the track of comparison criteria.
AGREED
    A:    We replaced Figure S2, S4, S6, S8, S10, S12, S14, S16, S19 showing SE with the one showing RMSE, please find the new figures on P6, 8, 10, 12, 14, 16, 18, 20, 23 of the revised supplementary material below (III. MARKED-UP MANUSCRIPT AND SUPPLEMENTARY MATERIAL VERSION).

**RC3:**

L1, P8. Please mention the correlation between THS and BD, lets arguably consider THS equal to total porosity, does the 1-BD/PD, assuming PD=2.65 give better RMSE than PTF03 for THS? or you might easily obtain the best PD to predict THS by this formula. In PTF 32, the relative importance of BD is almost 100%.

DISAGREE

A: Thank you for this idea, however, to remain consistent in methodology we would make use of the better performing PTF based on the random forest. The reason: the correlation between THS and BD is -0.92. We have computed the porosity on the test dataset of PTF03 (N = 1274) based on BD and PD (=2.65 g/cm3), then the RMSE of it. We found that the RMSE of PTF03 is smaller than that of porosity (POR_calc), please find the performance of POR_calc and PTF03 in the below table.

| Method | ME ($cm^3 cm^{-3}$) | RMSE ($cm^3 cm^{-3}$) | $R^2$ | N |
|---|---|---|---|---|
| POR_calc | -0.007 | 0.038 | 0.789 | 1274 |
| PTF03 | 0.000 | 0.031 | 0.862 | 1274 |

**RC3:**

L31, P10. Elaborate the range of Ks values used in training for PTF02, so reader can judge how low is RMSE of 0.94.

AGREED.

A: Thank you for giving this helpful viewpoint. We will added it in that sentence (P10 L11-12):

*"In the case of KS prediction, the simplest best performing PTF – which was derived on a training dataset with KS ranging between -3.00 and 4.67 $log_{10}(cm\ day^{-1})$ – has an RMSE of 0.94 $log_{10}(cm\ day^{-1})$ …"*

**RC3:**

L1-8, P10. You can compare the randomized RMSE by PTF02 (RMSE/(maxKs-minKs)) by some studies in the literature (preferably Europe or at least temperate soils)

AGREED

A: Thank you for this suggestion. We computed it for all the derived PTFs and highlighted this error measure in the case of KS and called it normalized RMSE (NRMSE). We also computed the NRMSE for
- the literature referred in the manuscript:
  - Zhang and Schaap (2017) (ROSETTA3): 0.11 log10 (cm/day) (PSD+BD)
  - Lilly et al. (2018) (HYPRES) 0.18 log10 (cm/day) (topsoil/subsoil distinction+USDA soil texture class+PSD+BD+OC),
  - Araya and Ghezehei (2019) (USKSAT database) 0.06 log10 (cm/day) (PSD+BD+OC),
- Nemes et al. (2005) (HYPRES) 0.15 log10 (cm/day).

We added the information on computing NRMSE to P6 L15-19 in the manuscript as mentioned above, and the following:

P10 L12-13:
*"… has an RMSE of 0.94 $log_{10}(cm\ day^{-1})$ and NRMSE 0.14 $log_{10}(cm\ day^{-1})$ (Table S3)."*

P10 L17-22:
*"ROSETTA3 PTF with PSD and BD predictors had and RMSE of 0.68 $log_{10} (cm\ day^{-1})$ with an NRMSE of 0.11 $log_{10} (cm\ day^{-1})$ (Zhang and Schaap, 2017). Araya and Ghezzehei*

*(2019) published PTF using PSD, BD and OC predictors with highest accuracy in the literature with an RMSE of 0.34 log$_{10}$ (cm day$^{-1}$) and NRMSE of 0.06 log$_{10}$ (cm day$^{-1}$). In Lilly et al. (2008), the performance of the KS predictions and findings were similar to this study. They report an RMSE between 0.95 and 1.08 log$_{10}$(cm day$^{-1}$) – with an NRMSE between 0.17 and 0.20 log$_{10}$(cm day$^{-1}$) – for the KS prediction when analysed with several input combinations."*

**RC3:**

L19, P10. I expect to see the high importance of clay in THETAr. It is not clear exactly how to estimate VG and MVG parameters.

AGREE SOMEWHAT

A: It is right, expectation is not supported by the data, please see our answer above related to correlation plot: scatterplot of THR vs USCLAY. The reason for it can be that THETAr is a fitting parameter and for most of the samples it was close to 0. Please find here (Fig_responses_8) the histogram of THETAr and clay content based on all EU-HYDI samples that has measured chemical properties and fitted THETAr values:

[Figure]

[Figure]

Fig_responses_8

We also point out, that during the estimation of THR in the original model fitting of VG or MVG, THR is not only influenced by clay content, but also by pore connectivity, next to other soil structural properties. Importantly, THR is also influenced by the data range available during the fitting of the original data (Weber et al., 2020), which is a viable reason for the correlation between THR and USCLAY not to be as pronounced as one would expect.

Weber, T.K.D., Finkel, M., Conceição Gonçalves, M., Vereecken, H., Diamantopoulos, E., 2020. Pedotransfer function for the Brunswick soil hydraulic property model and comparison to the van Genuchten-Mualem model. Water Resour. Res. https://doi.org/10.1029/2019WR026820

Each VG and MVG parameters are predicted separately with random forest models.

**RC3:**

L23, P10. K0, matching point should be defined earlier.

AGREED

A:    The following text was added to P4 L13-16:

*Similarly to euptfv1, for the description of the moisture retention curve (MRC), we predicted the VG model parameters: the residual water content ($\vartheta_r$), the saturated water content ($\vartheta_s$), and shape parameters α and n. For the hydraulic conductivity curve, two additional parameters: the hydraulic conductivity acting as a matching point at saturation $K_0$ and a shape parameter related to pore tortuosity (L) are estimated too.*

**RC3:**

L25-30, P11. How many of K data are obtained from evaporation method, this method usually goes up to -1000 cm, is it why overestimation occurs in Fig S21 in drier conditions or another reason? Note that in this dry region K data is obviously small and mean error of about 0.8 is significant.

Moreover, comparing Fig s21 with Fig S1b (Toth et al., 2015), there less error in this dry region was observed.

AGREED

A:    We deleted the sentence starting with "In parts, this is …" (it was on P11 L32- P12 L2 in the discussion paper available from https://gmd.copernicus.org/preprints/gmd-2020-36/gmd-2020-36.pdf) and added the following text to P12 L19-27:

*"Samples with measurements of the HCC at pressure heads < -1000 cm are less frequent and are not as numerous within a dataset of a single sample, if it was measured. Since the dataset of estimated VG model parameters were identical in this study and in Tóth et al. (2015), differences between the two studies of the unsaturated HCC are related to the PTF methods involved. However, at pressure heads <-1000 cm, the HCC is dominated by non-capillary conductivity (Weber et al., 2019, Streck and Weber 2020), which is not included in the MVG model. The considerable data mismatch observable for the dry range (Fig. 6) can only be overcome by a different soil hydraulic property model and by a different PTF, because of compensatory effects in the VG. With this we mean that better data descriptions in the dry end, will lead to a larger mismatch in the wet end, as a consequence of the rigid model structure in the MVG model, which only accounts for capillary storage and conductivity. For better data description at <-1000 cm other more comprehensive models need to be adopted (Weber et al. 2020)."*

Streck, T., Weber, T.K.D., 2020. Analytical expressions for noncapillary soil water retention based on popular capillary retention models. Vadose Zo. J. 19, 1–5. https://doi.org/10.1002/vzj2.20042

Weber, T.K.D., Finkel, M., Conceição Gonçalves, M., Vereecken, H., Diamantopoulos, E., 2020. Pedotransfer function for the Brunswick soil hydraulic property model and comparison to the van Genuchten-Mualem model. Water Resour. Res. https://doi.org/10.1029/2019WR026820

**RC3:**

Fig2, 5. Explain the term "count" in legend

AGREED

A:    The following was added to

Figure 2 and Figure S1:

*"; Count: the number of cases in each rectangle."*

Figures 5 and 6:

*"; Count: the number of cases in each hexagon."*

**RC3:**

Table 7. RMSE is log10(cm/d) but this belongs to retention curve.

AGREED

A: Thank you for noting it, the unit was wrongly written in the title, we corrected it to cm³ cm⁻³.

**RC3:**

Table 8. this RMSE was computed only by K(h) data? Did you consider Lambda=0.5?

AGREED

A: Yes, the RMSE is based on the predicted and measured K(h) data.
We did not set Lambda = 0.5, but fitted it for the dataset based on measured K(h) data. For the description of the hydraulic conductivity curve we predicted all of the following parameters: $\theta_r$: residual water content (cm³ cm⁻³), $\theta_s$: saturated water content (cm³ cm⁻³), $\alpha$ (cm⁻¹) and n (-): fitting parameters, $K_0$: the hydraulic conductivity acting as a matching point at saturation (cm day⁻¹) and L: shape parameter related to pore tortuosity (-). Parameter m is provided based on m=1-1/n (van Genuchten, 1980). Thank you for highlighting it.

We added a paragraph entitled "Practical guidance on how to use the PTFs" on P13 L25 – P14 L19, in which we shortly summarize what parameters are predicted with euptfv2.

**RC3:**

L5, P 12. That's interesting to show Comparison of point and parameter predictions, however, you should emphasize that this works only when water retention curve matters. Because one can use the n value of WRC and l=0.5 for K function.

AGREED

A: We will strengthen the description on why point and parameter predictions were compared. To overcome this confusion, we added to P6 L25-26:

*"The aim of this comparison was to analyse whether point or parametric prediction performs better when only THS and/or FC/FC_2 and/or WP are needed."*

and included the complementary information on P13 L5-6:

*"When moisture retention curve is not needed, but only THS and/or FC/FC_2 and/or WP, we recommend to compute those with the point PTFs, more detailed explanation on it is included in Tóth et al. (2015)."*

**RC3:**

During some trials to run the package, I have faced with various errors such as
Error in source_data
("https://github.com/TothSzaboBrigitta/euptfv2/blob/master/suggested_PTFs/FC_EUHYDI/FC_PTF07.rdata?raw=True") : could not find function "source_data"
please check the files again in the attached zip files. I could not also find neither euptf1nor 2 in CRAN repository.

AGREED

A: As mentioned above, the github repository includes the R scripts, which were used to develop the predictions and the derived pedotransfer functions. The dataset which we used for training and testing the algorithms cannot be shared according to the agreement between the data holders. We added the following to P15 L3-4 for clarification:

*"The training data set cannot be made publicly available due to legal restrictions of the EU-HYDI dataset, thus only a test sample is provided along with the model code."*

euptfv1 is available from: https://esdac.jrc.ec.europa.eu/themes/soil-hydraulic-properties                                                                                                          ,
https://esdac.jrc.ec.europa.eu/public_path/shared_folder/themes/euptf.zip .
The PTFs of euptfv2 are available from the web interface which can be used without any coding skills. The R package is under construction. After finalizing the package it will be available from the European Soil Data Center site of the EC JRC (https://esdac.jrc.ec.europa.eu/). It will not be possible to have the package in the CRAN repository because it will have too large size for it – it will include several RF models.

**II. LIST OF AUTHOR'S CHANGES IN MANUSCRIPT AND SUPPLEMENTARY MATERIAL**

Page and line numbering refer to that of the revised manuscript with track changes included under III. MARKED-UP MANUSCRIPT AND SUPPLEMENTARY MATERIAL VERSION section of this document.

**The following changes have been made in the manuscript:**
- P1 L17: matric potential is corrected to 15,000 cm,
- P1 L25: sentence is added on practical guidance,
- P1 L30 – P2 L1: sentence is rephrased,
- P2 L17-19: more information is added related to uncertainty assessment,
- P2 L21: matric potential is corrected to 15,000 cm,
- P3 L2-3: further information is added about the estimation of uncertainty,
- P4 L4: matric potential is corrected to 15,000 cm,
- P4 L13-16: description of model parameters is added,
- P6 L1-2: information on relative importance is added,
- P6 L15-19: computation of normalized root mean square error (NRMSE) is added,
- P6 L25-26: clarifying sentence – related to point and parametric prediction – is added,
- P7 L15: RMSE of KS is rounded to two digits,
- P7 L16-18: sentence on table of NRMSE is added,
- P7 L30: format of citation is corrected,
- P10 L11-13: range of KS in the training dataset and NRMSE is added,
- P10 L16: RMSE is adjusted,
- P10 L17-21: NRMSE of other published PTFs is added,
- P10 L21: RMSE is rechecked and corrected,
- P11 L11: ";" is changed to ",",
- P11 L31: matric potential is corrected to 15,000 cm,
- P12 L1: matric potential is corrected to 15,000 cm,
- P12 L16-18: the sentence starting with "In parts, this is …" is deleted,
- P12 L19-L27: information is added to interpret the results,
- P12 L29: language correction,
- P12 L30 – P13 L2 and P13 L4-6: information and discussion is added,
- P13 L9-13 and L14-21: discussion is added,
- P13 L14: language correction – were instead of was,
- P13 L25 – P14 L19: section on "Practical guidance on how to use the PTFs" is added,

- P14 L25: matric potential is corrected to 15,000 cm,
- P15 L3-4: information about the training dataset is added,
- P16 L25-27: reference is added,
- P18 L31-32: reference is added,
- P19 L8-10: reference is added,
- P19 L31-32: reference is added,
- P29: "a" is formatted to superscript in the 3rd row of WP,
- P31: Table 11 is added, which lists the recommended PTFs,
- P33-34: Figure 2 is reedited and replaced – $R^2$ is added,
- P34 L7: description of count is added,
- P35: Figure 3 is reedited and replaced – relative importance is shown,
- P35 L3: figure's caption is corrected,
- P37: Figure 5 is replaced – $R^2$ is added,
- P37 L6-7: description of count is added,
- P38: Figure 6 is replaced – $R^2$ is added,
- P38 L6-7: description of count is added,
- P39: Figure 7 is reedited and replaced – relative importance is shown,
- P39 L3-5: figure's caption is corrected.

**The following changes have been made in the short summary:**
- complementing the following sentence with the text in blue: "The influence of predictor variables on predicted soil hydraulic properties is explored and practical guidance on how to use the derived PTFs is provided."

**The following changes have been made in the supplementary material:**
- P1-2: Figure S1: figure is reedited and replaced – $R^2$ is added, meaning of count is added in the caption (L7).
- P5: Table S3 on NRMSE values added,
- P6-17, 19: Figures S2, S4, S6, S8, S10, S12, S14, S16, S19 are reedited and replaced and "Squared error (SE)" is changed to "Root mean square error (RMSE)" in the figures' captions,
- P21-22: Table S3 is moved to the manuscript.

**III. MARKED-UP MANUSCRIPT AND SUPPLEMENTARY MATERIAL VERSION**

Please find revised marked-up manuscript and supplementary material on the following pages.

[revised manuscript text omitted]

\* Correspondence: toth.brigitta@agrar.mta.hu (B.Sz.)

[Figure]

**Figure S1.** The scatter plot of the measured versus predicted plant available water content values of the worst and best performing PTF with 90% prediction interval on test datasets. AWC_2: plant available water content based on filed capacity at -100 cm matric potential head (PTF01 vs. PTF03); AWC: plant available water content based on filed capacity at -330 cm matric potential head (PTF01 vs. PTF03); PSD: particle size distribution (sand, 50–2000 μm; silt, 2–50 μm; clay, <2 μm (mass %)); DEPTH_M: mean soil depth (cm); BD: bulk density (g cm$^{-3}$); Count: the number of cases in each rectangle.

**Table S1.** Performance of pedotransfer functions (PTF) by input combination on training and test datasets to predict the plant available water content of the soil (AWC_2) belonging to the -100 cm matric potential head. N: number of samples, RMSE: root mean square error (cm$^3$ cm$^{-3}$), and R$^2$: determination coefficient, TEST_BASIC: samples with measured PSD, DEPTH, OC and BD; TEST_CHEM+: samples with measured PSD, DEPTH, OC, BD, CACO3, PH_H2O and CEC. Recommended PTFs are highlighted in bold.

| Name of PTF in euptfv2 | Predictor variables[1] | Training set | | | Test set | | | Sign. difference[2] | | Recommended PTF |
|---|---|---|---|---|---|---|---|---|---|---|
| | | N | RMSE | R$^2$ | N | RMSE | R$^2$ | TEST_BASIC set | TEST_CHEM+ set | |
| **PTF01** | **PSD+DEPTH** | 3528 | 0.062 | 0.446 | 1372 | 0.060 | 0.432 | a | ab | PTF01 |
| **PTF02** | **PSD+DEPTH+OC** | 3208 | 0.055 | 0.540 | 1372 | 0.054 | 0.544 | b | abcd | PTF02 |
| **PTF03** | **PSD+DEPTH+BD** | 3472 | 0.054 | 0.581 | 1372 | 0.053 | 0.552 | b | abcd | PTF03 |
| PTF04 | PSD+DEPTH+CACO3 | 1548 | 0.050 | 0.326 | 274 | 0.055 | 0.219 | - | abcd | PTF01 |
| PTF05 | PSD+DEPTH+PH_H2O | 1849 | 0.058 | 0.463 | 274 | 0.055 | 0.216 | - | a | PTF01 |
| PTF06 | PSD+DEPTH+CEC | 1550 | 0.059 | 0.512 | 274 | 0.060 | 0.050 | - | abcd | PTF01 |
| PTF07 | PSD+DEPTH+OC+BD | 3197 | 0.051 | 0.609 | 1372 | 0.051 | 0.588 | b | abcd | PTF03 |
| PTF08 | PSD+DEPTH+OC+CACO3 | 1464 | 0.048 | 0.353 | 274 | 0.053 | 0.257 | - | abcd | PTF02 |
| PTF09 | PSD+DEPTH+OC+PH_H2O | 1615 | 0.055 | 0.490 | 274 | 0.053 | 0.270 | - | abc | PTF02 |
| PTF10 | PSD+DEPTH+OC+CEC | 1358 | 0.054 | 0.563 | 274 | 0.053 | 0.278 | - | abcd | PTF02 |
| PTF11 | PSD+DEPTH+BD+CACO3 | 1545 | 0.044 | 0.470 | 274 | 0.048 | 0.396 | - | d | PTF03 |
| PTF12 | PSD+DEPTH+BD+PH_H2O | 1796 | 0.052 | 0.565 | 274 | 0.048 | 0.406 | - | abcd | PTF03 |
| PTF13 | PSD+DEPTH+BD+CEC | 1498 | 0.053 | 0.598 | 274 | 0.048 | 0.398 | - | abcd | PTF03 |
| PTF14 | PSD+DEPTH+CACO3+PH_H2O | 1195 | 0.051 | 0.341 | 274 | 0.052 | 0.284 | - | abcd | PTF01 |
| PTF15 | PSD+DEPTH+CACO3+CEC | 726 | 0.050 | 0.286 | 274 | 0.052 | 0.303 | - | abcd | PTF01 |
| PTF16 | PSD+DEPTH+PH_H2O+CEC | 1255 | 0.058 | 0.539 | 274 | 0.051 | 0.331 | - | abcd | PTF01 |
| PTF17 | PSD+DEPTH+OC+BD+CACO3 | 1464 | 0.044 | 0.465 | 274 | 0.048 | 0.390 | - | bcd | PTF03 |
| PTF18 | PSD+DEPTH+OC+BD+PH_H2O | 1607 | 0.051 | 0.556 | 274 | 0.048 | 0.407 | - | abcd | PTF03 |
| PTF19 | PSD+DEPTH+OC+BD+CEC | 1349 | 0.052 | 0.593 | 274 | 0.046 | 0.441 | - | abcd | PTF03 |
| PTF20 | PSD+DEPTH+OC+CACO3+PH_H2O | 1130 | 0.050 | 0.367 | 274 | 0.051 | 0.309 | - | abcd | PTF02 |
| PTF21 | PSD+DEPTH+OC+CACO3+CEC | 683 | 0.049 | 0.305 | 274 | 0.050 | 0.359 | - | abcd | PTF02 |
| PTF22 | PSD+DEPTH+OC+PH_H2O+CEC | 1067 | 0.054 | 0.561 | 274 | 0.049 | 0.367 | - | abcd | PTF02 |
| PTF23 | PSD+DEPTH+BD+CACO3+PH_H2O | 1192 | 0.046 | 0.471 | 274 | 0.049 | 0.375 | - | bcd | PTF03 |
| PTF24 | PSD+DEPTH+BD+CACO3+CEC | 725 | 0.045 | 0.420 | 274 | 0.046 | 0.444 | - | d | PTF03 |
| PTF25 | PSD+DEPTH+BD+PH_H2O+CEC | 1204 | 0.052 | 0.621 | 274 | 0.046 | 0.456 | - | abcd | PTF03 |
| PTF26 | PSD+DEPTH+CACO3+PH_H2O+CEC | 684 | 0.049 | 0.318 | 274 | 0.048 | 0.388 | - | abcd | PTF01 |
| PTF27 | PSD+DEPTH+OC+BD+CACO3+PH_H2O | 1130 | 0.045 | 0.475 | 274 | 0.049 | 0.367 | - | abcd | PTF03 |
| PTF28 | PSD+DEPTH+OC+BD+CACO3+CEC | 683 | 0.045 | 0.408 | 274 | 0.045 | 0.466 | - | bcd | PTF03 |
| PTF29 | PSD+DEPTH+OC+BD+PH_H2O+CEC | 1059 | 0.052 | 0.603 | 274 | 0.045 | 0.473 | - | bcd | PTF03 |
| PTF30 | PSD+DEPTH+OC+CACO3+PH_H2O+CEC | 641 | 0.049 | 0.330 | 274 | 0.048 | 0.393 | - | abcd | PTF02 |
| PTF31 | PSD+DEPTH+BD+CACO3+PH_H2O+CEC | 683 | 0.044 | 0.450 | 274 | 0.045 | 0.480 | - | cd | PTF03 |
| PTF32 | PSD+DEPTH+OC+BD+CACO3+PH_H2O+CEC | 641 | 0.045 | 0.425 | 274 | 0.045 | 0.471 | - | cd | PTF03 |

[1]PSD: particle size distribution (sand, 50–2000 μm; silt, 2–50 μm; clay, <2 μm (mass %)); DEPTH: mean soil depth (cm); OC: organic carbon content (mass %); BD: bulk density (g cm$^{-3}$); CACO3: calcium carbonate content (mass %); PH_H2O: pH in water (-); CEC: cation exchange capacity (cmol (+) kg$^{-1}$).

[2]Different letters indicate significant differences at the 0.05 level between the accuracy of the methods based on the squared error; for example performance indicated with the letter c is significantly better than the one noted with letters b and a.

**Table S2.** Performance of pedotransfer functions (PTF) by input combination on training and test datasets to predict the plant available water content of the soil (AWC) belonging to the -330 cm matric potential head. N: number of samples, RMSE: root mean square error (cm$^3$ cm$^{-3}$), and R$^2$: determination coefficient, TEST_BASIC: samples with measured PSD, DEPTH, OC and BD; TEST_CHEM+: samples with measured PSD, DEPTH, OC, BD, CACO3, PH_H2O and CEC. Recommended PTFs are highlighted in bold.

| Name of PTF in euptfv2 | Predictor variables[1] | Training set N | RMSE | R$^2$ | Test set N | RMSE | R$^2$ | Sign. difference[2] TEST_BASIC set | TEST_CHEM+ set | Recommended PTF |
|---|---|---|---|---|---|---|---|---|---|---|
| **PTF01** | **PSD+DEPTH** | 1863 | 0.042 | 0.312 | 705 | 0.048 | 0.196 | a | a | PTF01 |
| PTF02 | PSD+DEPTH+OC | 1650 | 0.041 | 0.337 | 705 | 0.045 | 0.288 | ab | a | PTF01 |
| **PTF03** | **PSD+DEPTH+BD** | 1849 | 0.040 | 0.374 | 705 | 0.045 | 0.285 | ab | a | PTF01 |
| PTF04 | PSD+DEPTH+CACO3 | 1531 | 0.040 | 0.366 | 279 | 0.050 | 0.199 | - | a | PTF01 |
| PTF05 | PSD+DEPTH+PH_H2O | 1245 | 0.042 | 0.344 | 279 | 0.048 | 0.238 | - | a | PTF01 |
| PTF06 | PSD+DEPTH+CEC | 1092 | 0.041 | 0.356 | 279 | 0.053 | 0.078 | - | a | PTF01 |
| PTF07 | PSD+DEPTH+OC+BD | 1645 | 0.040 | 0.381 | 705 | 0.043 | 0.337 | b | a | PTF03 |
| PTF08 | PSD+DEPTH+OC+CACO3 | 1336 | 0.041 | 0.345 | 279 | 0.049 | 0.219 | - | a | PTF01 |
| PTF09 | PSD+DEPTH+OC+PH_H2O | 1074 | 0.042 | 0.345 | 279 | 0.048 | 0.242 | - | a | PTF01 |
| PTF10 | PSD+DEPTH+OC+CEC | 998 | 0.039 | 0.413 | 279 | 0.051 | 0.147 | - | a | PTF01 |
| PTF11 | PSD+DEPTH+BD+CACO3 | 1522 | 0.038 | 0.428 | 279 | 0.048 | 0.258 | - | a | PTF01 |
| PTF12 | PSD+DEPTH+BD+PH_H2O | 1236 | 0.039 | 0.429 | 279 | 0.047 | 0.287 | - | a | PTF01 |
| PTF13 | PSD+DEPTH+BD+CEC | 1088 | 0.038 | 0.429 | 279 | 0.049 | 0.231 | - | a | PTF01 |
| PTF14 | PSD+DEPTH+CACO3+PH_H2O | 1230 | 0.041 | 0.376 | 279 | 0.047 | 0.263 | - | a | PTF01 |
| PTF15 | PSD+DEPTH+CACO3+CEC | 791 | 0.041 | 0.366 | 279 | 0.049 | 0.214 | - | a | PTF01 |
| PTF16 | PSD+DEPTH+PH_H2O+CEC | 739 | 0.042 | 0.321 | 279 | 0.048 | 0.237 | - | a | PTF01 |
| PTF17 | PSD+DEPTH+OC+BD+CACO3 | 1334 | 0.039 | 0.399 | 279 | 0.048 | 0.262 | - | a | PTF03 |
| PTF18 | PSD+DEPTH+OC+BD+PH_H2O | 1072 | 0.040 | 0.393 | 279 | 0.047 | 0.293 | - | a | PTF03 |
| PTF19 | PSD+DEPTH+OC+BD+CEC | 995 | 0.038 | 0.432 | 279 | 0.049 | 0.223 | - | a | PTF03 |
| PTF20 | PSD+DEPTH+OC+CACO3+PH_H2O | 1059 | 0.042 | 0.362 | 279 | 0.047 | 0.289 | - | a | PTF01 |
| PTF21 | PSD+DEPTH+OC+CACO3+CEC | 707 | 0.041 | 0.358 | 279 | 0.049 | 0.229 | - | a | PTF01 |
| PTF22 | PSD+DEPTH+OC+PH_H2O+CEC | 660 | 0.041 | 0.339 | 279 | 0.048 | 0.253 | - | a | PTF01 |
| PTF23 | PSD+DEPTH+BD+CACO3+PH_H2O | 1221 | 0.039 | 0.442 | 279 | 0.047 | 0.267 | - | a | PTF01 |
| PTF24 | PSD+DEPTH+BD+CACO3+CEC | 788 | 0.039 | 0.405 | 279 | 0.047 | 0.269 | - | a | PTF01 |
| PTF25 | PSD+DEPTH+BD+PH_H2O+CEC | 736 | 0.039 | 0.402 | 279 | 0.046 | 0.307 | - | a | PTF01 |
| PTF26 | PSD+DEPTH+CACO3+PH_H2O+CEC | 732 | 0.040 | 0.405 | 279 | 0.048 | 0.254 | - | a | PTF01 |
| PTF27 | PSD+DEPTH+OC+BD+CACO3+PH_H2O | 1057 | 0.040 | 0.415 | 279 | 0.046 | 0.312 | - | a | PTF03 |
| PTF28 | PSD+DEPTH+OC+BD+CACO3+CEC | 705 | 0.040 | 0.383 | 279 | 0.047 | 0.277 | - | a | PTF03 |
| PTF29 | PSD+DEPTH+OC+BD+PH_H2O+CEC | 658 | 0.040 | 0.385 | 279 | 0.046 | 0.315 | - | a | PTF03 |
| PTF30 | PSD+DEPTH+OC+CACO3+PH_H2O+CEC | 653 | 0.040 | 0.395 | 279 | 0.047 | 0.274 | - | a | PTF01 |
| PTF31 | PSD+DEPTH+BD+CACO3+PH_H2O+CEC | 729 | 0.039 | 0.431 | 279 | 0.047 | 0.290 | - | a | PTF01 |
| PTF32 | PSD+DEPTH+OC+BD+CACO3+PH_H2O+CEC | 651 | 0.039 | 0.403 | 279 | 0.046 | 0.307 | - | a | PTF03 |

[1]PSD: particle size distribution (sand, 50–2000 μm; silt, 2–50 μm; clay, <2 μm (mass %)); DEPTH: mean soil depth (cm); OC: organic carbon content (mass %); BD: bulk density (g cm$^{-3}$); CACO3: calcium carbonate content (mass %); PH_H2O: pH in water (-); CEC: cation exchange capacity (cmol (+) kg$^{-1}$).
[2]Different letters indicate significant differences at the 0.05 level between the accuracy of the methods based on the squared error;for example performance indicated with the letter c is significantly better than the one noted with letters b and a.

**Table S3.** Normalized root mean square error (NRMSE) of the point predictions by soil hydraulic properties computed on the test datasets in $cm^3$ $cm^{-3}$ for water retention and $log_{10}$ (cm day$^{-1}$) for saturated hydraulic conductivity. In case of PTF01, 02, 03 and 07 TEST_BASIC set was used for the analysis, for the rest of the PTFs TEST_CHEM+ set was considered.

| Name of PTF in euptfv2 | Predictor variables[1] | NRMSE in test sets[2] | | | | | | |
|---|---|---|---|---|---|---|---|---|
| | | THS | FC_2 | FC | WP | AWC_2 | AWC | KS |
| PTF01 | PSD+DEPTH_M | 0.104 | 0.090 | 0.082 | 0.105 | 0.126 | 0.140 | 0.17 |
| PTF02 | PSD+DEPTH_M+OC | 0.086 | 0.083 | 0.076 | 0.102 | 0.112 | 0.132 | 0.14 |
| PTF03 | PSD+DEPTH_M+BD | 0.048 | 0.079 | 0.074 | 0.100 | 0.111 | 0.132 | 0.17 |
| PTF04 | PSD+DEPTH_M+CACO3 | 0.191 | 0.107 | 0.113 | 0.122 | 0.164 | 0.145 | 0.19 |
| PTF05 | PSD+DEPTH_M+PH_H2O | 0.176 | 0.112 | 0.114 | 0.126 | 0.164 | 0.142 | 0.19 |
| PTF06 | PSD+DEPTH_M+CEC | 0.191 | 0.107 | 0.107 | 0.118 | 0.181 | 0.156 | 0.19 |
| PTF07 | PSD+DEPTH_M+OC+BD | 0.047 | 0.075 | 0.073 | 0.097 | 0.107 | 0.127 | 0.14 |
| PTF08 | PSD+DEPTH_M+OC+CACO3 | 0.184 | 0.097 | 0.109 | 0.117 | 0.160 | 0.143 | 0.19 |
| PTF09 | PSD+DEPTH_M+OC+PH_H2O | 0.167 | 0.095 | 0.107 | 0.119 | 0.158 | 0.141 | 0.18 |
| PTF10 | PSD+DEPTH_M+OC+CEC | 0.172 | 0.098 | 0.108 | 0.116 | 0.158 | 0.150 | 0.18 |
| PTF11 | PSD+DEPTH_M+BD+CACO3 | 0.072 | 0.091 | 0.105 | 0.115 | 0.144 | 0.140 | 0.19 |
| PTF12 | PSD+DEPTH_M+BD+PH_H2O | 0.069 | 0.086 | 0.103 | 0.117 | 0.143 | 0.137 | 0.19 |
| PTF13 | PSD+DEPTH_M+BD+CEC | 0.070 | 0.091 | 0.100 | 0.115 | 0.144 | 0.142 | 0.19 |
| PTF14 | PSD+DEPTH_M+CACO3+PH_H2O | 0.168 | 0.101 | 0.109 | 0.121 | 0.157 | 0.139 | 0.18 |
| PTF15 | PSD+DEPTH_M+CACO3+CEC | 0.179 | 0.102 | 0.106 | 0.113 | 0.155 | 0.144 | 0.19 |
| PTF16 | PSD+DEPTH_M+PH_H2O+CEC | 0.183 | 0.098 | 0.104 | 0.115 | 0.152 | 0.142 | 0.19 |
| PTF17 | PSD+DEPTH_M+OC+BD+CACO3 | 0.070 | 0.089 | 0.102 | 0.111 | 0.145 | 0.139 | 0.18 |
| PTF18 | PSD+DEPTH_M+OC+BD+PH_H2O | 0.070 | 0.083 | 0.103 | 0.116 | 0.143 | 0.136 | 0.18 |
| PTF19 | PSD+DEPTH_M+OC+BD+CEC | 0.070 | 0.087 | 0.099 | 0.113 | 0.139 | 0.143 | 0.18 |
| PTF20 | PSD+DEPTH_M+OC+CACO3+PH_H2O | 0.166 | 0.105 | 0.107 | 0.114 | 0.154 | 0.137 | 0.18 |
| PTF21 | PSD+DEPTH_M+OC+CACO3+CEC | 0.171 | 0.090 | 0.104 | 0.108 | 0.149 | 0.142 | 0.18 |
| PTF22 | PSD+DEPTH_M+OC+PH_H2O+CEC | 0.166 | 0.089 | 0.102 | 0.111 | 0.148 | 0.140 | 0.18 |
| PTF23 | PSD+DEPTH_M+BD+CACO3+PH_H2O | 0.071 | 0.089 | 0.104 | 0.116 | 0.147 | 0.139 | 0.18 |
| PTF24 | PSD+DEPTH_M+BD+CACO3+CEC | 0.071 | 0.085 | 0.099 | 0.110 | 0.138 | 0.139 | 0.19 |
| PTF25 | PSD+DEPTH_M+BD+PH_H2O+CEC | 0.067 | 0.084 | 0.100 | 0.112 | 0.137 | 0.135 | 0.19 |
| PTF26 | PSD+DEPTH_M+CACO3+PH_H2O+CEC | 0.163 | 0.094 | 0.103 | 0.111 | 0.145 | 0.140 | 0.18 |
| PTF27 | PSD+DEPTH_M+OC+BD+CACO3+PH_H2O | 0.072 | 0.086 | 0.101 | 0.111 | 0.148 | 0.135 | 0.18 |
| PTF28 | PSD+DEPTH_M+OC+BD+CACO3+CEC | 0.070 | 0.082 | 0.098 | 0.106 | 0.136 | 0.138 | 0.18 |
| PTF29 | PSD+DEPTH_M+OC+BD+PH_H2O+CEC | 0.068 | 0.083 | 0.095 | 0.109 | 0.135 | 0.134 | 0.18 |
| PTF30 | PSD+DEPTH_M+OC+CACO3+PH_H2O+CEC | 0.162 | 0.100 | 0.101 | 0.108 | 0.145 | 0.138 | 0.17 |
| PTF31 | PSD+DEPTH_M+BD+CACO3+PH_H2O+CEC | 0.070 | 0.081 | 0.097 | 0.108 | 0.134 | 0.137 | 0.18 |
| PTF32 | PSD+DEPTH_M+OC+BD+CACO3+PH_H2O+CEC | 0.069 | 0.079 | 0.097 | 0.107 | 0.135 | 0.135 | 0.18 |

[1]PSD: particle size distribution (sand, 50–2000 μm; silt, 2–50 μm; clay, <2 μm (mass %)); DEPTH: mean soil depth (cm); OC: organic carbon content (mass %); BD: bulk density (g cm$_{-3}$); CACO3: calcium carbonate content (mass %); PH_H2O: pH in water (-); CEC: cation exchange capacity (cmol (+) kg$^{-1}$).

[2]THS: saturated water content (pF 0); FC_2: water content at -100 cm matric potential head (pF 2.0); FC: water content at -330 cm matric potential head (pF 2.5); AWC_2: plant available water content based on FC_2; AWC: plant available water content based on FC; WP: water content at wilting point (pF 4.2); KS: saturated hydraulic conductivity;

[Figure]

**Figure S2.** Root mean sSquared error (RMSE) of the pedotransfer functions derived to predict water content at saturation (THS) computed on TEST_BASIC and TEST_CHEM+ set. USSAND: sand (50–2000 µm) content (mass %); USSILT: silt (2–50 µm) content (mass %), USCLAY: clay (<2 µm) content (mass %); DEPTH_M: mean soil depth (cm); OC: organic carbon content (mass %); BD: bulk density (g cm$^{-3}$); CACO3: calcium carbonate content (mass %); PH_H2O: pH in water (-); CEC: cation exchange capacity (cmol (+) kg$^{-1}$).

[Figure]

**Figure S3.** Density plot of observed (OBS) and predicted median (PSD+DEPTH_M+*) water content at saturation (THS) for selected pedotransfer functions, computed on TEST_BASIC and TEST_CHEM+ set. USSAND: sand (50–2000 μm) content (mass %); USSILT: silt (2–50 μm) content (mass %), USCLAY: clay (<2 μm) content (mass %); DEPTH_M: mean soil depth (cm); OC: organic carbon content (mass %); BD: bulk density (g cm$^{-3}$); CACO3: calcium carbonate content (mass %); PH_H2O: pH in water (-); CEC: cation exchange capacity (cmol (+) kg$^{-1}$).

[Figure]

[Figure]

**Figure S4.** Root mean sSquared error (RMSE) of the pedotransfer functions derived to predict water content at -100 cm matric potential head (FC_2) computed on TEST_BASIC and TEST_CHEM+ set. USSAND: sand (50–2000 μm) content (mass %); USSILT: silt (2–50 μm) content (mass %), USCLAY: clay (<2 μm) content (mass %); DEPTH_M: mean soil depth (cm); OC: organic carbon content (mass %); BD: bulk density (g cm$^{-3}$); CACO3: calcium carbonate content (mass %); PH_H2O: pH in water (-); CEC: cation exchange capacity (cmol (+) kg$^{-1}$).

[Figure]

**Figure S5.** Density plot of observed (OBS) and predicted median (PSD+DEPTH_M+*) water content at -100 cm matric potential head (FC_2) for selected pedotransfer functions, computed on TEST_BASIC and TEST_CHEM+ set. USSAND: sand (50–2000 μm) content (mass %); USSILT: silt (2–50 μm) content (mass %), USCLAY: clay (<2 μm) content (mass %); DEPTH_M: mean soil depth (cm); OC: organic carbon content (mass %); BD: bulk density (g cm$^{-3}$); CACO3: calcium carbonate content (mass %); PH_H2O: pH in water (-); CEC: cation exchange capacity (cmol (+) kg$^{-1}$).

[Figure]

**Figure S6.** Root mean sSquared error (RMSE) of the pedotransfer functions derived to predict water content at -330 cm matric potential head (FC) computed on TEST_BASIC and TEST_CHEM+ set. USSAND: sand (50–2000 μm) content (mass %); USSILT: silt (2–50 μm) content (mass %), USCLAY: clay (<2 μm) content (mass %); DEPTH_M: mean soil depth (cm); OC: organic carbon content (mass %); BD: bulk density (g cm$^{-3}$); CACO3: calcium carbonate content (mass %); PH_H2O: pH in water (-); CEC: cation exchange capacity (cmol (+) kg$^{-1}$).

[Figure]

**Figure S7.** Density plot of observed (OBS) and predicted median (PSD+DEPTH_M+*) water content at -330 cm matric potential head (FC) for selected pedotransfer functions, computed on TEST_BASIC and TEST_CHEM+ set. USSAND: sand (50–2000 μm) content (mass %); USSILT: silt (2–50 μm) content (mass %), USCLAY: clay (<2 μm) content (mass %); DEPTH_M: mean soil depth (cm); OC: organic carbon content (mass %); BD: bulk density (g cm$^{-3}$); CACO3: calcium carbonate content (mass %); PH_H2O: pH in water (-); CEC: cation exchange capacity (cmol (+) kg$^{-1}$).

[Figure]

[Figure]

**Figure S8.** Root mean sSquared error (RMSE) of the pedotransfer functions derived to predict water content at wilting point (WP) computed on TEST_BASIC and TEST_CHEM+ set. USSAND: sand (50–2000 µm) content (mass %); USSILT: silt (2–50 µm) content (mass %), USCLAY: clay (<2 µm) content (mass %); DEPTH_M: mean soil depth (cm); OC: organic carbon content (mass %); BD: bulk density (g cm$^{-3}$); CACO3: calcium carbonate content (mass %); PH_H2O: pH in water (-); CEC: cation exchange capacity (cmol (+) kg$^{-1}$).

[Figure]

**Figure S9.** Density plot of observed (OBS) and predicted median (PSD+DEPTH_M+*) water content at wilting point (WP) for selected pedotransfer functions, computed on TEST_BASIC and TEST_CHEM+ set. USSAND: sand (50–2000 µm) content (mass %); USSILT: silt (2–50 µm) content (mass %), USCLAY: clay (<2 µm) content (mass %); DEPTH_M: mean soil depth (cm); OC: organic carbon content (mass %); BD: bulk density (g cm$^{-3}$); CACO3: calcium carbonate content (mass %); PH_H2O: pH in water (-); CEC: cation exchange capacity (cmol (+) kg$^{-1}$).

[Figure]

**Figure S10.** Root mean sSquared error (RMSE) of the pedotransfer functions derived to predict plant available water content (AWC_2) considering field capacity at -100 matric potential head (FC_2), computed on TEST_BASIC and TEST_CHEM+ set. USSAND: sand (50–2000 μm) content (mass %); USSILT: silt (2–50 μm) content (mass %), USCLAY: clay (<2 μm) content (mass %); DEPTH_M: mean soil depth (cm); OC: organic carbon content (mass %); BD: bulk density (g cm$^{-3}$); CACO3: calcium carbonate content (mass %); PH_H2O: pH in water (-); CEC: cation exchange capacity (cmol (+) kg$^{-1}$).

[Figure]

**Figure S11.** Density plot of observed (OBS) and predicted median (PSD+DEPTH_M+*) plant available water content (AWC_2) considering field capacity at -100 matric potential head (FC_2) for selected pedotransfer functions, computed on TEST_BASIC and TEST_CHEM+ set. USSAND: sand (50–2000 μm) content (mass %); USSILT: silt (2–50 μm) content (mass %), USCLAY: clay (<2 μm) content (mass %); DEPTH_M: mean soil depth (cm); OC: organic carbon content (mass %); BD: bulk density (g cm$^{-3}$); CACO3: calcium carbonate content (mass %); PH_H2O: pH in water (-); CEC: cation exchange capacity (cmol (+) kg$^{-1}$).

[Figure]

[Figure]

**Figure S12.** Root mean sSquared error (RMSE) of the pedotransfer functions derived to predict plant available water content (AWC) considering field capacity at -330 matric potential head (FC), computed on TEST_BASIC and TEST_CHEM+ set. USSAND: sand (50–2000 μm) content (mass %); USSILT: silt (2–50 μm) content (mass %), USCLAY: clay (<2 μm) content (mass %); DEPTH_M: mean soil depth (cm); OC: organic carbon content (mass %); BD: bulk density (g cm$^{-3}$); CACO3: calcium carbonate content (mass %); PH_H2O: pH in water (-); CEC: cation exchange capacity (cmol (+) kg$^{-1}$).

[Figure]

**Figure S13.** Density plot of observed (OBS) and predicted median (PSD+DEPTH_M+*) plant available water content (AWC) considering field capacity at -330 matric potential head (FC) for selected pedotransfer functions, computed on TEST_BASIC and TEST_CHEM+ set. USSAND: sand (50–2000 μm) content (mass %); USSILT: silt (2–50 μm) content (mass %), USCLAY: clay (<2 μm) content (mass %); DEPTH_M: mean soil depth (cm); OC: organic carbon content (mass %); BD: bulk density (g cm$^{-3}$); CACO3: calcium carbonate content (mass %); PH_H2O: pH in water (-); CEC: cation exchange capacity (cmol (+) kg$^{-1}$).

[Figure]

**Figure S14.** Root mean sSquared error (RMSE) of the pedotransfer functions derived to predict saturated hydraulic conductivity (KS), computed on TEST_BASIC and TEST_CHEM+ set. USSAND: sand (50–2000 μm) content (mass %); USSILT: silt (2–50 μm) content (mass %), USCLAY: clay (<2 μm) content (mass %); DEPTH_M: mean soil depth (cm); OC: organic carbon content (mass %); BD: bulk density (g cm$^{-3}$); CACO3: calcium carbonate content (mass %); PH_H2O: pH in water (-); CEC: cation exchange capacity (cmol (+) kg$^{-1}$).

[Figure]

**Figure S15.** Density plot of observed (OBS) and predicted median (PSD+DEPTH_M+*) saturated hydraulic conductivity (KS) for selected pedotransfer functions, computed on TEST_BASIC and TEST_CHEM+ set. USSAND: sand (50–2000 μm) content (mass %); USSILT: silt (2–50 μm) content (mass %), USCLAY: clay (<2 μm) content (mass %); DEPTH_M: mean soil depth (cm); OC: organic carbon content (mass %); BD: bulk density (g cm$^{-3}$); CACO3: calcium carbonate content (mass %); PH_H2O: pH in water (-); CEC: cation exchange capacity (cmol (+) kg$^{-1}$).

[Figure]

[Figure]

**Figure S16.** Squared error (SE) of the pedotransfer functions derived to predict parameters of the van Genuchten model for the description of the moisture retention curve (MRC), computed on TEST_BASIC and TEST_CHEM+ set. USSAND: sand (50–2000 μm) content (mass %); USSILT: silt (2–50 μm) content (mass %), USCLAY: clay (<2 μm) content (mass %); DEPTH_M: mean soil depth (cm); OC: organic carbon content (mass %); BD: bulk density (g cm$^{-3}$); CACO3: calcium carbonate content (mass %); PH_H2O: pH in water (-); CEC: cation exchange capacity (cmol (+) kg$^{-1}$).

[Figure]

**Figure S17.** Density plot of observed (OBS) and predicted median (PSD+DEPTH_M+*) water retention values (MRC) computed based on the parameters of the van Genuchten model, computed on TEST_BASIC and TEST_CHEM+ set. Predicted values of those PTFs are shown which use the most often available predictor variables. USSAND: sand (50–2000 μm) content (mass %); USSILT: silt (2–50 μm) content (mass %), USCLAY: clay (<2 μm) content (mass %); DEPTH_M: mean soil depth (cm); OC: organic carbon content (mass %); BD: bulk density (g cm$^{-3}$); CACO3: calcium carbonate content (mass %); PH_H2O: pH in water (-); CEC: cation exchange capacity (cmol (+) kg$^{-1}$).

[Figure]

[Figure]

**Figure S18.** Mean error of the pedotransfer functions derived to predict parameters of the van Genuchten model for the description of the moisture retention curve, computed on TEST_BASIC (N = 1591) (A) and TEST_CHEM+ (N = 288) (B) sets by matric potential head values.

[Figure]

**Figure S19.** Root mean sSquared error (RMSE) of the pedotransfer functions derived to predict parameters of the Mualem-van Genuchten model for the description of the hydraulic conductivity curve (HCC), computed on TEST_BASIC and TEST_CHEM+ set. USSAND: sand (50–2000 μm) content (mass %); USSILT: silt (2–50 μm) content (mass %), USCLAY: clay (<2 μm) content (mass %); DEPTH_M: mean soil depth (cm); OC: organic carbon content (mass %); BD: bulk density (g cm$^{-3}$); CACO3: calcium carbonate content (mass %); PH_H2O: pH in water (-); CEC: cation exchange capacity (cmol (+) kg$^{-1}$).

[Figure]

**Figure S20.** Density plot of observed (OBS) and predicted median (PSD+DEPTH_M+*) hydraulic conductivity values (HCC) computed based on the parameters of the Mualem-van Genuchten model, computed on TEST_BASIC and TEST_CHEM+ set. Predicted values of those PTFs are shown which use the most often available predictor variables. USSAND: sand (50–2000 µm) content (mass %); USSILT: silt (2–50 µm) content (mass %), USCLAY: clay (<2 µm) content (mass %); DEPTH_M: mean soil depth (cm); OC: organic carbon content (mass %); BD: bulk density (g cm$^{-3}$); CACO3: calcium carbonate content (mass %); PH_H2O: pH in water (-); CEC: cation exchange capacity (cmol (+) kg$^{-1}$).

[Figure]

[Figure]

**Figure S21.** Mean error of the pedotransfer functions derived to predict parameters of the Mualem-van Genuchten model for the description of the hydraulic conductivity curve, computed on TEST_BASIC (N = 176) (A) and TEST_CHEM+ (N = 57) (B) sets by matric potential head values.

|  |  | | | | | | | | |
|---|---|---|---|---|---|---|---|---|---|
| |  |  |  |  |  |  |  |  |  |
|  |  |  |  |  |  |  |  |  |  |
|  |  |  |  |  |  |  |  |  |  |

| | | | | | | | | | |
|---|---|---|---|---|---|---|---|---|---|
| PSD+DEPTH_M+BD | PTF0 | PTF0 | PTF0 | PTF0 | PTF0 | PTF01 | PTF0 | PTF0 | PTF0 |
| PSD+DEPTH_M+CACO3 | PTF0 | PTF0 | PTF0 | PTF0 | PTF0 | PTF01 | PTF0 | PTF0 | PTF0 |
| PSD+DEPTH_M+PH_H2O | PTF0 | PTF0 | PTF0 | PTF0 | PTF0 | PTF01 | PTF0 | PTF0 | PTF0 |
| PSD+DEPTH_M+CEC | PTF0 | PTF0 | PTF0 | PTF0 | PTF0 | PTF01 | PTF0 | PTF0 | PTF0 |
| PSD+DEPTH_M+OC+BD | PTF0 | PTF0 | PTF0 | PTF0 | PTF0 | PTF03 | PTF0 | PTF0 | PTF0 |
| PSD+DEPTH_M+OC+CACO3 | PTF0 | PTF0 | PTF0 | PTF0 | PTF0 | PTF01 | PTF0 | PTF0 | PTF0 |
| PSD+DEPTH_M+OC+PH_H2O | PTF0 | PTF0 | PTF0 | PTF0 | PTF0 | PTF01 | PTF0 | PTF0 | PTF0 |
| PSD+DEPTH_M+OC+CEC | PTF0 | PTF0 | PTF0 | PTF0 | PTF0 | PTF01 | PTF0 | PTF1 | PTF0 |
| PSD+DEPTH_M+BD+CACO3 | PTF0 | PTF0 | PTF0 | PTF0 | PTF0 | PTF01 | PTF0 | PTF1 | PTF0 |
| PSD+DEPTH_M+BD+PH_H2O | PTF0 | PTF0 | PTF0 | PTF0 | PTF0 | PTF01 | PTF0 | PTF1 | PTF1 |
| PSD+DEPTH_M+BD+CEC | PTF0 | PTF1 | PTF0 | PTF0 | PTF0 | PTF01 | PTF0 | PTF1 | PTF1 |
| PSD+DEPTH_M+CACO3+PH_H2O | PTF0 | PTF0 | PTF0 | PTF0 | PTF0 | PTF01 | PTF0 | PTF1 | PTF0 |
| PSD+DEPTH_M+CACO3+CEC | PTF0 | PTF0 | PTF0 | PTF0 | PTF0 | PTF01 | PTF0 | PTF1 | PTF0 |
| PSD+DEPTH_M+PH_H2O+CEC | PTF0 | PTF0 | PTF0 | PTF0 | PTF0 | PTF01 | PTF0 | PTF1 | PTF0 |
| PSD+DEPTH_M+OC+BD+CACO3 | PTF0 | PTF0 | PTF0 | PTF0 | PTF0 | PTF03 | PTF0 | PTF1 | PTF0 |
| PSD+DEPTH_M+OC+BD+PH_H2O | PTF0 | PTF0 | PTF0 | PTF0 | PTF0 | PTF03 | PTF0 | PTF1 | PTF0 |
| PSD+DEPTH_M+OC+BD+CEC | PTF0 | PTF1 | PTF0 | PTF0 | PTF0 | PTF03 | PTF0 | PTF1 | PTF0 |
| PSD+DEPTH_M+OC+CACO3+PH_H2O | PTF0 | PTF0 | PTF0 | PTF0 | PTF0 | PTF01 | PTF0 | PTF0 | PTF2 |
| PSD+DEPTH_M+OC+CACO3+CEC | PTF0 | PTF0 | PTF0 | PTF0 | PTF0 | PTF01 | PTF0 | PTF2 | PTF2 |
| PSD+DEPTH_M+OC+PH_H2O+CEC | PTF0 | PTF0 | PTF0 | PTF0 | PTF0 | PTF01 | PTF0 | PTF2 | PTF2 |
| PSD+DEPTH_M+BD+CACO3+PH_H2O | PTF0 | PTF0 | PTF0 | PTF0 | PTF0 | PTF01 | PTF0 | PTF1 | PTF2 |
| PSD+DEPTH_M+BD+CACO3+CEC | PTF0 | PTF1 | PTF1 | PTF0 | PTF0 | PTF01 | PTF0 | PTF2 | PTF0 |
| PSD+DEPTH_M+BD+PH_H2O+CEC | PTF0 | PTF1 | PTF1 | PTF0 | PTF0 | PTF01 | PTF0 | PTF2 | PTF1 |
| PSD+DEPTH_M+CACO3+PH_H2O+CEC | PTF0 | PTF0 | PTF1 | PTF0 | PTF0 | PTF01 | PTF0 | PTF1 | PTF0 |
| PSD+DEPTH_M+OC+BD+CACO3+PH_H2O | PTF0 | PTF0 | PTF0 | PTF0 | PTF0 | PTF03 | PTF0 | PTF1 | PTF2 |
| PSD+DEPTH_M+OC+BD+CACO3+CEC | PTF0 | PTF1 | PTF0 | PTF1 | PTF0 | PTF03 | PTF0 | PTF2 | PTF2 |
| PSD+DEPTH_M+OC+BD+PH_H2O+CEC | PTF0 | PTF1 | PTF0 | PTF0 | PTF0 | PTF03 | PTF0 | PTF2 | PTF2 |
| PSD+DEPTH_M+OC+CACO3+PH_H2O+CE | PTF0 | PTF0 | PTF0 | PTF0 | PTF0 | PTF01 | PTF0 | PTF2 | PTF2 |
| PSD+DEPTH_M+BD+CACO3+PH_H2O+CE | PTF0 | PTF0 | PTF0 | PTF0 | PTF0 | PTF01 | PTF0 | PTF2 | PTF1 |
| PSD+DEPTH_M+OC+BD+CACO3+PH_H2O | PTF0 | PTF1 | PTF0 | PTF0 | PTF0 | PTF03 | PTF0 | PTF2 | PTF2 |

[1] PSD: particle size distribution (sand, 50–2000 μm; silt, 2–50 μm; clay, <2 μm (mass %)); DEPTH: mean soil depth (cm); OC: organic carbon content (mass %); BD: bulk density (g cm⁻³); CACO3: calcium carbonate content (mass %); PH_H2O: pH in water ( ); CEC: cation exchange capacity (cmol (+) kg⁻¹).
[2] THS: saturated water content (pF 0); FC_2: water content at –100 cm matric potential head (pF 2.0); FC: water content at –330 cm matric potential head (pF 2.5); AWC_2: plant available water content based on FC_2; AWC: plant available water content based on FC; WP: water content at wilting point (pF 4.2);
5 KS: saturated hydraulic conductivity; VG: parameters of the van Genuchten model; MVG: parameters of the Mualem—van Genuchten model; TEST_BASIC: samples with measured PSD, DEPTH, OC and BD; TEST_CHEM+: samples with measured PSD, DEPTH, OC, BD, CACO3, PH_H2O and CEC.

---

## Author Response (AR2)

**Revision of manuscript on "Updated European hydraulic pedotransfer functions with communicated uncertainties in the predicted variables (euptfv2)"**

**Content of the document**

**I. POINT BY POINT AUTHORS' RESPONSE TO THE EDITOR AND REVIEWER#3**

**1. RESPONSE TO THE EDITOR**

Thank you for accepting our changes and your suggestion for providing more information related to uncertainty. We addressed the comment in a revised version of the article. Below we give details on exactly how we addressed the concern raised. Please note the following during reading the response:

- the responses are in blue regular font and follow the questions,
- new text parts that were added to the manuscript are in blue italic font,
- the reference to the lines (L) and pages (P) relates to the marked up version of the manuscript available in III. MARKED-UP MANUSCRIPT AND SUPPLEMENTARY MATERIAL VERSION section of this document.

**Editor:**

in addition to the editorial changes, I would like to raise one further point: After re-reading the paper I realized that there is a bit of a mismatch between mentioning the capability of the model to provide uncertainties in the introduction and title, and the actual content of the results section where this aspect is largely underrepresented. Therefore, I would suggest you to add at least one or two paragraphs that describe how uncertainties are derived by the model and that provide examples how this uncertainty prediction manifests in your test cases.

AGREED

    A:     Thank you for highlighting it. We added the following to be more specific about the uncertainty:

        P6 L4-9, information on how uncertainties are derived by the model:
        *"To quantify the prediction uncertainties, quantile regression was used (Meinshausen, 2006). In random forest, as implemented in ranger, it is called quantile regression forest. For each node in each tree, the quantile regression forest not only keeps the*

*mean of the predicted target variable, but all observations that belong to that node from which the full conditional distribution of the predicted variable is estimated. The width of the prediction interval varies with the predictor variables. The smaller the range of the prediction interval, the more accurate the prediction is."*

P7 L27-31, how the uncertainty prediction manifests in the test cases:
*"The largest reduction in the width of the inner 90% of the prediction interval is visible for THS. Specifically this value decreased from 0.21 to 0.10 $cm^3$ $cm^{-3}$ for THS, from 0.19 to 0.14 $cm^3$ $cm^{-3}$ for FC_2, from 0.17 to 0.14 $cm^3$ $cm^{-3}$ for FC, from 0.15 to 0.14 $cm^3$ $cm^{-3}$ for WP, from 0.19 to 0.17 $cm^3$ $cm^{-3}$ for AWC_2, from 4.1 to 3.2 $log_{10}$ ($cm$ $day^{-1}$) for KS. In the case of AWC the mean 90 % mean interval did not change (0.15 $cm^3$ $cm^{-3}$)."*

**2. RESPONSE TO REFEREE #3**

Thank you for accepting the revised version of our manuscript and your help to further improve the English of the text. Below we give details on exactly how we addressed those. Please note the following during reading the responses:
- the responses are in blue regular font and follow the referee's questions (RC3),
- new text parts that were added to the manuscript are in blue italic font,
- the reference to the lines (L) and pages (P) relates to the marked up version of the manuscript available in III. MARKED-UP MANUSCRIPT AND SUPPLEMENTARY MATERIAL VERSION section of this document.

**RC3:**
L8. P3. a quantification of, "a" is redundant
AGREED
  A:  We corrected it (P3 L8).

above mentioned misses a hyphen
AGREED
  A:  We corrected it (P3 L8).

L20. P8 changes in OC "do" not influence
AGREED
  A:  We corrected it (P8 L30).

L6 .P9 improves "the" prediction of FC
AGREED
  A:  We corrected it (P9 L15).

L27. P9. nor CEC improves the …
AGREED
  A:  We corrected it (P10 L6).

L3. P10. recommend computing + L4. P13
AGREED
  A:  We corrected it (P10 L14 and P13 L14-15).

L8. P10. on "the" largest
AGREED
  A:  We corrected it (P10 L19).

L18. P10. "the" highest
AGREED
  A:  We corrected it (P10 L29).

L29. P10. ..., otherwise, ...
AGREED
  A:  We corrected it (P11 L7).

L6. P11. For the prediction of ...
AGREED
  A:  We corrected it (P11 L16).

L9. P11. some of which are not direction. This looks vague!
AGREED
  A:  We deleted it (P11 L19).

L12. P11. and, moreover, is... (something is missing, and, moreover..)
AGREED
  A:  Moreover was wrongly written twice in the sentence, we deleted the one which was unnecessary one (P11 L21).

L4. P12. significantly improves...
AGREED
  A:  We corrected it (P12 L14).

L10. P12. information on ...
AGREED
  A:  We corrected it (P12 L20).

L 16. P12. An increase ...
AGREED
  A:  We corrected it (P12 L26).

L19. P12. was identical ...
AGREED
  A:  We corrected it (P12 L29).

L 24. P12. with this, ... in the dry end will lead ...
AGREED
  A:  We corrected it (P13 L1).

L29. P12. ... further 8 cases, RMSE was ...
AGREED
  A:  We corrected it (P13 L7).

L8. P13. In the remaining 5 cases, ...
AGREED
  A:  We corrected it (P13 L18).

L13. P13. a reduced ...
AGREED

A:   We corrected it (P13 L23).

L 26. P13. reformulate the below part
soil depth needs to be considered in regard to the depth of the other input properties and soil hydraulic data needs, e.g. if the soil hydraulic properties of the top 20 cm (0-20 cm) is needed,
AGREED

A:   We rephrased it:
P14 L5-9:
*"Soil depth is defined as the mean sampling depth, if e.g. PSD, BD and OC are provided for a soil sample from a depth of 0-20 cm, then the soil depth input (DEPTH) to the prediction algorithm is set to 10 cm."*

L 19. 14. the core ... provide ...
Nothing changed.

A:   In P14 L29-30 we meant the following: The "Code and data availability" section provides information on how to access this resource.
We thought that this sentence might not need the suggested correction.

**II. LIST OF AUTHOR'S CHANGES IN MANUSCRIPT AND SUPPLEMENTARY MATERIAL**

Page and line numbering refer to that of the revised manuscript with track changes included under III. MARKED-UP MANUSCRIPT VERSION AND SUPPLEMENTARY MATERIAL VERSION section of this document.

**The following changes have been made in the manuscript:**
- P1 L10: e-mail address was corrected, it was needed due to organizational changes,
- P3 L8: "a" deleted and hyphen added,
- P6 L4-10: text rephrased and added to provide more information on how uncertainties are derived,
- P7 L24: enter was added,
- P7 L26: "computed on the test sets" added to specify information provided in Figures 2 and S1,
- P7 L27-31: information on how the uncertainty prediction manifests in the test cases added,
- P8 L30: "does" corrected to "do",
- P9 L15: "the" added,
- P10 L6: "significantly" deleted, "improve" corrected to "improves",
- P10 L14: language correction to "recommend computing",
- P10 L19: "the" added,
- P10 L29: "the" added,
- P11 L7: "," added,
- P11 L16: "the" added,
- P11 L19: following is deleted: ", some of which are not direction",
- P11 L21: following is deleted: ", moreover,",
- P11 L22: "," deleted,

- P12 L14: language correction to "improves",
- P12 L20: "of" changed to "on",
- P12 L26: language correction to "An increase",
- P12 L29: "were" changed to "was",
- P13 L1: "," deleted,
- P13 L7: language correction to "8 cases, RMSE was",
- P13 L14-15: language correction to "recommend computing",
- P13 L18: "," added,
- P13 L23: "a" added,
- P14 L5-9: sentence rephrased,
- P32: size of the page change according to the Copernicus template.

**The following changes have been made in the supplementary material:**
- P1 L6 e-mail address was corrected, it was needed due to organizational changes.

**III. MARKED-UP MANUSCRIPT AND SUPPLEMENTARY MATERIAL VERSION**

Please find revised marked-up manuscript and supplementary material on the following pages.

[revised manuscript text omitted]

* Correspondence: toth.brigitta@atk.hu (B.Sz.)

[Figure]

**Figure S1.** The scatter plot of the measured versus predicted plant available water content values of the worst and best performing PTF with 90% prediction interval on test datasets. AWC_2: plant available water content based on filed capacity at -100 cm matric potential head (PTF01 vs. PTF03); AWC: plant available water content based on filed capacity at -330 cm matric potential head (PTF01 vs. PTF03); PSD: particle size distribution (sand, 50–2000 μm; silt, 2–50 μm; clay, <2 μm (mass %)); DEPTH_M: mean soil depth (cm); BD: bulk density (g cm$^{-3}$); Count: the number of cases in each rectangle.

**Table S1.** Performance of pedotransfer functions (PTF) by input combination on training and test datasets to predict the plant available water content of the soil (AWC_2) belonging to the -100 cm matric potential head. N: number of samples, RMSE: root mean square error ($cm^3$ $cm^{-3}$), and $R^2$: determination coefficient, TEST_BASIC: samples with measured PSD, DEPTH, OC and BD; TEST_CHEM+: samples with measured PSD, DEPTH, OC, BD, CACO3, PH_H2O and CEC. Recommended PTFs are highlighted in bold.

| Name of PTF in euptfv2 | Predictor variables[1] | Training set N | RMSE | $R^2$ | Test set N | RMSE | $R^2$ | TEST_BASIC set | TEST_CHEM+ set | Recommended PTF |
|---|---|---|---|---|---|---|---|---|---|---|
| **PTF01** | **PSD+DEPTH** | 3528 | 0.062 | 0.446 | 1372 | 0.060 | 0.432 | a | ab | PTF01 |
| **PTF02** | **PSD+DEPTH+OC** | 3208 | 0.055 | 0.540 | 1372 | 0.054 | 0.544 | b | abcd | PTF02 |
| **PTF03** | **PSD+DEPTH+BD** | 3472 | 0.054 | 0.581 | 1372 | 0.053 | 0.552 | b | abcd | PTF03 |
| PTF04 | PSD+DEPTH+CACO3 | 1548 | 0.050 | 0.326 | 274 | 0.055 | 0.219 | - | abcd | PTF01 |
| PTF05 | PSD+DEPTH+PH_H2O | 1849 | 0.058 | 0.463 | 274 | 0.055 | 0.216 | - | a | PTF01 |
| PTF06 | PSD+DEPTH+CEC | 1550 | 0.059 | 0.512 | 274 | 0.060 | 0.050 | - | abcd | PTF01 |
| PTF07 | PSD+DEPTH+OC+BD | 3197 | 0.051 | 0.609 | 1372 | 0.051 | 0.588 | b | abcd | PTF03 |
| PTF08 | PSD+DEPTH+OC+CACO3 | 1464 | 0.048 | 0.353 | 274 | 0.053 | 0.257 | - | abcd | PTF02 |
| PTF09 | PSD+DEPTH+OC+PH_H2O | 1615 | 0.055 | 0.490 | 274 | 0.053 | 0.270 | - | abc | PTF02 |
| PTF10 | PSD+DEPTH+OC+CEC | 1358 | 0.054 | 0.563 | 274 | 0.053 | 0.278 | - | abcd | PTF02 |
| PTF11 | PSD+DEPTH+BD+CACO3 | 1545 | 0.044 | 0.470 | 274 | 0.048 | 0.396 | - | d | PTF03 |
| PTF12 | PSD+DEPTH+BD+PH_H2O | 1796 | 0.052 | 0.565 | 274 | 0.048 | 0.406 | - | abcd | PTF03 |
| PTF13 | PSD+DEPTH+BD+CEC | 1498 | 0.053 | 0.598 | 274 | 0.048 | 0.398 | - | abcd | PTF03 |
| PTF14 | PSD+DEPTH+CACO3+PH_H2O | 1195 | 0.051 | 0.341 | 274 | 0.052 | 0.284 | - | abcd | PTF01 |
| PTF15 | PSD+DEPTH+CACO3+CEC | 726 | 0.050 | 0.286 | 274 | 0.052 | 0.303 | - | abcd | PTF01 |
| PTF16 | PSD+DEPTH+PH_H2O+CEC | 1255 | 0.058 | 0.539 | 274 | 0.051 | 0.331 | - | abcd | PTF01 |
| PTF17 | PSD+DEPTH+OC+BD+CACO3 | 1464 | 0.044 | 0.465 | 274 | 0.048 | 0.390 | - | bcd | PTF03 |
| PTF18 | PSD+DEPTH+OC+BD+PH_H2O | 1607 | 0.051 | 0.556 | 274 | 0.048 | 0.407 | - | abcd | PTF03 |
| PTF19 | PSD+DEPTH+OC+BD+CEC | 1349 | 0.052 | 0.593 | 274 | 0.046 | 0.441 | - | abcd | PTF03 |
| PTF20 | PSD+DEPTH+OC+CACO3+PH_H2O | 1130 | 0.050 | 0.367 | 274 | 0.051 | 0.309 | - | abcd | PTF02 |
| PTF21 | PSD+DEPTH+OC+CACO3+CEC | 683 | 0.049 | 0.305 | 274 | 0.050 | 0.359 | - | abcd | PTF02 |
| PTF22 | PSD+DEPTH+OC+PH_H2O+CEC | 1067 | 0.054 | 0.561 | 274 | 0.049 | 0.367 | - | abcd | PTF02 |
| PTF23 | PSD+DEPTH+BD+CACO3+PH_H2O | 1192 | 0.046 | 0.471 | 274 | 0.049 | 0.375 | - | bcd | PTF03 |
| PTF24 | PSD+DEPTH+BD+CACO3+CEC | 725 | 0.045 | 0.420 | 274 | 0.046 | 0.444 | - | d | PTF03 |
| PTF25 | PSD+DEPTH+BD+PH_H2O+CEC | 1204 | 0.052 | 0.621 | 274 | 0.046 | 0.456 | - | abcd | PTF03 |
| PTF26 | PSD+DEPTH+CACO3+PH_H2O+CEC | 684 | 0.049 | 0.318 | 274 | 0.048 | 0.388 | - | abcd | PTF01 |
| PTF27 | PSD+DEPTH+OC+BD+CACO3+PH_H2O | 1130 | 0.045 | 0.475 | 274 | 0.049 | 0.367 | - | abcd | PTF03 |
| PTF28 | PSD+DEPTH+OC+BD+CACO3+CEC | 683 | 0.045 | 0.408 | 274 | 0.045 | 0.466 | - | bcd | PTF03 |
| PTF29 | PSD+DEPTH+OC+BD+PH_H2O+CEC | 1059 | 0.052 | 0.603 | 274 | 0.045 | 0.473 | - | bcd | PTF03 |
| PTF30 | PSD+DEPTH+OC+CACO3+PH_H2O+CEC | 641 | 0.049 | 0.330 | 274 | 0.048 | 0.393 | - | abcd | PTF02 |
| PTF31 | PSD+DEPTH+BD+CACO3+PH_H2O+CEC | 683 | 0.044 | 0.450 | 274 | 0.045 | 0.480 | - | cd | PTF03 |
| PTF32 | PSD+DEPTH+OC+BD+CACO3+PH_H2O+CEC | 641 | 0.045 | 0.425 | 274 | 0.045 | 0.471 | - | cd | PTF03 |

[1]PSD: particle size distribution (sand, 50–2000 μm; silt, 2–50 μm; clay, <2 μm (mass %)); DEPTH: mean soil depth (cm); OC: organic carbon content (mass %); BD: bulk density (g $cm^{-3}$); CACO3: calcium carbonate content (mass %); PH_H2O: pH in water (-); CEC: cation exchange capacity (cmol (+) $kg^{-1}$).
[2]Different letters indicate significant differences at the 0.05 level between the accuracy of the methods based on the squared error; for example performance indicated with the letter c is significantly better than the one noted with letters b and a.

**Table S2.** Performance of pedotransfer functions (PTF) by input combination on training and test datasets to predict the plant available water content of the soil (AWC) belonging to the -330 cm matric potential head. N: number of samples, RMSE: root mean square error ($cm^3\ cm^{-3}$), and $R^2$: determination coefficient, TEST_BASIC: samples with measured PSD, DEPTH, OC and BD; TEST_CHEM+: samples with measured PSD, DEPTH, OC, BD, CACO3, PH_H2O and CEC. Recommended PTFs are highlighted in bold.

| Name of PTF in euptfv2 | Predictor variables[1] | Training set | | | Test set | | | Sign. difference[2] | | Recommended PTF |
|---|---|---|---|---|---|---|---|---|---|---|
| | | N | RMSE | $R^2$ | N | RMSE | $R^2$ | TEST_BASIC set | TEST_CHEM+ set | |
| **PTF01** | **PSD+DEPTH** | 1863 | 0.042 | 0.312 | 705 | 0.048 | 0.196 | a | a | PTF01 |
| PTF02 | PSD+DEPTH+OC | 1650 | 0.041 | 0.337 | 705 | 0.045 | 0.288 | ab | a | PTF01 |
| **PTF03** | **PSD+DEPTH+BD** | 1849 | 0.040 | 0.374 | 705 | 0.045 | 0.285 | ab | a | PTF01 |
| PTF04 | PSD+DEPTH+CACO3 | 1531 | 0.040 | 0.366 | 279 | 0.050 | 0.199 | - | a | PTF01 |
| PTF05 | PSD+DEPTH+PH_H2O | 1245 | 0.042 | 0.344 | 279 | 0.048 | 0.238 | - | a | PTF01 |
| PTF06 | PSD+DEPTH+CEC | 1092 | 0.041 | 0.356 | 279 | 0.053 | 0.078 | - | a | PTF01 |
| PTF07 | PSD+DEPTH+OC+BD | 1645 | 0.040 | 0.381 | 705 | 0.043 | 0.337 | b | a | PTF03 |
| PTF08 | PSD+DEPTH+OC+CACO3 | 1336 | 0.041 | 0.345 | 279 | 0.049 | 0.219 | - | a | PTF01 |
| PTF09 | PSD+DEPTH+OC+PH_H2O | 1074 | 0.042 | 0.345 | 279 | 0.048 | 0.242 | - | a | PTF01 |
| PTF10 | PSD+DEPTH+OC+CEC | 998 | 0.039 | 0.413 | 279 | 0.051 | 0.147 | - | a | PTF01 |
| PTF11 | PSD+DEPTH+BD+CACO3 | 1522 | 0.038 | 0.428 | 279 | 0.048 | 0.258 | - | a | PTF01 |
| PTF12 | PSD+DEPTH+BD+PH_H2O | 1236 | 0.039 | 0.429 | 279 | 0.047 | 0.287 | - | a | PTF01 |
| PTF13 | PSD+DEPTH+BD+CEC | 1088 | 0.038 | 0.429 | 279 | 0.049 | 0.231 | - | a | PTF01 |
| PTF14 | PSD+DEPTH+CACO3+PH_H2O | 1230 | 0.041 | 0.376 | 279 | 0.047 | 0.263 | - | a | PTF01 |
| PTF15 | PSD+DEPTH+CACO3+CEC | 791 | 0.041 | 0.366 | 279 | 0.049 | 0.214 | - | a | PTF01 |
| PTF16 | PSD+DEPTH+PH_H2O+CEC | 739 | 0.042 | 0.321 | 279 | 0.048 | 0.237 | - | a | PTF01 |
| PTF17 | PSD+DEPTH+OC+BD+CACO3 | 1334 | 0.039 | 0.399 | 279 | 0.048 | 0.262 | - | a | PTF03 |
| PTF18 | PSD+DEPTH+OC+BD+PH_H2O | 1072 | 0.040 | 0.393 | 279 | 0.047 | 0.293 | - | a | PTF03 |
| PTF19 | PSD+DEPTH+OC+BD+CEC | 995 | 0.038 | 0.432 | 279 | 0.049 | 0.223 | - | a | PTF03 |
| PTF20 | PSD+DEPTH+OC+CACO3+PH_H2O | 1059 | 0.042 | 0.362 | 279 | 0.047 | 0.289 | - | a | PTF01 |
| PTF21 | PSD+DEPTH+OC+CACO3+CEC | 707 | 0.041 | 0.358 | 279 | 0.049 | 0.229 | - | a | PTF01 |
| PTF22 | PSD+DEPTH+OC+PH_H2O+CEC | 660 | 0.041 | 0.339 | 279 | 0.048 | 0.253 | - | a | PTF01 |
| PTF23 | PSD+DEPTH+BD+CACO3+PH_H2O | 1221 | 0.039 | 0.442 | 279 | 0.047 | 0.267 | - | a | PTF01 |
| PTF24 | PSD+DEPTH+BD+CACO3+CEC | 788 | 0.039 | 0.405 | 279 | 0.047 | 0.269 | - | a | PTF01 |
| PTF25 | PSD+DEPTH+BD+PH_H2O+CEC | 736 | 0.039 | 0.402 | 279 | 0.046 | 0.307 | - | a | PTF01 |
| PTF26 | PSD+DEPTH+CACO3+PH_H2O+CEC | 732 | 0.040 | 0.405 | 279 | 0.048 | 0.254 | - | a | PTF01 |
| PTF27 | PSD+DEPTH+OC+BD+CACO3+PH_H2O | 1057 | 0.040 | 0.415 | 279 | 0.046 | 0.312 | - | a | PTF03 |
| PTF28 | PSD+DEPTH+OC+BD+CACO3+CEC | 705 | 0.040 | 0.383 | 279 | 0.047 | 0.277 | - | a | PTF03 |
| PTF29 | PSD+DEPTH+OC+BD+PH_H2O+CEC | 658 | 0.040 | 0.385 | 279 | 0.046 | 0.315 | - | a | PTF03 |
| PTF30 | PSD+DEPTH+OC+CACO3+PH_H2O+CEC | 653 | 0.040 | 0.395 | 279 | 0.047 | 0.274 | - | a | PTF01 |
| PTF31 | PSD+DEPTH+BD+CACO3+PH_H2O+CEC | 729 | 0.039 | 0.431 | 279 | 0.047 | 0.290 | - | a | PTF01 |
| PTF32 | PSD+DEPTH+OC+BD+CACO3+PH_H2O+CEC | 651 | 0.039 | 0.403 | 279 | 0.046 | 0.307 | - | a | PTF03 |

[1]PSD: particle size distribution (sand, 50–2000 μm; silt, 2–50 μm; clay, <2 μm (mass %)); DEPTH: mean soil depth (cm); OC: organic carbon content (mass %); BD: bulk density (g $cm^{-3}$); CACO3: calcium carbonate content (mass %); PH_H2O: pH in water (-); CEC: cation exchange capacity (cmol (+) $kg^{-1}$).
[2]Different letters indicate significant differences at the 0.05 level between the accuracy of the methods based on the squared error; for example performance indicated with the letter c is significantly better than the one noted with letters b and a.

**Table S3.** Normalized root mean square error (NRMSE) of the point predictions by soil hydraulic properties computed on the test datasets in cm$^3$ cm$^{-3}$ for water retention and $\log_{10}$ (cm day$^{-1}$) for saturated hydraulic conductivity. In case of PTF01, 02, 03 and 07 TEST_BASIC set was used for the analysis, for the rest of the PTFs TEST_CHEM+ set was considered.

| Name of PTF in euptfv2 | Predictor variables[1] | NRMSE in test sets[2] | | | | | | |
|---|---|---|---|---|---|---|---|---|
| | | THS | FC_2 | FC | WP | AWC_2 | AWC | KS |
| PTF01 | PSD+DEPTH_M | 0.104 | 0.090 | 0.082 | 0.105 | 0.126 | 0.140 | 0.17 |
| PTF02 | PSD+DEPTH_M+OC | 0.086 | 0.083 | 0.076 | 0.102 | 0.112 | 0.132 | 0.14 |
| PTF03 | PSD+DEPTH_M+BD | 0.048 | 0.079 | 0.074 | 0.100 | 0.111 | 0.132 | 0.17 |
| PTF04 | PSD+DEPTH_M+CACO3 | 0.191 | 0.107 | 0.113 | 0.122 | 0.164 | 0.145 | 0.19 |
| PTF05 | PSD+DEPTH_M+PH_H2O | 0.176 | 0.112 | 0.114 | 0.126 | 0.164 | 0.142 | 0.19 |
| PTF06 | PSD+DEPTH_M+CEC | 0.191 | 0.107 | 0.107 | 0.118 | 0.181 | 0.156 | 0.19 |
| PTF07 | PSD+DEPTH_M+OC+BD | 0.047 | 0.075 | 0.073 | 0.097 | 0.107 | 0.127 | 0.14 |
| PTF08 | PSD+DEPTH_M+OC+CACO3 | 0.184 | 0.097 | 0.109 | 0.117 | 0.160 | 0.143 | 0.19 |
| PTF09 | PSD+DEPTH_M+OC+PH_H2O | 0.167 | 0.095 | 0.107 | 0.119 | 0.158 | 0.141 | 0.18 |
| PTF10 | PSD+DEPTH_M+OC+CEC | 0.172 | 0.098 | 0.108 | 0.116 | 0.158 | 0.150 | 0.18 |
| PTF11 | PSD+DEPTH_M+BD+CACO3 | 0.072 | 0.091 | 0.105 | 0.115 | 0.144 | 0.140 | 0.19 |
| PTF12 | PSD+DEPTH_M+BD+PH_H2O | 0.069 | 0.086 | 0.103 | 0.117 | 0.143 | 0.137 | 0.19 |
| PTF13 | PSD+DEPTH_M+BD+CEC | 0.070 | 0.091 | 0.100 | 0.115 | 0.144 | 0.142 | 0.19 |
| PTF14 | PSD+DEPTH_M+CACO3+PH_H2O | 0.168 | 0.101 | 0.109 | 0.121 | 0.157 | 0.139 | 0.18 |
| PTF15 | PSD+DEPTH_M+CACO3+CEC | 0.179 | 0.102 | 0.106 | 0.113 | 0.155 | 0.144 | 0.19 |
| PTF16 | PSD+DEPTH_M+PH_H2O+CEC | 0.183 | 0.098 | 0.104 | 0.115 | 0.152 | 0.142 | 0.19 |
| PTF17 | PSD+DEPTH_M+OC+BD+CACO3 | 0.070 | 0.089 | 0.102 | 0.111 | 0.145 | 0.139 | 0.18 |
| PTF18 | PSD+DEPTH_M+OC+BD+PH_H2O | 0.070 | 0.083 | 0.103 | 0.116 | 0.143 | 0.136 | 0.18 |
| PTF19 | PSD+DEPTH_M+OC+BD+CEC | 0.070 | 0.087 | 0.099 | 0.113 | 0.139 | 0.143 | 0.18 |
| PTF20 | PSD+DEPTH_M+OC+CACO3+PH_H2O | 0.166 | 0.105 | 0.107 | 0.114 | 0.154 | 0.137 | 0.18 |
| PTF21 | PSD+DEPTH_M+OC+CACO3+CEC | 0.171 | 0.090 | 0.104 | 0.108 | 0.149 | 0.142 | 0.18 |
| PTF22 | PSD+DEPTH_M+OC+PH_H2O+CEC | 0.166 | 0.089 | 0.102 | 0.111 | 0.148 | 0.140 | 0.18 |
| PTF23 | PSD+DEPTH_M+BD+CACO3+PH_H2O | 0.071 | 0.089 | 0.104 | 0.116 | 0.147 | 0.139 | 0.18 |
| PTF24 | PSD+DEPTH_M+BD+CACO3+CEC | 0.071 | 0.085 | 0.099 | 0.110 | 0.138 | 0.139 | 0.19 |
| PTF25 | PSD+DEPTH_M+BD+PH_H2O+CEC | 0.067 | 0.084 | 0.100 | 0.112 | 0.137 | 0.135 | 0.19 |
| PTF26 | PSD+DEPTH_M+CACO3+PH_H2O+CEC | 0.163 | 0.094 | 0.103 | 0.111 | 0.145 | 0.140 | 0.18 |
| PTF27 | PSD+DEPTH_M+OC+BD+CACO3+PH_H2O | 0.072 | 0.086 | 0.101 | 0.111 | 0.148 | 0.135 | 0.18 |
| PTF28 | PSD+DEPTH_M+OC+BD+CACO3+CEC | 0.070 | 0.082 | 0.098 | 0.106 | 0.136 | 0.138 | 0.18 |
| PTF29 | PSD+DEPTH_M+OC+BD+PH_H2O+CEC | 0.068 | 0.083 | 0.095 | 0.109 | 0.135 | 0.134 | 0.18 |
| PTF30 | PSD+DEPTH_M+OC+CACO3+PH_H2O+CEC | 0.162 | 0.100 | 0.101 | 0.108 | 0.145 | 0.138 | 0.17 |
| PTF31 | PSD+DEPTH_M+BD+CACO3+PH_H2O+CEC | 0.070 | 0.081 | 0.097 | 0.108 | 0.134 | 0.137 | 0.18 |
| PTF32 | PSD+DEPTH_M+OC+BD+CACO3+PH_H2O+CEC | 0.069 | 0.079 | 0.097 | 0.107 | 0.135 | 0.135 | 0.18 |

[1]PSD: particle size distribution (sand, 50–2000 μm; silt, 2–50 μm; clay, <2 μm (mass %)); DEPTH: mean soil depth (cm); OC: organic carbon content (mass %); BD: bulk density (g cm$_{-3}$); CACO3: calcium carbonate content (mass %); PH_H2O: pH in water (-); CEC: cation exchange capacity (cmol (+) kg$^{-1}$).

[2]THS: saturated water content (pF 0); FC_2: water content at -100 cm matric potential head (pF 2.0); FC: water content at -330 cm matric potential head (pF 2.5); AWC_2: plant available water content based on FC_2; AWC: plant available water content based on FC; WP: water content at wilting point (pF 4.2); KS: saturated hydraulic conductivity;

[Figure]

**Figure S2.** Root mean square error (RMSE) of the pedotransfer functions derived to predict water content at saturation (THS) computed on TEST_BASIC and TEST_CHEM+ set. USSAND: sand (50–2000 μm) content (mass %); USSILT: silt (2–50 μm) content (mass %), USCLAY: clay (<2 μm) content (mass %); DEPTH_M: mean soil depth (cm); OC: organic carbon content (mass %); BD: bulk density (g cm$^{-3}$); CACO3: calcium carbonate content (mass %); PH_H2O: pH in water (-); CEC: cation exchange capacity (cmol (+) kg$^{-1}$).

[Figure]

**Figure S3.** Density plot of observed (OBS) and predicted median (PSD+DEPTH_M+*) water content at saturation (THS) for selected pedotransfer functions, computed on TEST_BASIC and TEST_CHEM+ set. USSAND: sand (50–2000 μm) content (mass %); USSILT: silt (2–50 μm) content (mass %), USCLAY: clay (<2 μm) content (mass %); DEPTH_M: mean soil depth (cm); OC: organic carbon content (mass %); BD: bulk density (g cm$^{-3}$); CACO3: calcium carbonate content (mass %); PH_H2O: pH in water (-); CEC: cation exchange capacity (cmol (+) kg$^{-1}$).

[Figure]

**Figure S4.** Root mean square error (RMSE) of the pedotransfer functions derived to predict water content at -100 cm matric potential head (FC_2) computed on TEST_BASIC and TEST_CHEM+ set. USSAND: sand (50–2000 μm) content (mass %); USSILT: silt (2–50 μm) content (mass %), USCLAY: clay (<2 μm) content (mass %); DEPTH_M: mean soil depth (cm); OC: organic carbon content (mass %); BD: bulk density (g cm$^{-3}$); CACO3: calcium carbonate content (mass %); PH_H2O: pH in water (-); CEC: cation exchange capacity (cmol (+) kg$^{-1}$).

[Figure]

**Figure S5.** Density plot of observed (OBS) and predicted median (PSD+DEPTH_M+*) water content at -100 cm matric potential head (FC_2) for selected pedotransfer functions, computed on TEST_BASIC and TEST_CHEM+ set. USSAND: sand (50–2000 μm) content (mass %); USSILT: silt (2–50 μm) content (mass %), USCLAY: clay (<2 μm) content (mass %); DEPTH_M: mean soil depth (cm); OC: organic carbon content (mass %); BD: bulk density (g cm$^{-3}$); CACO3: calcium carbonate content (mass %); PH_H2O: pH in water (-); CEC: cation exchange capacity (cmol (+) kg$^{-1}$).

[Figure]

**Figure S6.** Root mean square error (RMSE) of the pedotransfer functions derived to predict water content at -330 cm matric potential head (FC) computed on TEST_BASIC and TEST_CHEM+ set. USSAND: sand (50–2000 μm) content (mass %); USSILT: silt (2–50 μm) content (mass %), USCLAY: clay (<2 μm) content (mass %); DEPTH_M: mean soil depth (cm); OC: organic carbon content (mass %); BD: bulk density (g cm$^{-3}$); CACO3: calcium carbonate content (mass %); PH_H2O: pH in water (-); CEC: cation exchange capacity (cmol (+) kg$^{-1}$).

[Figure]

**Figure S7.** Density plot of observed (OBS) and predicted median (PSD+DEPTH_M+*) water content at -330 cm matric potential head (FC) for selected pedotransfer functions, computed on TEST_BASIC and TEST_CHEM+ set. USSAND: sand (50–2000 μm) content (mass %); USSILT: silt (2–50 μm) content (mass %), USCLAY: clay (<2 μm) content (mass %); DEPTH_M: mean soil depth (cm); OC: organic carbon content (mass %); BD: bulk density (g cm$^{-3}$); CACO3: calcium carbonate content (mass %); PH_H2O: pH in water (-); CEC: cation exchange capacity (cmol (+) kg$^{-1}$).

[Figure]

**Figure S8.** Root mean square error (RMSE) of the pedotransfer functions derived to predict water content at wilting point (WP) computed on TEST_BASIC and TEST_CHEM+ set. USSAND: sand (50–2000 μm) content (mass %); USSILT: silt (2–50 μm) content (mass %), USCLAY: clay (<2 μm) content (mass %); DEPTH_M: mean soil depth (cm); OC: organic carbon content (mass %); BD: bulk density (g cm$^{-3}$); CACO3: calcium carbonate content (mass %); PH_H2O: pH in water (-); CEC: cation exchange capacity (cmol (+) kg$^{-1}$).

[Figure]

**Figure S9.** Density plot of observed (OBS) and predicted median (PSD+DEPTH_M+*) water content at wilting point (WP) for selected pedotransfer functions, computed on TEST_BASIC and TEST_CHEM+ set. USSAND: sand (50–2000 μm) content (mass %); USSILT: silt (2–50 μm) content (mass %), USCLAY: clay (<2 μm) content (mass %); DEPTH_M: mean soil depth (cm); OC: organic carbon content (mass %); BD: bulk density (g cm$^{-3}$); CACO3: calcium carbonate content (mass %); PH_H2O: pH in water (-); CEC: cation exchange capacity (cmol (+) kg$^{-1}$).

[Figure]

**Figure S10.** Root mean square error (RMSE) of the pedotransfer functions derived to predict plant available water content (AWC_2) considering field capacity at -100 matric potential head (FC_2), computed on TEST_BASIC and TEST_CHEM+ set. USSAND: sand (50–2000 μm) content (mass %); USSILT: silt (2–50 μm) content (mass %), USCLAY: clay (<2 μm) content (mass %); DEPTH_M: mean soil depth (cm); OC: organic carbon content (mass %); BD: bulk density (g cm$^{-3}$); CACO3: calcium carbonate content (mass %); PH_H2O: pH in water (-); CEC: cation exchange capacity (cmol (+) kg$^{-1}$).

[Figure]

**Figure S11.** Density plot of observed (OBS) and predicted median (PSD+DEPTH_M+*) plant available water content (AWC_2) considering field capacity at -100 matric potential head (FC_2) for selected pedotransfer functions, computed on TEST_BASIC and TEST_CHEM+ set. USSAND: sand (50–2000 μm) content (mass %); USSILT: silt (2–50 μm) content (mass %), USCLAY: clay (<2 μm) content (mass %); DEPTH_M: mean soil depth (cm); OC: organic carbon content (mass %); BD: bulk density (g cm$^{-3}$); CACO3: calcium carbonate content (mass %); PH_H2O: pH in water (-); CEC: cation exchange capacity (cmol (+) kg$^{-1}$).

[Figure]

**Figure S12.** Root mean square error (RMSE) of the pedotransfer functions derived to predict plant available water content (AWC) considering field capacity at -330 matric potential head (FC), computed on TEST_BASIC and TEST_CHEM+ set. USSAND: sand (50–2000 μm) content (mass %); USSILT: silt (2–50 μm) content (mass %), USCLAY: clay (<2 μm) content (mass %); DEPTH_M: mean soil depth (cm); OC: organic carbon content (mass %); BD: bulk density (g cm$^{-3}$); CACO3: calcium carbonate content (mass %); PH_H2O: pH in water (-); CEC: cation exchange capacity (cmol (+) kg$^{-1}$).

[Figure]

**Figure S13.** Density plot of observed (OBS) and predicted median (PSD+DEPTH_M+*) plant available water content (AWC) considering field capacity at -330 matric potential head (FC) for selected pedotransfer functions, computed on TEST_BASIC and TEST_CHEM+ set. USSAND: sand (50–2000 μm) content (mass %); USSILT: silt (2–50 μm) content (mass %), USCLAY: clay (<2 μm) content (mass %); DEPTH_M: mean soil depth (cm); OC: organic carbon content (mass %); BD: bulk density (g cm$^{-3}$); CACO3: calcium carbonate content (mass %); PH_H2O: pH in water (-); CEC: cation exchange capacity (cmol (+) kg$^{-1}$).

[Figure]

**Figure S14.** Root mean square error (RMSE) of the pedotransfer functions derived to predict saturated hydraulic conductivity (KS), computed on TEST_BASIC and TEST_CHEM+ set. USSAND: sand (50–2000 µm) content (mass %); USSILT: silt (2–50 µm) content (mass %), USCLAY: clay (<2 µm) content (mass %); DEPTH_M: mean soil depth (cm); OC: organic carbon content (mass %); BD: bulk density (g cm$^{-3}$); CACO3: calcium carbonate content (mass %); PH_H2O: pH in water (-); CEC: cation exchange capacity (cmol (+) kg$^{-1}$).

[Figure]

**Figure S15.** Density plot of observed (OBS) and predicted median (PSD+DEPTH_M+*) saturated hydraulic conductivity (KS) for selected pedotransfer functions, computed on TEST_BASIC and TEST_CHEM+ set. USSAND: sand (50–2000 µm) content (mass %); USSILT: silt (2–50 µm) content (mass %), USCLAY: clay (<2 µm) content (mass %); DEPTH_M: mean soil depth (cm); OC: organic carbon content (mass %); BD: bulk density (g cm$^{-3}$); CACO3: calcium carbonate content (mass %); PH_H2O: pH in water (-); CEC: cation exchange capacity (cmol (+) kg$^{-1}$).

[Figure]

**Figure S16.** Root mean square error (RMSE) of the pedotransfer functions derived to predict parameters of the van Genuchten model for the description of the moisture retention curve (MRC), computed on TEST_BASIC and TEST_CHEM+ set. USSAND: sand (50–2000 μm) content (mass %); USSILT: silt (2–50 μm) content (mass %), USCLAY: clay (<2 μm) content
5    (mass %); DEPTH_M: mean soil depth (cm); OC: organic carbon content (mass %); BD: bulk density (g cm$^{-3}$); CACO3: calcium carbonate content (mass %); PH_H2O: pH in water (-); CEC: cation exchange capacity (cmol (+) kg$^{-1}$).

[Figure]

**Figure S17.** Density plot of observed (OBS) and predicted median (PSD+DEPTH_M+*) water retention values (MRC)
10    computed based on the parameters of the van Genuchten model, computed on TEST_BASIC and TEST_CHEM+ set. Predicted values of those PTFs are shown which use the most often available predictor variables. USSAND: sand (50–2000 μm) content (mass %); USSILT: silt (2–50 μm) content (mass %), USCLAY: clay (<2 μm) content (mass %); DEPTH_M: mean soil depth (cm); OC: organic carbon content (mass %); BD: bulk density (g cm$^{-3}$); CACO3: calcium carbonate content (mass %); PH_H2O: pH in water (-); CEC: cation exchange capacity (cmol (+) kg$^{-1}$).

[Figure]

[Figure]

**Figure S18.** Mean error of the pedotransfer functions derived to predict parameters of the van Genuchten model for the description of the moisture retention curve, computed on TEST_BASIC (N = 1591) (A) and TEST_CHEM+ (N = 288) (B) sets by matric potential head values.

[Figure]

**Figure S19.** Root mean square error (RMSE) of the pedotransfer functions derived to predict parameters of the Mualem-van Genuchten model for the description of the hydraulic conductivity curve (HCC), computed on TEST_BASIC and TEST_CHEM+ set. USSAND: sand (50–2000 μm) content (mass %); USSILT: silt (2–50 μm) content (mass %), USCLAY: clay (<2 μm) content (mass %); DEPTH_M: mean soil depth (cm); OC: organic carbon content (mass %); BD: bulk density (g cm$^{-3}$); CACO3: calcium carbonate content (mass %); PH_H2O: pH in water (-); CEC: cation exchange capacity (cmol (+) kg$^{-1}$).

[Figure]

**Figure S20.** Density plot of observed (OBS) and predicted median (PSD+DEPTH_M+*) hydraulic conductivity values (HCC) computed based on the parameters of the Mualem-van Genuchten model, computed on TEST_BASIC and TEST_CHEM+ set. Predicted values of those PTFs are shown which use the most often available predictor variables. USSAND: sand (50–2000 μm) content (mass %); USSILT: silt (2–50 μm) content (mass %), USCLAY: clay (<2 μm) content (mass %); DEPTH_M: mean soil depth (cm); OC: organic carbon content (mass %); BD: bulk density (g cm$^{-3}$); CACO3: calcium carbonate content (mass %); PH_H2O: pH in water (-); CEC: cation exchange capacity (cmol (+) kg$^{-1}$).

[Figure]

[Figure]

**Figure S21.** Mean error of the pedotransfer functions derived to predict parameters of the Mualem-van Genuchten model for the description of the hydraulic conductivity curve, computed on TEST_BASIC (N = 176) (A) and TEST_CHEM+ (N = 57) (B) sets by matric potential head values.

---

## Author Response (AR3)

**Revision of manuscript on "Updated European hydraulic pedotransfer functions with communicated uncertainties in the predicted variables (euptfv2)"**

**Content of the document**

| I. AUTHORS' RESPONSE TO THE EDITOR                                   | 1 |
|----------------------------------------------------------------------|---|
| II. LIST OF AUTHOR'S CHANGES IN MANUSCRIPT, SHORT SUMMARY AND ASSETS | 1 |
| III. MARKED-UP MANUSCRIPT AND SHORT SUMMARY VERSION                  | 1 |

**I. AUTHORS' RESPONSE TO THE EDITOR**

Thank you for accepting our manuscript and all your valuable comments, which helped us to improve our manuscript and make our results more user oriented. Special thanks for encouraging us to finalize the package and providing extension of the deadline to let us perform it. This way our results can be used by wider audience.

Due to the addition information about the R package into the manuscript the following documents have been modified:

- the manuscript: Code and data availability and Author contribution sections and reference of the R package was added,
- short summary,
- assests.

Please find more detailed information about the changes in the section below.

**II. LIST OF AUTHOR'S CHANGES IN MANUSCRIPT, SHORT SUMMARY AND ASSETS**

Page and line numbering refer to that of the revised manuscript with track changes included under III. MARKED-UP MANUSCRIPT VERSION AND SHORT SUMMARY section of this document.

**The following changes have been made in the manuscript:**

- P15 L9-11: information about euptf2 R package added,
- P15 L15-16: authors' contributions in R package development added,
- P19 L23-24: reference of R package added by Mendeley, therefore change is not visible in track changes.

**The following changes have been made in the short summary:**

– information on R package added.

**The following changes have been made in the assets:**

- information on R package added to "Model code and software".

**III. MARKED-UP MANUSCRIPT AND SHORT SUMMARY VERSION**

Please find revised marked-up manuscript and short summary on the following pages.

[revised manuscript text omitted]
 0.068 cm3 cm-3 for THS (Table 2), 0.046 and 0.055 cm3 cm-3 for FC (Table 3), 0.040 and 0.060 cm3 cm-3 for FC\_2 (Table 4), 0.037 and 0.048 cm3 cm-3 for WP (Table 5), 0.043 and 0.053 cm3 cm-3 for AWC (Table S1), 0.045 and 0.060 cm3 cm-3 for AWC\_2 (Table S2), and 0.09 and 1.18 log10 (cm day-1) for KS (Table 6) in the case of including different predictor variables computed on the test sets. Table S3 shows the NRMSE for the point predictions
- 20 computed for the TEST\_BASIC and TEST\_CHEM+ sets to provide possibility for comparison with other PTFs available from the literature. In the case of VG and MVG, RMSE for the entire matric potential head range was between 0.041 and 0.068 cm3 cm-3 for the moisture retention (Table 7) and 0.61 and 0.71 log10 (cm day-1) for the hydraulic conductivity (Table 8). These RMSE values are within the range of recently published PTFs (McNeill et al., 2018; Nguyen et al., 2017; Román Dobarco et al., 2019; Zhang and Schaap, 2017).
- In the case of the point estimations, Figures 2, S1 depict the scatterplots of measured and predicted soil hydraulic parameters with 90% prediction interval computed on the test sets. Performance of the worst to best PTFs are shown. The addition of predictors that significantly improve the predictions also decreases the uncertainty. The largest reduction in the width of the inner 90% of the prediction interval is visible for THS. Specifically this value decreased from 0.21 to 0.10 cm3 cm-3 for THS, from 0.19 to 0.14 cm3 cm-3 for FC\_2, from 0.17 to 0.14 cm3 cm-3 for FC, from 0.15 to 0.14 cm3 cm-3 for WP, from 0.19 to 0.17

[revised manuscript text omitted]